

# Analysis of model error in forecast errors of Extended Atmospheric Lorenz' 05 Systems and the ECMWF system

Hynek Bednář[1] and Holger Kantz[2]

[1]Department of Atmospheric Physics, Faculty of Mathematics and Physics, Charles University, 18000, Prague, Czech Republic
[2]Max Planck Institute for the Physics of Complex Systems (MPIPKS), D-01187, Dresden, Germany

*Correspondence to*: Hynek Bednář (hynek.bednar@matfyz.cuni.cz)

**Abstract.** The forecast error growth as a function of lead time of atmospheric phenomena is caused by initial and model errors. When studying the initial error growth, it turns out that small scale phenomena, which contribute little to the forecast product, significantly affect the ability to predict this. The question under investigation is whether omitting these atmospheric
phenomena will improve the predictability of the resulting value. The topic is studied in the extended Lorenz (2005) system. This system shows that omitting small spatiotemporal scales will reduce predictability more than modeling it. Generally, a system with model error (omitting phenomena) will not improve predictability. A theory explaining and describing this behavior is developed, with the difference between systems (model error) produced at each time step seen as the error of the initial conditions. The resulting model error is then defined as the sum of the increments of the time evolution of the initial
conditions so defined. The theory is compared to the fit parameters that define the model error in certain approximations of the average forecast error growth. Parameters are interpreted in this context, and the hypotheses are used to estimate the errors described in the theory. It is proposed how to distinguish increments to prediction error growth from small spatiotemporal-scales phenomena and model error. Results are presented for the error growth of the ECMWF system, where a 40% reduction in model error between 1987 and 2011 is calculated based on the developed theory, while over the same time, the instability
of the system with respect to initial condition errors has grown.

## 1. Introduction

Forecast errors in numerical weather prediction systems grow in time due to the inaccuracy of the initial state (initial error), amplified by the chaotic nature of the system itself and the model imperfections (model error). In the setting of classical low-dimensional chaos, one would observe an exponential error growth of any tiny initial error whose exponent is given by the
largest Lyapunov exponent of the system, with some saturation when the error reaches the magnitude of the standard deviation of the quantity to be predicted. In contrast to this, several authors have observed in the past (Toth and Kalnay, 1993; Lorenz, 1969; Aurell et al., 1996, 1997; Boffetta et al., 1998) that the proper Lyapunov exponent of a dynamical system might not be a relevant description of the initial error growth. Brisch and Kantz (2019) and Zhang et al. (2019) associated initial error growth with scale-dependent error growth, where tiny errors grow much faster than larger ones. Lorenz (1996) gave a sketch of such





error growth: a typical quantity to be predicted is a superposition of the dynamics on different scales. After a fast growth of the small-scale errors with saturation at these very same small scales, the large-scale errors continue to grow at a slower rate until even these saturate. Therefore, Lyapunov exponents of structures of various spatiotemporal scales are taken as the previously mentioned scale-dependent quantity, and they determine the error growth on their respective scales. It is also evident that in practice, initial errors are not infinitesimal in the mathematical sense, and therefore the exponential growth of

infinitesimal errors might be irrelevant for the growth of forecast errors in operational weather forecasts.

    In numerical weather predictions, the average forecast error as function of lead time is influenced by many deviations from a simple exponential growth: there is the saturation effect of larger errors, the potential scale dependence of the instability, and also the fact that real initial errors are not infinitesimal and might not point into the locally most unstable direction. Moreover, if the model error is due to neglecting small scale and fast phenomena, it is highly fluctuating along the model trajectory. In

order to describe such effects, the literature contains different phenomenological approximations for the time derivative of the average error magnitude (error growth rate or tendency) as a function of the error magnitude in numerical weather predictions. We will briefly recall these here since we will use suitable fits to the observed error growth as function of error magnitude for our atmospheric model simulations later. We will show that initial errors of magnitudes that are comparable to real weather forecasts do not play a dominant role in our studied model systems, while model errors do. Moreover, we will present an

explanation of the observed error growth in terms of an averaged model error called the *drift*.

    In low-dimensional chaotic systems with at least one positive Lyapunov exponent, the growth of infinitesimal errors is exponential, given by a linear time derivative:

$$dE_{\exp} = \lambda_{\exp} E dt, \quad E_{\exp}(t) = E_0 e^{\lambda_{\exp} t}, \tag{1}$$

    where $E(t)$ is the error magnitude, $t$ is time, and $\lambda$ is the largest Lyapunov exponent of the system. Since $E_{\exp}(t)$ in Eq. (1)

grows unboundedly, this can be true only as long as $E(t)$ is small since every error has to saturate at the latest when it has grown to the order of magnitude of the diameter of the attractor (the invariant set). This saturation effect was considered by Lorenz (1982), who introduced the *quadratic hypothesis* $E_{qu}(t)$:

$$dE_{qu} = \lambda_{qu} E \cdot \left(1 - \frac{E}{E_{lim}}\right) dt, \quad E_{qu}(t) = \frac{E_0 E_{lim}}{E_0 + (E_{lim} - E_0) e^{-\lambda_{qu} t}}, \tag{2}$$

    where $E_{lim}$ is the limit (saturation) value of the error magnitude. As a function of time, the error $E_{qu}(t)$ shows a sigmoidal

shape, see Fig. 5b.

    For a scale-dependent error growth in the spirit of Lorenz (1996), Brisch and Kantz (2019) proposed using a power law divergence of the effective, scale-dependent Lyapunov exponent $\lambda(E) \propto E^{-b}$ which gives the time evolution:

$$dE_p = a E^{1-b} dt, \quad E_p(t) = \left(E_0^b + abt\right)^{1/b}, \tag{3}$$





where the exponent $b$ connects Lyapunov exponents and limit errors of the different scales (Brisch and Kantz, 2019), and the

coefficient $a$ determines the degree of the scales' coupling (Bednar and Kantz, 2022). The forecast error then grows as a power

law in time, $E_p(t)$, with a very fast growth rate when it is still small and a slow growth rate when it is large. Since $E_p(t)$, Eq.

(3), again grows unboundedly, Bednar and Kantz (2022) introduced the extended power law $E_{ep}(t)$ that allows saturation

using the same trick as Lorenz (1982):

$$dE_{ep} = aE^{1-b}\left(1 - \frac{E}{E_{lim}}\right)dt. \tag{4}$$


Zhang et al. (2019) described scale-dependent error growth differently. They took a two-parametric hypothesis:

$$dE_r = \left(\lambda_r E + \beta_r\right)dt, \quad E_r(t) = E_0 e^{\lambda_r t} + \frac{\beta_r}{\lambda_r}\left(e^{\lambda_r t} - 1\right), \tag{5}$$

where $\lambda_r$ is a synoptic-scale error growth rate and $\beta_r$ is an upscale error growth rate from small-scale processes. If $\beta_r / \lambda_r$

is large, this leads to a super-exponential growth of small errors and to the classical exponential error growth when $E_r(t)$ is

large. We can and should again include saturation of the error by the factor $(1 - E / E_{lim})$:

$$dE_q = \left(\lambda_q E + \beta_q\right) \cdot \left(1 - \frac{E}{E_{lim}}\right)dt. \tag{6}$$

The two-parametric model $dE_r$ Eq. (5) was originally designed to describe initial and model error growth (Leith, 1978). In

this interpretation, $\lambda_r$ is the largest Lyapunov exponent of the system (similarly to $\lambda_{exp}$ in Eq.(1)), and $\beta_r$ is the model error

source term due to the imperfect representation of the atmosphere. Also, $dE_q$, Eq.(6) is called the quadratic hypothesis with

model error for the same reason as for $dE_r$ (Savijarvi, 1995; Dalcher and Kalnay, 1987), although compared to Eq.(2), it

includes a constant term and therefore allows for some skewness in $dE / dt$ as a function of $E$.

When we will present the results of our error growth numerical analysis, we will particularly present the error growth rate as

a function of error magnitude. This allows us to better distinguish between these different error growth models than studying

the error magnitude as a function of time, even though error magnitude as function of time is relevant in predictions.

While the above-listed error growth laws are supposed to approximate the effectively observed average error growth in

operational forecasts, let us now focus on the model error. The model is given as a set of the first order in time differential

equations of the form $\frac{dX}{dt} = G(X(t))$ where $X$ is a (high dimensional) phase space vector which describes the current state

of the atmosphere and $G(X)$ is a vector-valued function which defines the rate of change of this vector at every possible state.

In operational weather forecasts, the core of such a system in given by the six variables wind speed, pressure, density, and

temperature, and the minimal setting for $G$ is then called the "primitive equations" (Phillips, 1973). Following (part of) the



meteorological literature, we will call the right-hand side $G(X)$ the *model tendency*, while in the context of dynamical systems, it is called the *vector field*.

Following, e.g., Orrell et al. (2001), the model error $G_e$ at a model space point $\tilde{X}(t)$ is described as the difference between the model vector field (tendency) $\frac{dX}{dt} = G(X(t))$ and the observed time derivative (tendency) $\frac{d\tilde{X}}{dt}$ of the projection

$\tilde{X}(t) := P\left(\hat{X}(t)\right)$ of the "reality" $\hat{X}(t)$ into the model space:

$$G_e\left(\tilde{X}(t)\right) = G\left(\tilde{X}(t)\right) - \frac{d\tilde{X}(t)}{dt}. \tag{7}$$

Let us stress that in operational forecasts, since we do not know the perfect model, the true time derivative $\frac{d\tilde{X}}{dt}$ is only known by observation, while in our later model studies, we have a mathematical expression for the vector field of "reality" as well. In a very strong simplification, one could assume that the absolute value of $G_e$ is, on average, the constant $b$ in Eq. (5) which,

irrespective of initial condition errors, will lead to a deviation of the model solution from reality. While it is evident how to define the model error in a single time step, we will later discuss how model errors propagate in time, how model errors at different positions along a trajectory accumulate, and also introduce the notion of drift for that purpose.

The main issue about forecasts is how far into the future they might be useful. The *prediction horizon* quantifies this as the time when the forecast error has grown to a certain percentage of the climatological uncertainty of the forecast target, where

the latter is approximated by $E_{lim}$ in the above error growth assumptions.

For exponential growth $E_{exp}(t) = E_0 e^{\lambda_{exp} t}$ and for an initial error $E_0$ going to zero, the time $t_{lim}$ at which the error reaches a limiting value $E_{lim}$, goes to infinity:

$$t_{lim} = \frac{\ln E_{lim} - \ln E_0}{\lambda_{exp}} \to \infty \;\; for \; E_0 \to 0. \tag{8}$$

However, a strict predictability limit $t_{lim}$ exists for scale-dependent error growth even when the initial error $E_0$ vanishes

(Palmer et al., 2014; Brisch and Kantz, 2019). For a description by a power law $dE_p$, Eq. (3), the predictability limit $t_{lim}$ is:

$$t = \left(E^b(t) - E_0^b\right)/\left(a \cdot b\right) \;\; \to \;\; t_{lim} = E_{lim}^b / (a \cdot b) < \infty \;\; for \; E_0 \to 0. \tag{9}$$

For an exponential growth with a non-zero $\beta_r$ parameter $dE_r$, Eq. (5), the prediction horizon $t_{lim}$ is:

$$t(E) = \frac{1}{\lambda}\left(\ln(E + \frac{\beta}{\lambda}) - \ln(E_0 + \frac{\beta}{\lambda})\right) \;\; \to \;\; t_{lim} = \frac{1}{\lambda}\ln\left(\frac{\lambda}{\beta}E_{lim} + 1\right) < \infty \;\; for \; E_0 \to 0. \tag{10}$$

Scale-dependent error growth implies that both model assumptions $E_p$ and $E_r$ grow faster than exponentially when errors are

small, thereby limiting the prediction horizon because further and further improvements of the precision of the initial condition





are compensated by a faster initial error growth. In the context of weather prediction, this means that the influence of small scale atmospheric phenomena, which contribute little to the final value, significantly affect the ability to predict this value. Figs. 1 - 3 show such behavior simulated in the extended Lorenz (2005) system with one (L05-1), two (L05-2), and three (L05-3) spatiotemporal scales (see Appendix A for more information on these systems). Fig. 1a shows the values of the L05-1

system variables at a given time. Because this is a single spatiotemporal scaled system, the average growth of the initially small error is exponential. The two initially nearby trajectories begin to diverge significantly in this setting after about 30 days (Fig. 1b). Adding a considerably smaller scale (L05-2 system) that does not significantly affect the overall value in sum (Figure 2a) reduces the closeness of the two initially nearby trajectories of an overall variable by ten days (Figure 2b). By adding a third medium scale (Figure 3a, L05-3 system), the two initially nearby trajectories of an overall variable start to diverge

significantly in about ten days (Figure 3b), which is about three times earlier than for the L05-1 system. This is a consequence of the much faster growth of the small scale errors.

Including small spatiotemporal scales, i.e., improving the model's spatial and temporal resolution, therefore enhances the instability with respect to initial condition errors. The question under investigation in this paper is whether omitting small scale atmospheric phenomena, which contribute little to the final value, will improve the predictability of the resulting value. In

other words, how does the average forecast error growth change in a model where small-scale phenomena are omitted but where model errors are therefore introduced, compared to a model where all phenomena are present but the average forecast error growth is scale-dependent. We will study this with the help of the one- and two-scale Lorenz-2005 (L05-1 and L05-2) model, Lorenz (2005), and its three-scale extension L05-3 introduced before in Bednar and Kantz (2022). Its definition and further details can be found in Appendix A.

The protocol how to measure the initial error growth for the L05 systems is defined in Section 2.1. and the results are presented and compared in Section 3.1. The model error scenario, where the L05-1 system is the model and the L05-2 and L05-3 systems are "reality," is defined in Section 2.2, and the results are presented and compared in Section 3.2. The variant with initial and model error is defined in Section 2.3, and the results are presented and compared in Section 3.3. The results of different error growth scenarios are compared and discussed in Section 3.4. Section 4.1 includes the calculation of the model error (drift)

defined in Section 2.4 and a theory linking the error so defined with the growth of the average model error determined by the difference between model and "reality." The meaning of the model error source $\beta$ in $dE_r$ of Eq.(5) and $dE_q$ Eq.(6) and how to link the value of $\beta$ with the value of the model error (drift) is discussed and explained in Section 4.2. Section 5 presents a similar analysis for the ECMWF forecast system data. Conclusions and discussions are then presented in the final section.

## 2. Error growth in the L05 systems - types and methods of calculation

The average error magnitude for L05 systems is calculated numerically using the method introduced by Lorenz (1996; 2005). Generally, we define an "error" as the distance between two trajectories where one, the reference trajectory, is supposed to be the "truth," and a second trajectory is generated either under perturbation of the initial condition or under perturbation of the



dynamical equations, or both. We measure the error magnitude $e(t)$ after fixed time intervals. We then calculate the mean

error magnitude $E(t)$ after fixed times, calculate the average growth tendency $\frac{dE}{dt}$ during the last time interval, and report

the mean error magnitudes versus time and the mean growth tendencies (rates) versus mean error magnitudes.

An alternative method for calculating scale-dependent error growth is called the "finite size Lyapunov exponent" (Aurell et

al., 1996; 1997; Boffetta et al., 1998; Cencini et al., 2013). In brief, a finite size Lyapunov exponent $\lambda = (1/E)(dE/dt)$ or

finite size error growth tendency $dE/dt$ can be defined as the ergodic average over phase space of the growth rate of

perturbations of a given magnitude $E$, where the growth rate is defined as the inverse of the time $t_f$ needed for the error

magnitude to increase by a pre-defined factor $f$, hence $\lambda(E) = (1/t_f)\ln f$. We choose the former method because it is closer

to the process of calculating the average forecast error magnitude of numerical weather prediction systems (Lorenz, 1982;

Savijarvi, 1994; Froude et al., 2013; Zhang et al., 2019) and because it is more consistent with the performance of forecasts.

For numerical weather prediction systems, errors in initial conditions and model errors (inaccurate representation of

atmospheric processes by the model) are sources of prediction inaccuracy. For the L05 systems, we simulate the initial error

growth (perfect model assumption), the model error growth (perfect initial conditions assumption), a combination of both

(initial and model error assumption), and the model error growth as defined by Orrel et al. (2001) (drift assumption). To

calculate the average error magnitude, a reference trajectory (considered the "truth" or verification) and a trajectory which is

the numerical solution of the systems with a given error, are repeatedly generated. For this scheme to be meaningful, we have

to ensure that the reference trajectory is on the system's attractor and that the repetition of this scheme samples the whole

attractor with correct weights (the invariant measure). We solve this issue in the following way: We first integrate the system

over ten years (175200 steps), starting from arbitrary initial conditions, and assume that after discarding this transient, the

trajectory is on the attractor. We continue to integrate this single trajectory and consider segments of it as reference trajectories

for error growth, i.e., the many reference trajectories are simply segments of one very long trajectory, which ensures not only

that all these segments are located on the attractor but that in addition, they sample the attractor according to the invariant

measure.

### 2.1. Initial error growth

By "initial error growth," we denote the growth of errors in the initial conditions, which limit predictability if a system is

chaotic. In order to determine numerically the largest Lyapunov exponent, we have to ensure that initial perturbations point

already into the locally most unstable direction since otherwise, errors might even shrink in short times (this is also a relevant

issue in ensemble forecasts, and there find its solutions in using bred vectors (Toth and Kalnay, 1997)). We solve these issues

in the following way: We start with a random perturbation of the reference trajectory of very small amplitude and let this

trajectory evolve over time before determining its distance toward the reference trajectory. In other words, we discard some



initial time interval of error growth since this is affected by some transient behavior before it starts to grow with the maximum

Lyapunov exponent.

We calculate the initial error growth in systems with one (L05-1), two (L05-2), and three (L05-3) scales to illustrate the

behavior in systems with a different number of spatiotemporal scales. The three spatial scales $X_1$, $X_2$, and $X_3$ for the L05-3

system and two spatial scales $X_1$, and $X_2$ for the L05-2 system cannot be separated in terms of a coordinate transform but are

intrinsically coupled and superimposed in the variables $X_{tot}$ of the system. The initial conditions of the "reality" for L05-3 and

L05-2 systems are called $X_{tot,0,n}$, from which one finds $X_{1,0,n}$, $X_{2,0,n}$, and $X_{3,0,n}$ through Eqs. (A10), (A11), and (A12) for the

L05-3 system, and $X_{1,0,n}$, $X_{2,0,n}$ through Eqs. (A4), and (A5) for the L05-2 system. The initial conditions of the "reality" for

the L05-1 system are called $X_{0,n}$. The initial values of the "prediction" are then for the L05-1 system $X'_{0,n} = X_{0,n} + e_n(0)$,

where $e_n(0)$ are the initial errors randomly selected from the normal distribution $ND(\mu = 0; \sigma = 0.01)$. Since the system's

state $X_{tot}$ is the sum over all spatiotemporal components, for L05-3 and L05-2 systems, any arbitrary but small error with

spatially uncorrelated components affects only the smallest scale component. Only a spatially correlated initial error would

appear in another component. However, since this error would immediately propagate into the small-scale variables and then

grow fastest in these, a perturbation with initial errors in the smallest scale component is the only practical choice. The initial

values of the "prediction" for the L05-3 system are then $X'_{tot,0,n} = X_{1,0,n} + X_{2,0,n} + X_{3,0,n} + e_{3,n} = X_{tot,0,n} + e_{3,n}$, where $e_{3,n}(0)$ are

the initial errors randomly selected from the normal distribution $ND(\mu = 0; \sigma = 0.001)$. Randomly selected from the same

normal distribution are also the initial errors $e_{2,n}(0)$ of the L05-2 system, where the initial values of the "prediction" are

$X'_{tot,0,n} = X_{1,0,n} + X_{2,0,n} + e_{2,n} = X_{tot,0,n} + e_{2,n}$.

From the initial values of "reality" and "prediction," we integrate the L05 systems equations (Eqs. (A1), (A8), and (A9)) for

41.7 days ($K = 2000$ steps). In each time step $k$ of the numerical integration, $X_{\tau,k,n}$, and $X'_{\tau,k,n}$ are obtained. The size of the

error at a given time $k\Delta t$ is $e_{\tau,n}(k \cdot \Delta t) = X'_{\tau,k,n} - X_{\tau,k,n}$, where $k = 1,\ldots,K$, $n = 1,\ldots,N$ ($N = 360$ variables for all used

systems). $\tau$ defines a scale, or sum of scales ($\tau = tot, 1, 2, 3$ (L05-3 system), $\tau = tot, 1, 2$ (L05-2 system)) and is therefore

omitted for the L05-1 system. We perform $M = 400$ runs to calculate the average error growth. In each new run, the initial

values $X_{\tau,0,n}$ are the last values $X_{\tau,K,n}$ of the previous run. The average initial error growth $E(t)$ is defined as the geometric

mean of the runs of the Euclidean distances between "reality" and "prediction":

$$E_{tot}(k \cdot \Delta t) = \sqrt[2M]{\prod_{m=1}^{M}\left(\frac{1}{N}\sum_{n=1}^{N}e_{\tau,n,m}^{2}(k \cdot \Delta t)\right)}. \tag{11}$$





As a result, we have numerical averages for the error growth as a function of time steps after perturbing the reference
trajectories in the full phase space and for each scale. We can convert these results into the error growth tendency (rate) as a
function of the error magnitude.

## 2.2. Model error growth

By "model error growth," we denote the growth of errors caused by the inaccurate description of "reality" by the "model."
This inaccuracy involves small-scale atmospheric processes unresolved by the model, which for numerical weather prediction
systems are approximated to the resolved scale by a procedure called parameterization. It also denotes model biases that are
either unknown or have not yet been addressed (Allen et al., 2006). It is a common expectation that model errors in numerical
weather forecasts can be reduced by improving the spatial and temporal resolution of the forecast system.

To simulate this in the L05 systems environment (Appendix A), we use the L05-2 and L05-3 systems as the "reality" and the
L05-1 system as the "model." Thus, the unresolved or unknown scale is the small scale for the L05-2 system and the small and
medium scale for the L05-3 system. This approach is justified by the fact that the L05-2 and L05-3 systems can be viewed as
a variant of the L05-1 system:

$$dX_{tot,n}/dt = [X_1, X_1]_{L,n} - X_{1,n} + \tilde{F}_n(t), \tag{12}$$

where $\tilde{F}_n(t) = b^2[X_2, X_2]_{1,n} + c[X_2, X_1]_{1,n} - bX_{2,n} + F$ for the L05-2 system and

$\tilde{F}_n(t) = b_1^2[X_2, X_2]_{1,n} + b_2^2[X_3, X_3]_{1,n} + c_1[X_2, X_1]_{1,n} + c_2[X_3, X_2]_{1,n} - b_1X_{2,n} - b_2X_{3,n} + F$ for the L05-3 system are treated as a

forcing, which varies in a complicated manner with time. We parameterize these small-scale phenomena contained in $\tilde{F}_n(t)$
by the average value of these phenomena, which is close to zero, and therefore we can write:

$$\langle \tilde{F}_n(t) \rangle \approx F = 15, \tag{13}$$

where $\langle \ldots \rangle$ represents the mean calculated over a long orbit on the L05-2 and L05-3 systems attractors.

To calculate the average model error growth, we first define initial conditions that are the same for "model" and "reality"
(perfect initial conditions assumption) and are determined from the values $X_{tot,0,n}$ of "reality" (L05-2 or L05-3 systems) at the
end of the initial transient. Let us stress that we can use $X_{tot,0,n}$ of our high-resolution L05-3 or L05-2 system without any
projection as the initial state of the L05-1 system and that the lack of smaller scales is only expressed by the lack of feedback
from the smaller scales in the equation of motion.

From these initial values, we integrate forward the L05-2 or L05-3 systems equations ("reality") and the L05-1 system
equations ("model") for 41.7 days ($K = 2000$ steps). In each time step $k$ of the numerical integration, $X_{tot,k,n}$ ("reality") and
$X_{k,n}$ ("model") are obtained. The size of the error at a given time $k\Delta t$ is $e_{M,n}(k \cdot \Delta t) = X_{tot,k,n} - X_{k,n}$, where $k = 1,\ldots,K$,
$n = 1,\ldots,N$ ($N = 360$ variables for all used systems). We perform $L = 400$ runs to calculate the average error growth. In





each new run, the initial values $X_{tot,0,n}$ are the last values $X_{tot,K,n}$ of the previous run. The average model error growth $E_M(t)$ is defined as the geometric mean of the runs of the Euclidean distances between "reality" and "model":


$$E_M(k \cdot \Delta t) = \sqrt[2L]{\prod_{l=1}^{L}\left(\frac{1}{N}\sum_{n=1}^{N}e_{M,n,l}^2(k \cdot \Delta t)\right)}. \tag{14}$$

As a result, we have numerical averages for the model error growth as a function of time steps. Note that in this framework, only $X_{tot,k,n}$ (L05-2 and L05-3 systems) are compared to $X_{k,n}$ (the L05-1 system) and not the individual scales. We can convert these results into the error growth tendency (rate) as a function of the error magnitude.

### 2.3.  Initial and model error growth

By "initial and model error growth," we denote the combination of the initial error growth defined in Section 2.1. and the model error growth defined in Section 2.2. We describe the L05-2 and L05-3 systems as "reality" and the L05-1 system with perturbations in the initial conditions of "reality" as "model prediction."

In this setting, we do not discard the initial time interval of initial error growth because this transition period is negligible compared to the model error growth. The initial conditions of the "reality" for L05-3 and L05-2 systems are called $X_{tot,0,n}$ and

determined in the same way described above. The initial values of the "model prediction" for the L05-1 system are then $X'_{0,n} = X_{tot,0,n} + e_n(0)$, where $e_n(0)$ are the initial errors randomly selected from the normal distributions $ND(\mu=0; \sigma=0.01)$ and $ND(\mu=0; \sigma=0.2)$. From initial values, we integrate forward the L05-2 or L05-3 systems equations ("reality") and the L05-1 systems equations ("model prediction") for 41.7 days ($K=2000$ steps). In each time step $k$ of the numerical integration, $X_{tot,k,n}$ ("reality") and $X'_{k,n}$ ("model prediction") are obtained. The size of the error at a given time $k\Delta t$ is

$e_{M+ie,n}(k \cdot \Delta t) = X_{tot,k,n} - X'_{k,n}$, where $k=1,\dots,K$, $n=1,\dots,N$ ($N=360$ variables for all used systems). We perform $L=400$ runs to calculate the average error growth. In each new run, the initial values $X_{tot,0,n}$ are the last values $X_{tot,K,n}$ of the previous run. The average initial and model error growth $E_{M+ie}(t)$ is defined as the geometric mean of the runs of the Euclidean distances between "reality" and "model prediction":

$$E_{M+ie}(k \cdot \Delta t) = \sqrt[2L]{\prod_{l=1}^{L}\left(\frac{1}{N}\sum_{n=1}^{N}e_{M+ie,n,l}^2(k \cdot \Delta t)\right)}. \tag{15}$$

As a result, we have numerical averages for the initial and model error growth as a function of time steps. Note that in this framework, only $X_{tot,k,n}$ (L05-2 and L05-3 systems) are compared to $X_{k,n}$ (the L05-1 system) and not the individual scales. We can convert these results into the error growth tendency (rate) as a function of the error magnitude.



## 2.4. Drift

Section 2.2 describes how we can numerically *measure* the effects of the model error on forecast accuracy. However, if we
want to *understand* how the model error drives the model trajectory away from reality, we need an additional concept. The
reason is that model errors at different positions along the trajectory are only weakly correlated. This is a consequence of the
fact that the lack of small scales and fast degrees of freedom in the model equations dominates model errors. But if model
errors at different positions along a trajectory are uncorrelated, then they can partially compensate each other, and their effect
is not the same as if we assume that model errors along a trajectory are everywhere about the same. Therefore, We will recall
the concept of *drift D* as Orrell et al. (2001) discussed. For these purposes, let us first generally define the "model" (L05-1
system in our case) as $d\vec{X}(t)/dt = \vec{G}(\vec{X}(t))$ where $\vec{X} \in \mathbb{R}^n$ is the "model" state space vector ($n = 360$ in our case) and the
"reality" state space vector $\vec{\tilde{X}}(t) \in \mathbb{R}^{\tilde{n}}$. In general, $\tilde{n} \neq n$ and it is necessary to project $\vec{\tilde{X}}$ from the state space of "reality" to
the state space of "model" (Data Assimilation for Numerical Prediction Models). In our case, $\tilde{n} = n = 360$, $\vec{\tilde{X}} = \vec{X}_{tot}$, and we
use either the L05-2 system $d\vec{X}_{tot}(t)/dt = \vec{\tilde{G}}(\vec{X}_1(t), \vec{X}_2(t))$ or the L05-3 system $d\vec{X}_{tot}(t)/dt = \vec{\tilde{\tilde{G}}}(\vec{X}_1(t), \vec{X}_2(t), \vec{X}_3(t))$ as
"reality." The model error $\vec{G}_e$ at the point $\vec{X}_{tot}(t)$ is then the difference between the "model" vector field (tendency) and the
tendency of the projection of "reality" into the "model" space. In our case, we can write:

$$\vec{G}_e(\vec{X}_{tot}(t)) = \vec{G}(\vec{X}_{tot}(t)) - \frac{d\vec{X}_{tot}(t)}{dt}. \tag{16}$$

The drift vector $\vec{d}(\tau)$ was introduced by Orrell et al. (2001) as

$$\vec{d}(\tau) = \int_0^\tau \vec{G}_e(\vec{X}_{tot}(t)) dt = \int_0^\tau \vec{G}(\vec{X}_{tot}(t)) dt - \vec{X}_{tot}(\tau) + \vec{X}_{tot}(0). \tag{17}$$

This is an accumulation of model errors along a piece of the model trajectory. As we will see in numerical simulations, the
drift $d(\tau)$ will not grow approximately linearly in time, i.e., it is not the same as accumulating the absolute value of the model
error $|G_e|$ along the same piece of the trajectory.

This is a consequence of the here considered case of neglected small scale motion: Since the ignored scales fluctuate fast, the
model errors at successive positions on the trajectory lose their correlations. We checked this for our L05-models explicitly by
calculating the auto-correlation function of the drift vectors as a function of their time lag and found a very fast decay within
a few time steps. Therefore, different from model errors in low-dimensional systems which can be assumed to be spatially
highly correlated, one here accumulates random vectors, and the drift, therefore, follows a path that resembles a Brownian
path, as already suggested in Orrel et al. (2001). There and in Orrell (2002), it is also shown how to approximate the integral
by summing a series of short-time model errors over finite time steps $\Delta t$. The drift $D(\tau)$ as a function of $\tau$ grows sub-
linearly, as will be demonstrated later and gives a more realistic estimate of the role of model errors. What, however, is ignored





here is that a model error in the first time step creates a kind of initial condition error for the second time step, which then would grow as an initial condition error. We will discuss this later.

To calculate the average drift $D$ comparable to previous cases, we first calculate the time evolution of "reality" $X_{tot,k,n}$ (L05-2 or L05-3 systems), calculated from the initial conditions after the transient period. From each time step $k$ of the time evolution of $X_{tot,k,n}$ "reality" (up to $K = 2000$ steps), we calculate the one-step $\Delta t$ time evolution of the "model." $X_{tot,k,n} = X_{k,n}$ are therefore viewed as initial conditions for the one-step $\Delta t$ time evolution of the "model." The size of the drift at a given time $k\Delta t$ is the sum of all previous and current error vectors: $d_n(k \cdot \Delta t) = \sum_{j=0}^{k-1}\left(X_{j,n}((j+1) \cdot \Delta t) - X_{tot,j+1,n}\right)$, where $k = 1,\ldots,K$, $n = 1,\ldots,N$ ($N = 360$ variables for all used systems). Notice that it is not the absolute value of the $\Delta t$-errors which are accumulated but the vectors (see Fig. 4), so that in the summation, there can be cancellation effects and hence a slower-than-linear growth of the drift with time.

We perform $L = 400$ runs in order to calculate the average error growth. In each new run, the initial values $X_{tot,0,n}$ are the last values $X_{tot,K,n}$ of the previous run. The average drift $D(t)$ is defined as the geometric mean of the runs of the Euclidean distances between "reality" and "model":

$$D(k \cdot \Delta t) = \sqrt[2L]{\prod_{l=1}^{L}\left(\frac{1}{N}\sum_{n=1}^{N}d_{n,l}^2(k \cdot \Delta t)\right)}. \tag{18}$$

As a result, we have numerical averages for the drift as a function of time steps. We can convert these results into the drift growth tendency as a function of the drift magnitude.

## 3. Error growth in the L05 systems – results and comparisons

Based on the described methods, we calculate the average prediction error growth for L05 systems. We approximate the numerical error growth curves using the hypotheses or laws Eqs. (1) - (6) and try to identify the most appropriate description. We use these results to determine how the average forecast error growth changes in a "model" where small-scale phenomena are omitted, but the model error is therefore created (perfect initial conditions assumption or initial and model error assumption) compared to a "model" where all phenomena are present, but the average forecast error growth is scale-dependent (perfect model assumption). The resulting behavior will be explained using the drift.

### 3.1. Initial error growth

Fig. 5a shows the initial error growth rate (tendency) $dE/dt$ as a function of the error magnitude $E$ for the L05-1 system, while Fig. 5b shows the error magnitude as a function of time. We also show the best fit results of the error growth models represented by Eqs. (1) to (6). It turns out that the initial part of the growth rate is linear without any significant offset, i.e., we



see a linear increase with a beginning at $(E = 0, dE / dt = 0)$. Therefore, constants in the error growth models which were included to represent the model error are consequently close to zero. Also, the power law fit yields a power close to 1. Because

of the saturation of the error at large times, the error growth rate decays to zero when the error is large, which can be well represented by the factor $(1 - E / E_{lim})$ in the error growth models. Hence, all models with this saturation term allow good fits to the error growth rate and the error magnitude as a function of time in the whole range and confirm that the L05-1 system is a classical chaotic system with the largest Lyapunov exponent of about $\lambda \approx 0.33$ 1/ day.

The behavior is obviously different for the L05-2 system, which contains additionally small scale degrees of freedom, as shown

in Fig. 6. Already, the initial part of the error growth rate (for small $E$) is curved, hence the exponential growth model does not anymore provide a good fit. Introducing a non-vanishing error growth rate right from the beginning, i.e., starting from $(E = 0, dE / dt = \beta_r)$ which is the description by $dE_r$, the approximation moves closer to the data, but this is in clear contradiction to the initial error growth idea: Due to the lack of model errors, the growth rate starts from 0. Also, the quadratic hypothesis is unable to reproduce this curvature well enough. Therefore, the data are best approximated by the power law in

the initial part and by the extended power law with saturation on the whole range.

What we found for the initial error growth of the L05-2 system is even more pronounced in the L05-3 system with three spatiotemporal scales. The superiority of approximations $dE_p$ and $dE_{ep}$ over the other approximations is enhanced by the even faster growth of $E_{tot}(t)$ compared to the exponential growth and $E_{tot}(t)$ for the L05-2 system (Fig. 7). The reason for this behavior is described in Brisch and Kantz (2019) or in Bednar and Kantz (2022). Lorenz's (1996) statement can summarise

it: a typical quantity to be predicted is a superposition of the dynamics on different scales. After a fast growth of the small-scale errors with saturation at these very same small scales, the large-scale errors continue to grow at a slower rate until even these saturate. This is the phenomenon of scale dependent error growth. We also see that if we interpret the three systems L05-1 to L05-3 as low and high resolution models, the high resolution model has larger instability and hence a shorter time until an ensemble of initial conditions has spread out on the attractor. If this were of relevance for the prediction horizon, then the

high-resolution model would be less useful for forecasting than the low resolution model.

### 3.2. Model error growth

We use the L05-2 system as reality and make forecasts using the L05-1 system. Their suitably averaged differences give rise to the model error as a function of lead time. Fig. 8a (full black curve) shows the model error growth rate $dE / dt$ as a function of the error magnitude $E$, while 8b shows the time evolution of this error. We see an initially very fast error growth caused

by the differences of the equations of motion of reality and model. After a short transient, we see in both panels of Fig. 8 a behaviour compatible with our error growth models. Those models with a constant term (i.e., the quadratic hypothesis with model error and the exponential growth with model error) provide the best fits, where, to be good in the whole range, the factor $(1 - E / E_{lim})$ of the quadratic hypothesis with model error is needed. In view of what will follow, we stress that based on the





data, both $E_r$ and $E_q$ provide good fits up to error magnitudes of about three units, with different values $\lambda_q \approx 0.27$ 1/day and
$\lambda_r \approx 0.17$ 1/day of the largest Lyapunov exponent.

When we use L05-3 as "reality" and L05-1 as "model," the same conclusions are valid for the model error growth rate $dE/dt$
as a function of the error magnitude $E$ (Fig. 9a) and $E_M(t)$ (Fig. 9b). Note, however, that the rates $dE/dt(E)$ have much
larger maximal values and that $E_M(t)$ grows faster than when taking L05-2 as "reality." But again, if we ignore the very initial
part of the error growth rate for small values $E$, which the error growth models cannot reproduce, we see that $E_r$ and $E_q$
provide the best fits.

### 3.3. Initial and model error growth

In both settings, we also show the results when we include a small initial condition error in addition to the model error. This
initial condition error implies that the forecast error as a function of time starts with a non-zero value and correspondingly with
a much lower growth rate than the model error alone, but apart from that, there are no strong effects. Figs. 8 and 9 show that
it is not the net sum of the initial error growth $E_{tot}(t)$ and the model error growth $E_M(t)$. $E_{M+ie}(t)$ goes from the initial value
$E_{M+ie}(0)$ through some transition period to the model error growth curve $E_M(t)$. $E_M(t)$ is therefore the limiting value to
which $E_{M+ie}(t)$ is attracted. Indeed, black solid and dashed curves in the insets of Fig. 8b and Fig. 9b show that already, after
time $t = 0.2$ the model error alone has grown so much that there is no effect of even of the larger initial condition error of
magnitude $E(0) = 0.2$ anymore. The larger the initial error and the smaller the model error, the longer the transition period,
but it is still short for realistic values of the initial condition error. For these reasons, it can be seen that the appropriate
approximation for describing the variant with initial and model error remains the same as for describing the variant with model
error only, which is the exponential growth with model error, $dE_r$ Eq. (5), for the early growth phase and the quadratic
hypothesis with model error $dE_q$, Eq. (6), for the entire length of the evolution (Fig. 8 and 9).

### 3.4. Comparison of initial and model error growth

We want to use the approximation formulae to construct the error curves for $E_{tot}(0) = 0$, $= 0.1$, $= 0.2$. The initial error
magnitudes of 0.1 and 0.2 correspond to the relative values of the initial errors of current numerical weather prediction
models for the L05 models. For the initial error growth $E_{tot}(t)$ (for simplicity, let us redefine $E_{tot}(t)$ to $E_{ie}(t)$), we use the
extended power law solution and find the parameter values $dE_{ep} = 0.28 \cdot E^{0.66}(1 - E/7)$ for the L05-2 system and
$dE_{ep} = 0.38 \cdot E^{0.41}(1 - E/7.1)$ for the L05-3 system, with initial values $E_{ie(0)}(0) \rightarrow 0$ (Fig. 10, full red curve), $E_{ie(0.1)}(0) = 0.1$
(Fig. 10a, dashed red curve), and $E_{ie(0.2)}(0) = 0.2$ (Fig. 10b, dashed red curve).



For the model error growth $E_M(t)$, we use the quadratic hypothesis with model error with the following best fit parameters: $dE_q(t)/dt = (0.27 \cdot E + 0.34)(1 - E/7.6)$ and $E_M(0) \approx 0.1$ for the L05-2 system and $dE_q(t)/dt = (0.38 \cdot E + 1.47)(1 - E/7.8)$ with $E_M(0) \approx 0.1$ for the L05-3 system.

The reason why $E_0$ is non-zero when using the approximation can be found in section 4.2. We can use the same
approximations for the initial and model error growth $E_{M+ie}(t)$ as for the model error growth alone, $E_M(t)$ with $E_{M+ie(0.1)}(0) = 0.1$ for the L05-2 system and $E_{M+ie(0.2)}(0) = 0.2$ for the L05-3 system. The justification can be found in Section 4.3.

In addition to the graphical representation (Fig. 10), we compare the variants by expressing the times $t_{95\%}$, $t_{71\%}$, $t_{50\%}$, $t_{25\%}$ when the error magnitude $E$ reaches 95%, 71%, 50%, and 25% of its limiting (saturated) value $E_{\lim}$. In the literature, $t_{95\%}$ is
understood as a practical predictability limit and $t_{71\%}$ corresponds to climatic variability, according to Savijarvi (1995). For Fig. 10, let us first comment on the difference between the limiting (saturation) values $E_{\lim}$ of the initial error curves $E_{ie}(t)$ and the model error curves $E_M(t)$ or $E_{M+ie}(t)$. The pure initial error curves are produced in the perfect model scenario, i.e., forecast and "reality" are obtained with the same system. When model error is present, the variability of forecast and reality is different, and hence the limiting error is larger, in agreement with Simmons et al. (1995) or Li et al. (2018).
Fig. 10 also provides insight into our question of whether high-resolution or low-resolution models will produce better forecasts. We recall that high-resolution models suffer from much faster initial condition error growth due to the stronger instability of small scale motion. However, the small scales usually do not contribute to the forecast target. Fig. 10 shows that predictability is significantly worsened by the model error rather than by the initial error growth. This means that the average forecast error in a "model" where small-scale phenomena are omitted, but the model error is therefore present (black curves
$E_M(t)$ and $E_{M+ie}(t)$) grows significantly faster compared to a "model" where all scales are resolved (no model error), but the average forecast error growth is scale-dependent (red courves $E_{ie}(t)$). However, this phenomenon depends on the relative magnitudes of the model error term $\beta$, the (effective) Lyapunov exponent $\lambda$, and the initial condition error. If the growth rate $\lambda$ were larger and the model error smaller, then the exponential error growth would overwhelm the contribution of model error in every time step. In our numerical experiments using the L05-models, the magnitude of the initial condition error is
tuned to values corresponding to initial condition errors in real weather forecasts, so we are tempted to believe that in weather forecasts, also high-resolution models should be superior.

Specifically, for the L05-2 system (Fig. 10a), the forecast with model error (i.e., using L05-1 for the forecast), the time $t_{25}$ is more than three times shorter than the limit without model and initial condition error ($E_{ie(0)}(0) \to 0$), being $t_{25\%,M} = 4$ days and $t_{25\%,ie(0)} = 13$ days, respectively. With increasing error magnitude, the ratio of forecast times decreases ($t_{50\%,M} = 6$ days vs.
$t_{50\%,ie(0)} = 19$ days and $t_{71\%,M} = 9$ days vs. $t_{71\%,ie(0)} = 24$ days) until $t_{95\%,M} = 16$ days, which is approximately half as large as





$t_{95\%,ie(0)} = 37$ days. Adding the error of the initial condition does not significantly change the error growth for the model error variant ($E_M(t) \approx E_{M+ie(0.1)}(t)$, see Section 4.3 for details). The error growth naturally increases for the variant with the initial error, and the ratio is reduced to twice the growth rate over the entire growth period ($t_{25\%,M+ie(0.1)} = 4$ days vs. $t_{25\%,ie(0.1)} = 9$ days, $t_{50\%,M+ie(0.1)} = 6$ days vs. $t_{50\%,ie(0.1)} = 14$ days, $t_{71\%,M+ie(0.1)} = 9$ days vs. $t_{71\%,ie(0.1)} = 19$ days, and $t_{95\%,M+ie(0.1)} = 16$ days vs.

$t_{95\%,ie(0.1)} = 32$ days).

For the L05-3 system (Fig. 10b) taken as truth, the effect es even more dramatic: Without initial error (respectively for $E_{ie(0)}(0) \to 0$), $t_{25\%,M} = 1$ days is seven times smaller than $t_{25\%,ie(0)} = 7$ days. Gradually, the ratio decreases ($t_{50\%,M} = 2$ days vs. $t_{50\%,ie(0)} = 12$ days, and $t_{71\%,M} = 4$ days vs. $t_{71\%,ie(0)} = 18$ days ) until $t_{95\%,M} = 8$ days, which is approximately four times smaller than $t_{95\%,ie(0)} = 34$ days. Adding the error of the initial condition does not significantly change the error growth for the

model error variant ($E_M(t) \approx E_{M+ie(0.2)}(t)$, see Section 4.3 for details). The growth changes for the variant with the initial error, and the ratio is reduced to five times the growth rate in the first half of growth ($t_{25\%,M+ie(0.2)} = 1$ days vs. $t_{25\%,ie(0.2)} = 5$ days, $t_{50\%,M+ie(0.2)} = 2$ days vs. $t_{50\%,ie(0.2)} = 10$ days, $t_{71\%,M+ie(0.2)} = 4$ days vs. $t_{71\%,ie(0.2)} = 16$ days, and $t_{95\%,M+ie(0.2)} = 8$ days vs. $t_{95\%,ie(0.2)} = 34$ days).

In the so far described numerical error growth study, we used L05-1 as a model and compared its performance to two types of
reality with different spatiotemporal resolutions. We complemented this study by using as a single "reality" the L05-3 system and comparing the performance of the L05-1 and L05-2 models when using then for forecasts as two forecasts with different spatiotemporal resolutions. These findings (not shown) confirm the dominance of the model error over the initial condition error growth, i.e., again, the model with the higher resolution provides better forecasts even for larger errors and therefore has a better prediction horizon. However, the effect in this setting is not as dramatic as in the above presented results.

**4. Error growth in the L05 systems – discussion and explanation of results**

In this section, we present considerations that explain the dominance of the model error in our forecast studies, where the notion of the drift (Section 2.4) plays a relevant role. We will consider the feedback of the drift as an initial error, which modifies the interpretation of the parameters in the exponential growth with model error $dE_r$, Eq. (5), and the quadratic hypothesis with model error $dE_q$, Eq. (6), and its relationship with the drift.





### 4.1. Drift and its role in explaining the model error growth

Fig. 11 compares the model error growth, $E_M(t)$ Eq.(14), the drift $D(t)$ Eq. (18), and the exponential growth approximation with model error $E_r(t)$ Eq. (5). The curves are determined from the difference between the L05-2 and the L05-1 systems (Fig. 11a) and between the L05-3 and the L05-1 systems (Fig. 11b) for error magnitudes where the saturation effect is not yet present (up to 2 and 3 days). We intend to understand the behavior of the model error since this is the issue in real forecasts. The drift $D(t)$ (Fig. 11, blue curve) describes very well the early part of the model error evolution $E_M(t)$ (Fig. 11, black curve), while at longer lead times, it is the exponential growth with model error contribution (Fig.11, black dashed line) which can be well fitted to the model error curve. Notice, however, that the drift is calculated numerically from the simulation data, like the model error curve, and hence does not contain any free parameters, while the exponential growth with model error has two free parameters, which allow us to optimally achieve the agreement between the black and dashed black curves.

In particular, in the L05-3 model, for the times where the model error growth $E_M(t)$ can be fitted by $E_r(t)$, the drift $D(t)$ first overestimates the model error growth $E_M$. Then, a significantly slower growth of the drift $D$ relative to the model error growth $E_M$ is observed, corresponding to a decrease in the growth rate (tendency) of the drift $dD/dt(D)$ compared to the increase in the model error growth rate (tendency) (insets in Fig. 11).

As already mentioned in section 2.4, the drift can be viewed as the sum of the displacements from "reality," created at each time step of the "model," similar to how an error in the initial conditions will create an initial displacement at the initial time. If we interpret the drift increment $D(t_k) - D(t_{k-1})$ at each time step $\Delta t = t_k - t_{k-1}$, $k = 1, \ldots, K$ as a new initial error at time $t_k$, then (similar to Orell et al. (2001)) we can model the model error growth $E_M$ by applying the same time evolution assumption to $D(t_k) - D(t_{k-1})$ as for the initial error growth, i.e., the exponential growth $e^{\lambda_D t}$ driven by the largest Lyapunov exponent $\lambda_D$ of the "model" (L05-1 system). However, this growth should set in only at some time in the future since $\vec{D}(t_k) - \vec{D}(t_{k-1})$ does not point into the locally most unstable direction (see Sections 2.1. and 3.1. for a description of the initial error growth for the L05-1 system). We approximate this behavior in two different ways and will explore which one gives better results. (Fig. 12). In the first, $D(t_k) - D(t_{k-1})$ evolves with time $t_i$ in a constant approximation as:

$$F_{con}\left(t_k; t_i\right) = \begin{cases} 1 & t_k \leq t_i \leq t_{M+k} \\ e^{\lambda_D\left(t_i - t_k\right)} & t_{M+k+1} \leq t_i \leq t_K \end{cases}, \qquad (19)$$

while in a linearly decaying approximation as:

$$F_{lin}\left(t_k; t_i\right) = \begin{cases} 1 - \sigma\left(t_i - t_k\right) & t_k \leq t_i \leq t_{M+k} \\ \left(1 - \sigma\left(t_{M+k} - t_k\right)\right) e^{\lambda_D\left(t_i - t_k\right)} & t_{M+k+1} \leq t_i \leq t_K \end{cases} \qquad (20)$$





$M$ and $\sigma$ are parameters that we fix empirically. We propose the hypothesis $E_D(t)$ as a description of the model error growth $E_M(t)$ based on the sum of the individual increments of the drift:

$$E_M(t_i) \approx E_{D,ap}(t_i) = \sum_{k=1}^{i} \left( D(t_k) - D(t_{k-1}) \right) \cdot F_{ap}(t_k;t_i), \qquad (21)$$

where $ap$ is the symbol for the constant ($con$) or linear ($lin$) approximation.

In comparison to the exponential error growth with model error (with or without saturation), this is a modification in two relevant details: First, the model error is not the same at all time steps into the future like the constant $\beta_r$ or $\beta_q$, but it is the time-into-the-future dependent increment of the drift $D(t)$, and second, the new contribution in a given time step is not amplified exponentially in the next step, but because of not pointing into the locally most unstable direction we let it either constant or even decay for a few time steps.

Fig. 13 demonstrates how well the model error growth can be approximated by $E_D(t)$, particularly by the approximation with an initial linear decay $E_{D,lin}(t)$. As the numerical value of the Lyapunov exponent $\lambda_D$ in the hypotheses $E_D(t)$ we use the one determined as $\lambda_q$ in the quadratic hypothesis with model error $dE_q$, Eq. (6), which is $\lambda_{q,L05-2} = 0.27$ 1/day from the difference between the L05-1 and L05-2 systems (Fig. 8) and $\lambda_{q,L05-3} = 0.38$ 1/day from the difference between the L05-1 and L05-3 systems (Fig. 9). The justification for using the parameter $\lambda_q$ as $\lambda_D$ can be found in Bednar et al. (2021). In short, we argue

that $\lambda_q$ is a better estimate of the Lyapunov exponent of the model system than, e.g., $\lambda_r$. The parameters $M$ and $\sigma$ of the hypotheses $E_D(t)$ are determined empirically. It is, however, a bit puzzling that when using the value $\lambda_q$ in $E_{D,lin}(t)$, we can approximate that part of the model error growth curve $E_M(t)$, which can also be well approximated by the exponential growth with model error $dE_r$ (Fig. 13), then leading to a value of the exponent which is different from $\lambda_q$. Therefore, in the next section, we discuss the relationship between the drift $D$ and parameters $\lambda$ and $\beta$ in $dE_q$ (Eq. (6)) and $dE_r$ (Eq. (5)).

**4.2. Understanding the drift through parameters of the quadratic hypothesis and exponential growth both with model error**

We saw two meaningful approximations to the model error growth curve over lead time: The quadratic hypothesis with model error and the exponential growth with model error. The best fit parameter values of these two approximations are listed in Table 1. This Table shows that $\overline{dD/dt} \approx \beta_q$, but the most evident is the difference of the exponential growth exponent $\lambda$,

where for both "realities" L05-2 and L05-3, the exponent $\lambda_q$ of the quadratic hypothesis is larger than $\lambda_r$ of the best fit exponential error growth.





To understand the meaning of the parameters of the exponential growth with model error $dE_r$ (Fig. 8 and 9), let us first define the model error growth $E_{D,0}(t)$ based on the drift $D$, ignoring the initial decrease caused by $\vec{D}(t)$ not pointing into the locally most unstable direction:


$$E_{D,0}(t_i) = \sum_{k=1}^{i} \left( D(t_k) - D(t_{k-1}) \right) \cdot e^{\lambda_D(t_i - t_k)} \quad t_k \leq t_i \leq t_K, \tag{22}$$

where $\lambda_D$ is the largest Lyapunov exponent of the "model" (L05-1 system). The time derivative (calculated from the difference at successive time steps) of $E_{D,0}(t)$ is:

$$\frac{dE_{D,0}}{dt}\left( E_{D,0} \right) = \lambda_D \cdot E_{D,0} + \frac{dD}{dt}\left( E_{D,0} \right), \tag{23}$$

where $dD/dt\left( E_{D,0}(t_k) \right) = dD/dt(t_k)$, i.e., due to the monotonicity of $E_{D,0}(t)$ in time, we can exchange the dependence on

$t$ by the dependence on $E_{D,0}(t)$ (see Fig. 14). Eq.(23) now claims that the red dashed curve, which is $\lambda_D E$, plus the values of the blue curve taken at corresponding times $t_k$, sum up to yield the red curve $E_D(E(t))$, where we approximate the slope of the linear increase of the red curve by the slope of the black curve, which describes the observed total error.

So we focus now on those parts of the three curves $E > 0.3$ for L02-5 or for $E > 0.5$ for L05-3. As said, we observe that in this range of $E$, $dE_{D,0}/dt(E_{D,0})$ (Fig. 14, red curve) has the same growth rate (tendency) as $dE_M/dt(E_M)$ (Fig. 14, black

curve), which is expressed by $\lambda_r$, fitting the $E_r(t)$ behavior of Eq. (5) to the data (Fig. 14, black dashed curve, $\lambda_{r,L05-2} = 0.17$ 1/day, $\lambda_{r,L05-3} = 0.25$ 1/day) If we compare these parameter values to the fit using the quadratic hypothesis with model error, we see that $\lambda_r$ is smaller than $\lambda_q$ of $dE_q$ ($\lambda_{q,L05-2} = 0.27$ 1/day, $\lambda_{q,L05-3} = 0.38$ 1/day). In our interpretation of the model errors involving the drift, this is due to the decrease of the drift growth rate $dD/dt\left( D(t_k) \right)$ over time (Fig. 14, blue curve). Hence, $\beta_r$ of the $dE_r$ approximation of $dE_{D,0}/dt(E_{D,0})$ is then an extrapolation of the linear decline of $dD/dt\left( E_{D,0}(t_k) \right)$ to

$E_{D,0}(t_0) = 0$. Therefore, if we solve Eq. (23) for $dD/dt$ and define $dD/dt\left( E_{D,0}(t_k) \right) \approx \beta_D - \alpha_D E_{D,0}$, then $\beta_r = \beta_D$. Eq. (23) can be used to determine the drift decrease rate $\alpha_D$: $\lambda_r E_{D,0} + \beta_r = \lambda_q E_{D,0} - \alpha_D E_{D,0} + \beta_D \to \alpha_D = \left( \lambda_q - \lambda_r \right)$. However, $\beta_r = \beta_D$ is not the same as $\beta_r$ in $dE_M/dt$ (Fig. 14, black curve) because it is reduced by the transition term expressed by Eqs. (20) and (21) and $\alpha_D$ is valid for $dD/dt(E_{D,0})$ and not for $dD/dt(D)$. In general, because $dD/dt(E_{D,0}(t_k)) \approx \beta_D - \alpha_D E_{D,0}$ tends to decrease, $\lambda_q > \lambda_r$. Since $dE_M/dt(E_M)$ is almost identical to $dE_{M+ie}/dt(E_M)$ and

differs only in the early stage of development, the approximations of $dE_q$ and $dE_r$ are only marginally affected (Figs. 8a and 9a). Therefore, information about the drift $D$ can be derived from these hypotheses also for the variant with initial and model error.



For the sole initial condition error, we found that $\lambda_q \leq \lambda_r$, but this describes a setting very different from model error: In the initial condition error, we compare the forecast and reality of a given high-resolution model, which indeed has much larger
error growth exponents for short times/small errors, due to small scale degrees of freedom. As soon as we talk about model errors (with or without initial error), we use the low-resolution L05-1 model for forecasts, and hence its parameters are relevant for the propagation of errors.

In summary, we propose a new interpretation of the growth of forecast errors due to model errors: model errors in successive time steps of the forecasts are only weakly correlated. Therefore, modeling them by a constant term in the error growth $dE$ is
inappropriate. The observed model forecast error growth can be modeled much more accurately if we use the accumulated model errors called drift, interpret the drift increments as additional initial condition errors, and propagate these forward in time. Then the decrease of the drift growth rate over forecast time can explain the growth rate of the model errors. Depending on what data are observed, one can either use the drift to predict the forecast errors, or use the forecast errors to infer the drift due to model error.

**5. Error growth in the ECMWF systems**

For the ECMWF forecasting system, we cannot perform error growth experiments, but we can check average forecast errors as a function of lead time. We, therefore, apply the new way of assessing the model error to the error growth $E_{EFS}(t)$ of the 500 hPa geopotential height values (Magnusson, 2013) calculated (Magnusson and Kallen, 2013) as 25 annual averages over the Northern Hemisphere (20–90∘) obtained daily from 1 January 1987 to 31 December 2011. Over this period, we determine
the decline in the average initial displacements of the "model" from "reality" per unit of time using the parameter $\beta_q$ of the quadratic hypothesis with model error. Since the parameter $\beta$ is also used to describe an upscale error growth rate from small-scale processes (Zhang et al., 2019), we check whether $\lambda_q > \lambda_r$, as defined in Section 5.2, for $\beta_q$ determined by model error and $\lambda_q \leq \lambda_r$ for $\beta_q$ determined from small-scale processes (see Section 4).

**5.1. Methods of calculation**

To eliminate the effects of model errors, the initial error growth curve $E_{EFS,ie}(t)$ is calculated as the differences between two operational forecasts issued with one day lag for the same day. Specifically, we evaluate these for 27 different lead times and used the following pairs of lead times in hours: 0–24, 6–30, ..., 96–120, with 6 hours shift, and from 108–132, 120–144, ..., 216–240 with 12 h shift. Detailed information about calculating the error growth of the ECMWF forecasting system can be found in Lorenz (1982). The error growth rate (tendency) is $dE_{EFS,ie} / dt \approx \left( E_{EFS,ie}(t + \Delta t) - E_{EFS,ie}(t) \right) / \Delta t$ with $\Delta t = 6$ hours
for the first seventeen time steps and $\Delta t = 12$ hours for the rest. It is evident that this data analysis solely used data produced by the very same model. One can understand this one-day offset between two forecasts in the following way: At day 0, we use





some initial condition and propagate it forward in time. At day 1, when a new forecast starts with new initial conditions, these can be interpreted as perturbations to the day-1 forecast started at day 0. So comparing now these two forecasts for the very same day as a function of lead time gives us the initial condition error growth. The only disadvantage of this procedure is that

we cannot control the magnitude of the perturbation: What we interpret as perturbation is the deviation of the true forecast at day one from the new analysis, which is used to initialize the new forecast.

The initial and model error growth curve $E_{EFS,M+ie}(t)$ is calculated as differences between operational forecasts and analyses from ERA-Interim for a given day. Forecasts range from 0.5 day ago relative to the given day to 10 days ago, with time step $\Delta t = 12$ hours. The difference between operational analysis and analysis from ERA-Interim is taken as the initial error. The

error growth rate (tendency) is $dE_{EFS,M+ie} / dt \approx \left( E_{EFS,M+ie}(t+\Delta t) - E_{EFS,M+ie}(t) \right) / \Delta t$ with $\Delta t = 12$ hours.

## 5.2. Results and comparisons

From the data, we calculate 25 annual averages of the initial error growth curve $E_{EFS,ie}(t)$ and 25 annual averages of the initial and model error growth curve $E_{EFS,M+ie}(t)$ and their growth rates (tendencies) $dE_{EFS,ie} / dt$ and $dE_{EFS,M+ie} / dt$. We approximate the growth rates by the exponential growth with model error $dE_r$ and by the quadratic hypothesis with model

error $dE_q$. Because the data are only up to 10 days and therefore do not cover the entire growth curve, and because $dE_q$ is a three-parameter approximation, we first discuss the error in parameter estimation. Magnusson and Kallen (2013) showed that the error saturation parameter $E_{\lim}$ estimated from the $dE_q$ hypothesis underestimates the true limiting value. Bednar et al. (2021) showed that the deviations of the values of $\lambda_q$ and $\beta_q$ of $dE_q$ from the true values are anti-correlated, i.e., when one is overestimated, then the other is underestimated. The average value of $\lambda_q$ over 25 annual averages of $E_{EFS,ie}(t)$ and

$E_{EFS,M+ie}(t)$ has been determined by Bednar et al. (2021) to be $\bar{\lambda}_q = 0.35 day^{-1}$, and to approximate the data, we fix $\lambda_q$ to this value. Therefore, we decrease the oscillation of $\beta_q$ and bring $E_{\lim}$ of $dE_q$ closer to the values determined by Magnusson and Kallen (2013). The fact that $E_{\lim}$ is closer to the theoretical limit values estimated by Magnusson and Kallen (2013) justifies this approach.

The resulting values are shown in Fig. 15. We find that $\lambda_{r,ie} \geq \lambda_{q,ie}$ and $\lambda_{r,M+ie} < \lambda_{q,M+ie}$, which satisfies the hypothesis presented in Section 4.2. Here, $\lambda_{r,M+ie}$ (Fig. 15a, full red curve) has an approximately constant value, showing approximately

the same decrease in $\alpha_D =$ the drift rate $dD / dt(D)$ over the years. Because it is shown in section 4.2 that

$\alpha_D = \left( \lambda_{q,M+ie} - \lambda_{r,M+ie} \right)$.



For $\lambda_{r,ie}$ (Fig. 15a, full blue curve), an increase with years is observed, indicating an increase in the resolution of the ECMWF system with smaller spatiotemporal scale phenomena with a larger error growth rate. The parameter $\beta_{q,M+ie}$ (Fig. 15b full blue curve) of the quadratic hypothesis with model error (Eq. (6)) shows an approximately linear decrease over the years. These values indicate that the average displacement value per unit of time $\beta \approx \overline{dD/dt}$ decreased by 40% from 1987 to 2011, from 5.6 m/day to 3.4 m/day. The parameter $\beta_{q,ie}$ (Fig. 15b, full red curve) for the variant with an initial error shows an approximately constant value. Together with a constant value of $\lambda_{q,ie}$ (Fig 15a, black curve), this means that the shape of the error growth rate (tendency) $dE_{ie}/dt(E_{ie})$ changes over the years only by adding a part for smaller $E_{ie}$ as the model better describes smaller spatiotemporal scale phenomena, but this does not change the overall approximation of $dE_q$, as can be seen in Fig. 16. These Figures also show the similarity to the error growth rates $dE/dt$ of the L05-2 and L05-3 systems (Figs. 6 - 9) and the relevance of fixing $\lambda_q$ to approximate the data using $dE_q$.

## 6.  Conclusion and discussion

Based on the fact that scale-dependent error growth implies an intrinsic predictability limit, we examined whether omitting atmospheric phenomena, which contribute little to the final value, will improve the predictability of the resulting value. In other words, how does the average forecast error growth change in a model where small-scale phenomena are omitted, but the model error is, therefore, larger, compared to a model where all phenomena are present, but the average forecast error growth is scale-dependent. For this, we used the L05 systems defined by Lorenz (2005) and Bednar and Kantz (2022) and the ECMWF systems with data from Magnusson (2013).

We confirmed that for the multi-scale systems L05-2 and L05-3, the initial error growth $E_{ie}(t)$ can be well described by the power law $dE_p$ Eq. (3) or the extended power law $dE_w$ Eq. (4), respectively, while a simple exponential growth with model error $dE_r$ (Eq. (5)) or the quadratic hypothesis with model error $dE_q$ (Eq. (6)) are less appropriate. However, the non-zero parameter $\beta$ in $dE_r$ and $dE_q$ describing the model error also generally relates the multi-scale nature of the system. We showed that in the L05 and ECMWF data (in contrast to the model error scenario) $\lambda_q \le \lambda_r$, i.e., the approximation of $dE_{ie}/dt(E_{ie})$ in the early stage grows faster than the approximation of the whole curve due to the presence of only smaller spatiotemporal scales in this part.

For the scenarios of model error growth $E_M(t)$ and both initial and model error growth $E_{M+ie}(t)$, we showed the appropriateness of the description using exponential growth with model error $dE_r$ and quadratic hypothesis with model error $dE_q$ (Figs. 8 and 5). For $dE_M/dt(E_M)$, we explained the initial decline and subsequent growth described by $dE_r$ using the



drift $D$ (Fig. 11) defined by Orrell (2001), which we extended by a hypothesis that views the drift $D$ as a succession of initial

errors followed by an exponential time evolution driven by the largest Lyapunov exponent $\lambda$ of the model after a transition

period (Fig. 12). We identified $\lambda \approx \lambda_q$, and the validity of $\lambda_q > \lambda_r$ based on the drift evolution was verified in the L05 and

ECMWF systems (Figs. 14 and Eq. (21)). For the L05 systems, we have demonstrated that $\beta_q \approx \overline{dD/dt}$, i.e., that from the $\beta_q$

value of the quadratic hypothesis with model error $dE_q$ the average displacement per unit time (average drift rate) can be

determined.

For ECMWF systems (forecast of 500 hPa geopotential height), this means that from 1987 to 2011, we observe a decrease of

the model error by approximately 40%, from an average displacement value of 5.6 m/day to 3.4 m/day. Note that while in

1987, the error in initial displacement (initial error) was approximately 16 m (Magnusson and Kallen (2013), Bednar (2021)),

for the variant with initial and model error, a displacement of 5.6 m is produced every day in addition to this value. In 2011,

the initial displacement was 6 m; for the variant with initial and model error, an average displacement of 3.4 m is produced

daily. Thus, we observe a significant contribution of the model error to the error growth. This is also why the findings from

the model error growth scenario can be applied to the initial and model error growth scenario, where the initial and model error

growth goes asymptotically to the model error growth.

It is also why omitting atmospheric phenomena, which contribute little to the final value, will not improve the predictability

of the resulting value. The average prediction error grows faster in a model where small-scale phenomena are omitted, but the

model error is therefore created, compared to a model where all phenomena are present, but the average forecast error growth

is scale-dependent (Fig. 10).

We now discuss the possibility that the growth of the displacement produced by the model error may also be scale-dependent.

In our case, the model was an L05-1 system, i.e., a system with one scale and exponential error growth ($E_D(t)$ hypothesis). A

variant where the L05-2 system was used as the model and the L05-3 system as the "reality" was also tested. The resulting

model error growth is approximately identical to the previous variant (L05-1 system as the model and L05-3 system as the

"reality"), i.e., adding a small scale did not affect the exponential growth of the drift $D$ increment. However, it should be

noted that the magnitude of the average drift per unit of time $\beta_q \approx \overline{dD/dt}$ is much greater than the limit (saturation) value of

small-scale error magnitude $E_{2,lim}$, so we are already in the region of exponential error growth of large-scale variables. For the

ECMWF system, it can be seen (Fig. 15) that over the years, the values of a parameter $\beta_q$ of the $dE_q$ approximation of average

initial error growth $E_{ie}(t)$ and initial and model error growth $E_{ie+M}(t)$ converge, and the growth curves of the two variants

are similar for the later analyzed years, as also confirmed by Froude et al. (2013). This means that, in contrast to the presented

results of the L05 system, the drift rate of these years is low, and the issue of scale-dependent growth of the drift $D$ increment

is relevant and should be further investigated.

We now discuss the possibility that the growth of the displacement produced by the model error may also be scale-dependent. In our case, the model was an L05-1 system, i.e., a system with one scale and exponential error growth ( $E_D(t)$ hypothesis). A variant where the L05-2 system was used as the model and the L05-3 system as the "reality" was also tested. The resulting model error growth is approximately identical to the previous variant (L05-1 system as the model and L05-3 system as the "reality"), i.e., adding a small scale did not affect the exponential growth of the drift $D$ increment. However, it should be

noted that the magnitude of the average drift per unit of time $\beta_q \approx \overline{dD/dt}$ is much greater than the limit (saturation) value of small-scale error magnitude $E_{2,\lim}$, so we are already in the region of exponential error growth of large-scale variables. For the ECMWF system, it can be seen (Fig. 15) that over the years, the values of a parameter $\beta_q$ of the $dE_q$ approximation of average initial error growth $E_{ie}(t)$ and initial and model error growth $E_{M+ie}(t)$ converge, and the growth curves of the two variants are similar for the later analyzed years, as also confirmed by Froude et al. (2013). This means that, in contrast to the presented

results of the L05 system, the drift rate of these years is low, and the issue of scale-dependent growth of the drift $D$ increment is relevant and should be further investigated.

Another topic for further research is extending the $E_D(t)$ hypothesis (Eq. (34)) to describe the model error growth $E_M(t)$ over the entire range up to saturation rather than just the early part where exponential growth is valid. For the part of the time evolution of the model error where the growth slows down and reaches saturation, it can be seen that the drift must reach its

limiting value and then no longer contributes to the model error growth and the exponential growth of the drift increment must slow down and reach its limiting value. However, the specific form needs to be investigated.

**Code and data availability**

The ECMWF forecasting system dataset was obtained from the personal repository of Linus Magnusson (Magnusson, 2013). L05-3 system dataset, products from the ECMWF forecasting system dataset, codes, and figures were conducted in Wolfram

Mathematica, and they are permanently stored at http://www.doi.org/10.17605/OSF.IO/2EWXB (Bednář, 2023).

**Author contributions**

H.B. proposed the idea, carried out the experiments, and wrote the paper. H.K. supervised the study and co-authored the paper.

**Competing interests**

The authors declare that they have no conflict of interest.



**Acknowledgements**

The authors are grateful to Linus Magnusson for offering Dataset (ECMWF forecasting system) from his personal repository.

**Financial support**

This work was funded by project Cooperatio "Sci-Physics", programme of the Charles University and by the Czech Science Foundation, through grant 19-16066S.

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





**Appendix A: Lorenz's L05 systems**

The L05-1 system was introduced by Lorenz (2005) as a spatial continuity modification of Lorenz's (1996) system with $N$ variables connected by governing equations:

$$\frac{dX_n}{dt} = [X,X]_{L,n} - X_n + F, \tag{A1}$$

where

$$[X,X]_{L,n} = \sum_{j=-J}^{J}{}' \sum_{i=-J}^{J}{}' \left( -X_{n-2L-i} X_{n-L-j} + X_{n-L+j-i} X_{n+L+j} \right) / L^2,$$

$n = 1,\ldots,N$. $X_n$ are unspecified (i.e., unrelated to actual physical variables), scalar meteorological quantities (units), $F$ is a constant representing external forcing and $t$ is time. The index is cyclic so that $X_{n-N} = X_{n+N} = X_n$ and variables can be viewed as existing around a latitude circle. If $L$ is even, $\sum{}'$ denotes a modified summation in which the first and last terms are to be divided by 2. If $L$ is odd, $\sum{}'$ denotes an ordinary summation. Generally, $L$ is much smaller than $N$ and $J = L/2$ if $K$ is even and $J = (L-1)/2$ if $L$ is odd. To keep 5 to 7 main highs and lows that correspond to planetary waves (Rossby

waves), Lorenz (2005) suggested a ratio $N/L = 30$ and $F = 15$. The choice of parameters $F$, and the setting of time unit = 5 days, is also made to obtain a similar value of the largest Lyapunov exponent as the ECMWF forecasting system (Lorenz, 2005).

The L05-2 system (extension to two Spatio-temporal scales) was also introduced by Lorenz (2005) as a modification of two scales Lorenz's (1996) system, where scales were coupled by linear terms that together do not alter the large-scale plus small-

scale energy and where small-scale variables were driven entirely by the coupling. Rewriting the equations of the L05-1 system, we would get:

$$dX_{1,n}/dt = [X_1,X_1]_{L,n} - X_{1,n} - cX_{2,n} + F, \tag{A2}$$

$$dX_{2,n}/dt = b^2 [X_2,X_2]_{1,n} - bX_{2,n} + cX_{1,n}, \tag{A3}$$

$c$ sets the rapidness of small scale compared to large scale, and $b$ sets the small-scale amplitude size compared to large scale.

Eqs. (25) and (26) have an unrealistic property compared to the numerical weather prediction systems. The large-scale and small-scale features are represented by separate sets of variables $X_1$ and $X_2$ instead of appearing as superimposed features of a single set $X_{tot}$. Lorenz (2005) wanted to keep the system as simple as possible, so instead of, for example, Fourier analysis, a procedure for expressing variables $X_{tot,n}$ as sums of $X_{1,n}$ and $X_{2,n}$ was introduced:



$$X_{1,n} = \sum_{i=-I}^{I}{}'\left(\alpha - \omega|i|\right) X_{tot,n+i}, \tag{A4}$$


$$X_{2,n} = X_{tot,n} - X_{1,n}. \tag{A5}$$

Parameters $\alpha$, $\omega$, and $I$ are chosen so that $X_1$ is a low-pass filtered version of $X_{tot}$, and $X_2$ represents the difference between the full signal $X_{tot}$ and the filtered signal. By this procedure, $X_2$ has a much smaller amplitude than $X_1$, and also its time evolution should be faster since the temporal derivative is related to the spatial derivative via the difference $(X_{1,n+1} - X_{1,n-2})$, which for the low pass filtered signal $X_1$ typically is smaller than for the signal $X_2$.

More precisely, Lorenz's (2005) idea is that the parameters $\alpha$, $\omega$ are chosen so that $X_1$ equals $X_{tot}$ whenever $X_{tot}$ changes quadratically over the longitudes (variables) $n - I$ through $n + I$. It is when $\sum_{i=-I}^{I}{}'\left(\alpha - \omega|i|\right) = 1$ and $\sum_{i=-I}^{I}{}'i^2\left(\alpha - \beta|i|\right) = 0$. By solving these equations, we get:

$$\alpha = \left(3I^2 + 3\right) / \left(2I^3 + 4I\right), \tag{A6}$$

$$\omega = \left(2I^2 + 1\right) / \left(I^4 + 2I^2\right). \tag{A7}$$

The procedures (Eqs. (A4) and (A5)) are functions of the interval length $[-I, I]$.

When creating a system $dX_{tot} / dt$ as the sum of $dX_1 / dt$ and $dX_2 / dt$ (sum of Eqs. (25) and (26)), the coupling term $cX_{1,n}$ in Eq. (A3), which enables short waves to develop, is combined with the dissipation term $-X_{1,n}$ in Eq. (A2). Therefore, the coupling term can be canceled entirely, or it can appear in $X_1$ rather than $X_2$ when $X_{tot}$ is analyzed, and there might be nothing to enable the short waves in $X_2$ to grow. Lorenz (2005) reformulated the coupling process by adding a small fraction

of $X_1$ to $X_2$ so small waves in $X_2$ can amplify. It is done by replacing $b^2\left[X_2, X_2\right]_{1,n} + cX_{1,n}$ by $\left[X_2, X_2 + c'X_1\right]_{1,n}$ in Eq. (A3), and L05-2 system would be:

$$dX_{tot,n} / dt = \left[X_1, X_1\right]_{L,n} + b^2\left[X_2, X_2\right]_{1,n} + c\left[X_2, X_1\right]_{1,n} - X_{1,n} - bX_{2,n} + F, \tag{A8}$$

where $c = c' \cdot b^2$.

Based on the L05-2 system (Eqs. (A4) - (A8)), Bednar and Kantz (2022) designed a three levels (scales) system (L05-3):

$dX_{tot,n} / dt = \left[X_1, X_1\right]_{L,n} + b_1^2\left[X_2, X_2\right]_{1,n} + b_2^2\left[X_3, X_3\right]_{1,n} + c_1\left[X_2, X_1\right]_{1,n} + c_2\left[X_3, X_2\right]_{1,n} - X_{1,n} - b_1 X_{2,n} - b_2 X_{3,n} + F,$ (A9)





where $c_1$, $c_2$, $b_1$, $b_2$ are parameters, and the procedure for expressing the variables are:

$$X_{1,n} = \sum_{i=-I_1}^{I_1}{}' \left( \left( \left( 3I_1^2 + 3 \right) / \left( 2I_1^3 + 4I_1 \right) \right) - \left( \left( 2I_1^2 + 1 \right) / \left( I_1^4 + 2I_1^2 \right) \right) |i| \right) X_{tot,n+i}, \tag{A10}$$

$$X_{2,n} = \sum_{j=-I_2}^{I_2}{}' \left( \left( \left( 3I_2^2 + 3 \right) / \left( 2I_2^3 + 4I_2 \right) \right) - \left( \left( 2I_2^2 + 1 \right) / \left( I_2^4 + 2I_2^2 \right) \right) |j| \right) \left( X_{tot,n+j} - X_{1,n+j} \right), \tag{A11}$$

$$X_{3,n} = X_{tot,n} - X_{2,n} - X_{1,n}, \tag{A12}$$

where $I_1$ and $I_2$ set the length of the intervals $[-I, I]$.

The parameters of L05 systems (L05-1, L05-2, L05-3) should be set so that all scales behave chaotically (the largest Lyapunov exponent of each scale is positive) and that all scales have a significant difference in amplitudes and fluctuation rates. For the L05-1 system (Eq. (A1)), the chaotic behavior is determined by the value of $F$ and the number of variables $N$. For Eqs. (A2) and (A3), where the forcing $F$ acts only on a large scale, the chaotic behavior of small scale is created by coupling. The

coupling size is cascaded from a large scale to a small one. Because the values of large-scale variables are determined by the forcing $F$, the $F$ value indirectly affects the small-scale chaotic behavior and must be chosen large enough to ensure chaotic behavior through coupling for all scales (levels). This fact must also apply to L05-2 and L05-3 systems, but procedures (A4) and (A5) for the L05-2 system and (A10) - (A12) for L05-3 system also affect the scales' chaotic behavior, amplitude, and fluctuation rate through the choice of $I$ (Lorenz, 2005).

To maintain the required properties $F = 15$, $N = 360$, $L = 12$, and $J = 6$ is chosen for the L05-1 system (Fig. 1a). To have the small scale one hundred times smaller than the large scale, $F = 15$, $N = 360$, $L = 12$, $J = 6$, $b = 10$, $c = 1$, $I = 10$ are selected for the L05-2 system (Fig. 2a). For the L05-3 system with requirements for the medium scale amplitude to be about ten times smaller than the large scale amplitude and the small scale amplitude to be about ten times smaller than the medium scale amplitude and for the scales to have different oscillation rates (Fig. 3a), $F = 15$, $N = 360$, $L = 12$, $J = 6$, $b_1 = 1$,

$b_2 = 10$, $c_1 = 1$, $c_2 = 1$, $I_1 = 20$, $I_2 = 10$. The calculation is done using a fourth-order Runge-Kutta method with a time step $\Delta t = 1/240$ or 0.5 hours.






|            | L05-2 | L05-3 |
| ---------- | ----- | ----- |
| $\lambda_r$ | 0.17  | 0.25  |
| $\beta_r$   | 0.33  | 1.15  |
| $\lambda_q$ | 0.27  | 0.38  |
| $\beta_q$   | 0.34  | 1.47  |
| $E_{lim}$   | 7.6   | 7.8   |
| $\overline{dD/dt}$ | 0.33 | 1.6 |

**Table 1: Table of fitted constants of the different error growth approximations for the L05-2 model and the L05-3 model. All values except $E_{lim}$ are given as units/day or 1/day.**



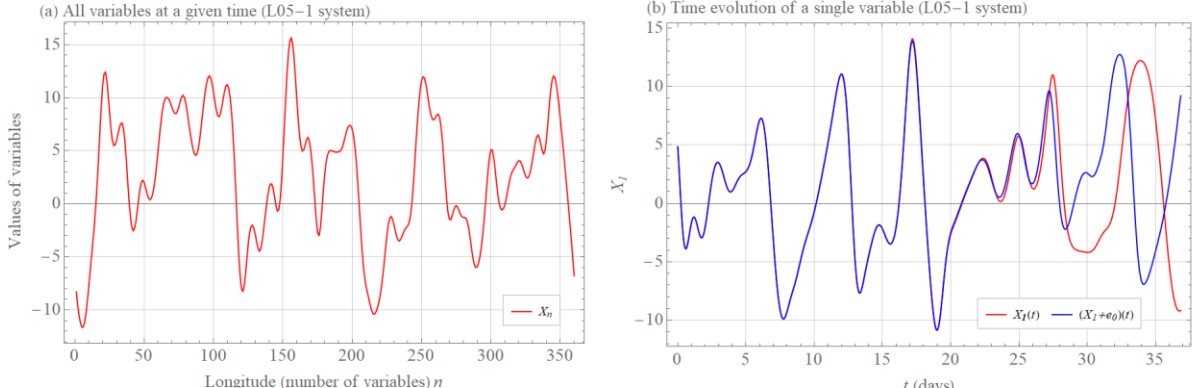

**Figure 1. (a) Values of** $X_n$ **,** $n = 1,\ldots,360$ **variables (red curve) of the L05-1 system at a given time (see Appendix A for more**
**information on the system and its settings). (b) The time evolution** $t$ **of the variable** $X_1(t)$ **(red curve) and the time evolution of the**
**initially nearby trajectory** $\left(X_1(0) + e_0\right)(t)$ **(blue curve) of the system L05-1, where** $e_0 = 0.01$ **.**





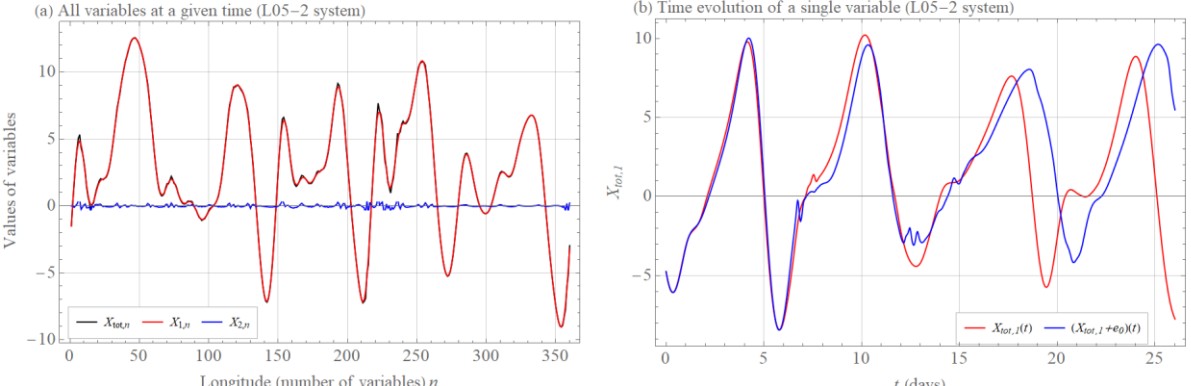

**Figure 2. (a) Values of** $X_{1,n}$ **(large scale, red curve),** $X_{2,n}$ **(small scale, blue curve),** $X_{tot,n}$ **(overall curve, black curve),** $n = 1, \ldots, 360$
**variables of the L05-2 system at a given time (see Appendix A for more information on the system and its settings). (b) The time**
**evolution** $t$ **of the variable** $X_{tot,1}(t)$ **(red curve) and the time evolution of the initially nearby trajectory** $\left(X_{tot,1}(0) + e_0\right)(t)$ **(blue**
**curve) of the system L05-2, where** $e_0 = 0.01$ **.**



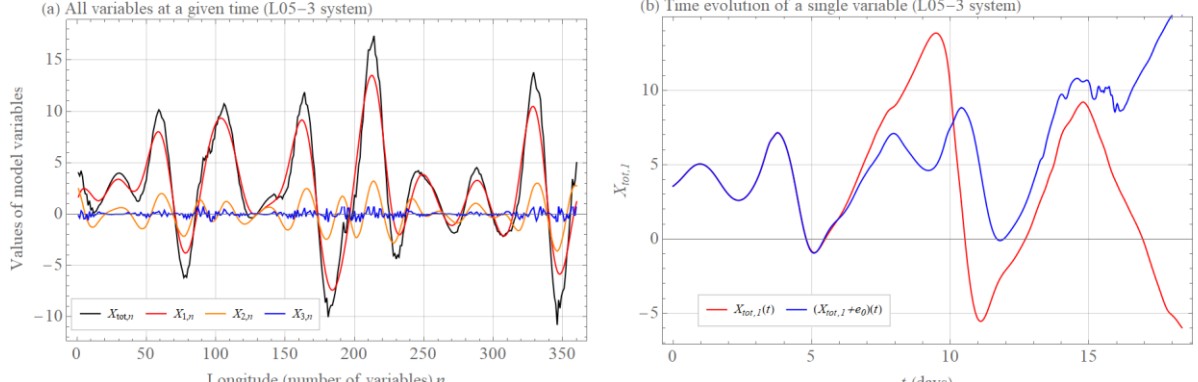

**Figure 3. (a) Values of** $X_{1,n}$ **(large scale, red curve),** $X_{2,n}$ **(medium scale, yellow curve),** $X_{3,n}$ **(small scale, blue curve),** $X_{tot,n}$ **(overall curve, black curve),** $n = 1,\ldots,360$ **variables of the L05-3 system at a given time (see Appendix A for more information on the system and its settings). (b) The time evolution** $t$ **of the variable** $X_{tot,1}(t)$ **(red curve) and the time evolution of the initially nearby trajectory** $(X_{tot,1}(0) + e_0)(t)$ **(blue curve) of the system L05-2, where** $e_0 = 0.01$ **.**



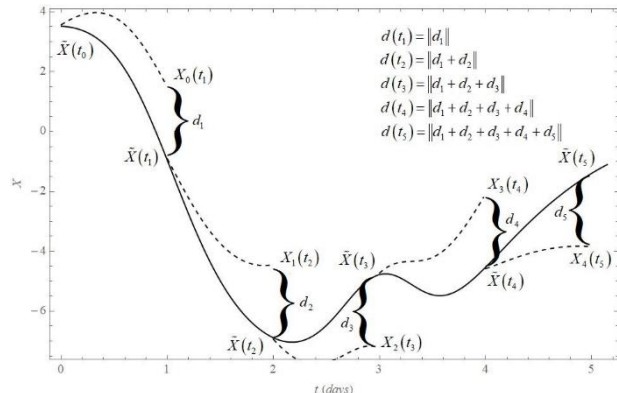

**Figure 4. Schematic description of drift $d$ calculation. The solid curve shows the time evolution of the "reality" projected into the "model" space $\tilde{X}(t_j)$ at the time $t_j = j \cdot \Delta t$ for $j = 1, \ldots, 5$. The dashed curves show the short ($\Delta t$) time evolutions of the "model"**

**$X_j(t)$ for $t \geq t_j$ initiated at the points $\tilde{X}(t_j)$. Drift $d(t)$ is then the sum of the difference $d(t_{j+1}) = X_j(t_{j+1}) - \tilde{X}(t_{j+1})$ at each time step.**



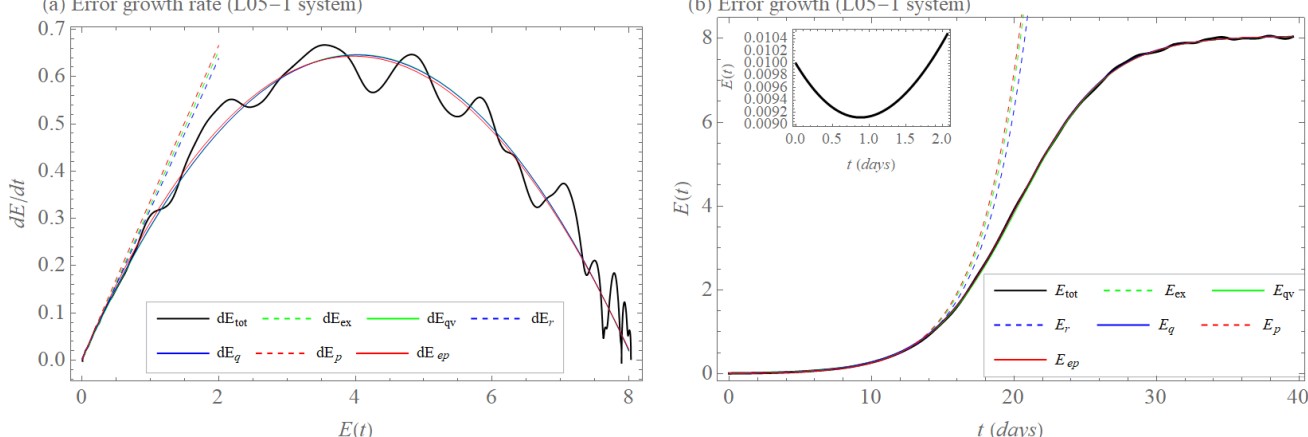

**Figure 5. (a) Initial error growth tendency (rate)** $dE/dt$ **as a function of the error magnitude** $E$ **(** $dE_{tot}$ **, black), approximation of**
**the early part of the growth by exponential growth** $dE_{ex}$ **(Eq. (1), green, dashed), exponential growth with model error** $dE_r$ **(Eq. (5),**
**blue, dashed), power law** $dE_p$ **(Eq. (3), red, dashed) and approximation of the full curve by quadratic hypothesis** $dE_{qv}$ **(Eq. (2),**
**green), quadratic hypothesis with model error** $dE_q$ **(Eq. (6), blue) and extended power law** $dE_{ep}$ **(Eq. (4), red) for the L05-1 system.**
**(b) Initial error growth** $E$ **as a function of time** $t$ **(** $E_{tot}$ **, Eq. (11), black), approximation of the early part of the growth by integration**
**of** $dE_{ex}$ **(** $E_{ex}$ **, green, dashed) with** $\lambda_{ex} = 0.33$ **1/day, integration of** $dE_r$ **(** $E_r$ **, blue, dashed) with** $\lambda_r = 0.32$ **1/day and** $\beta_r = 0.00006$
**unit/day, integrations of** $dE_p$ **(** $E_p$ **, red, dashed) with** $a = 0.34$ **unit$^{0.02}$/day and** $b = 0.02$ **and approximation of the full curve by**
**integration of** $dE_{qv}$ **(** $E_{qv}$ **, green) with** $\lambda_{qv} = 0.32$ **1/day and** $E_{lim} = 8.1$ **unit, integration of** $dE_q$ **(** $E_q$ **, blue) with** $\lambda_q = 0.32$ **1/day,**
$\beta_q = 0.003$ **unit/day and** $E_{lim} = 8.1$ **unit and integration of** $dE_{ep}$ **(** $E_{ep}$ **, red) with** $a = 0.33$ **unit$^{0.03}$/day,** $b = 0.03$ **and** $E_{lim} = 8.1$ **unit for**
**the L05-1 system. The inset shows transient behavior before the error magnitude grows (for more details, see Section 2.1).**



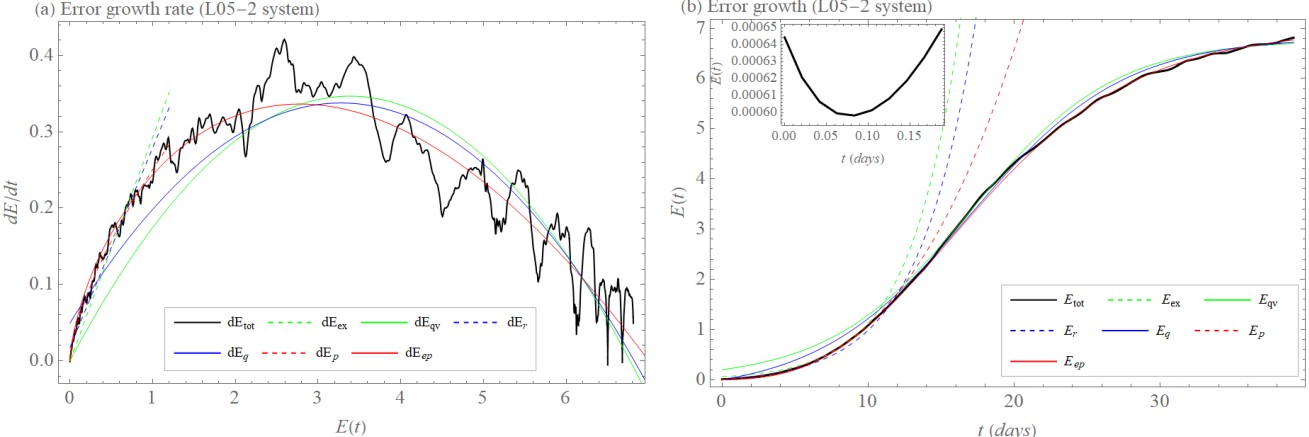

**Figure 6. (a) Initial error growth tendency (rate)** $dE/dt$ **as a function of the error magnitude** $E$ **(** $dE_{tot}$ **, black), approximation of the early part of the growth by exponential growth** $dE_{ex}$ **(Eq. (1), green, dashed), exponential growth with model error** $dE_r$ **(Eq. (5), blue, dashed), power law** $dE_p$ **(Eq. (3), red, dashed) and approximation of the full curve by quadratic hypothesis** $dE_{qv}$ **(Eq. (2), green), quadratic hypothesis with model error** $dE_q$ **(Eq. (6), blue) and extended power law** $dE_{ep}$ **(Eq. (4), red) for the L05-2 system. (b) Initial error growth** $E$ **as a function of time** $t$ **(** $E_{tot}$ **, Eq. (11), black), approximation of the early part of the growth by integration of** $dE_{ex}$ **(** $E_{ex}$ **, green, dashed) with** $\lambda_{ex} = 0.29$ **1/day, integration of** $dE_r$ **(** $E_r$ **, blue, dashed) with** $\lambda_r = 0.26$ **1/day and** $\beta_r = 0.02$ **unit/day, integrations of** $dE_p$ **(** $E_p$ **, red, dashed) with** $a = 0.25$ **unit$^{0.32}$/day and** $b = 0.32$ **and approximation of the full curve by integration of** $dE_{qv}$ **(** $E_{qv}$ **, green) with** $\lambda_{qv} = 0.2$ **1/day and** $E_{lim} = 6.8$ **unit, integration of** $dE_q$ **(** $E_q$ **, blue) with** $\lambda_q = 0.18$ **1/day,** $\beta_q = 0.05$ **unit/day and** $E_{lim} = 6.8$ **unit and integration of** $dE_{ep}$ **(** $E_{ep}$ **, red) with** $a = 0.28$ **unit$^{0.34}$/day,** $b = 0.34$ **and** $E_{lim} = 7$ **unit for the L05-2 system. The inset shows transient behavior before the error magnitude grows (for more details, see Section 2.1).**



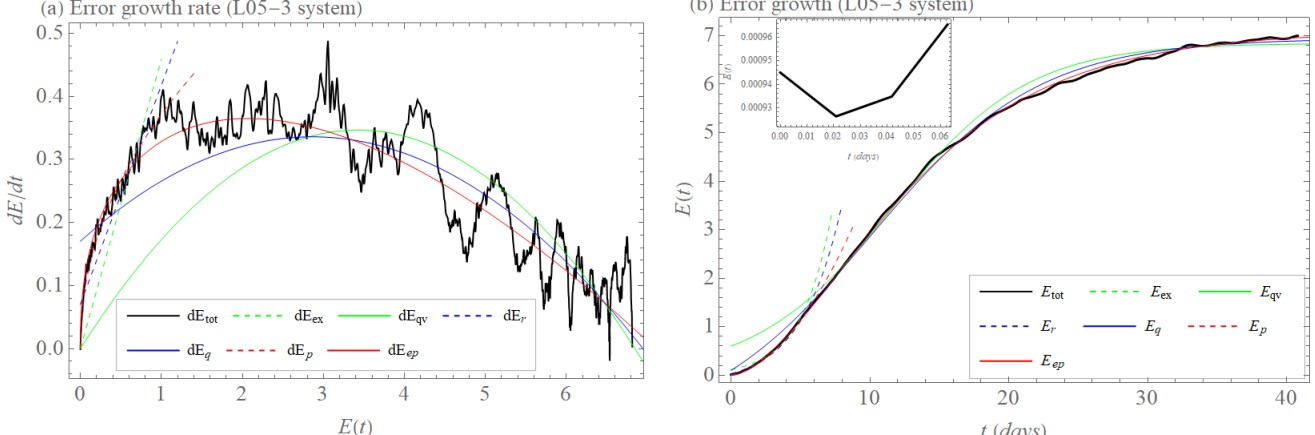

**Figure 7. (a) Initial error growth tendency (rate)** $dE/dt$ **as a function of the error magnitude** $E$ **(** $dE_{tot}$ **, black), approximation of the early part of the growth by exponential growth** $dE_{ex}$ **(Eq. (1), green, dashed), exponential growth with model error** $dE_r$ **(Eq. (5), blue, dashed), power law** $dE_p$ **(Eq. (3), red, dashed) and approximation of the full curve by quadratic hypothesis** $dE_{qv}$ **(Eq. (2), green), quadratic hypothesis with model error** $dE_q$ **(Eq. (6), blue) and extended power law** $dE_{ep}$ **(Eq. (4), red) for the L05-3 system. (b) Initial error growth** $E$ **as a function of time** $t$ **(** $E_{tot}$ **, Eq. (11), black), approximation of the early part of the growth by integration of** $dE_{ex}$ **(** $E_{ex}$ **, green, dashed) with** $\lambda_{ex} = 0.46$ **1/day, integration of** $dE_r$ **(** $E_r$ **, blue, dashed) with** $\lambda_r = 0.35$ **1/day and** $\beta_r = 0.07$ **unit/day, integrations of** $dE_p$ **(** $E_p$ **, red, dashed) with** $a = 0.37$ **unit$^{0.63}$/day and** $b = 0.63$ **and approximation of the full curve by integration of** $dE_{qv}$ **(** $E_{qv}$ **, green) with** $\lambda_{qv} = 0.2$ **1/day and** $E_{lim} = 6.9$ **unit, integration of** $dE_q$ **(** $E_q$ **, blue) with** $\lambda_q = 0.14$ **1/day,** $\beta_q = 0.17$ **unit/day and** $E_{lim} = 6.9$ **unit and integration of** $dE_{ep}$ **(** $E_{ep}$ **, red) with** $a = 0.38$ **unit$^{0.59}$/day,** $b = 0.59$ **and** $E_{lim} = 7.1$ **unit for the L05-3 system. The inset shows transient behavior before the error magnitude grows (for more details, see Section 2.1).**





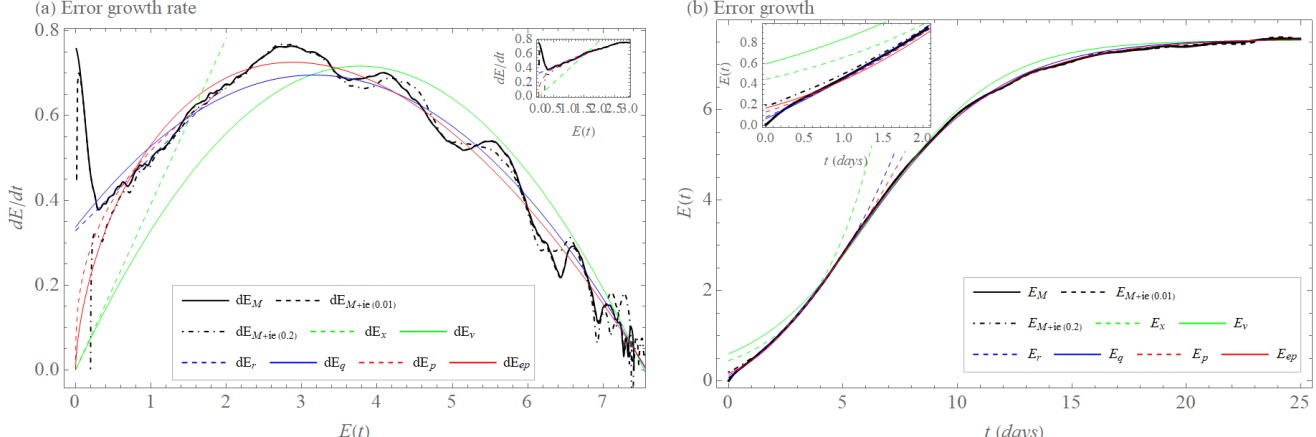

**Figure 8. (a) Model error growth tendency (rate)** $dE/dt$ **as a function of the error magnitude** $E$ **(** $dE_M$ **, black), initial and model error growth tendency** $dE/dt$ **as a function of the error magnitude** $E$ **(** $dE_{M+ie(0.01)}$ **, black, dashed for** $E(0)=0.01$ **and** $dE_{M+ie(0.2)}$ **, black, dot-dashed for** $E(0)=0.2$ **), approximation of the early part of the model growth by exponential growth** $dE_{ex}$ **(Eq. (1), green, dashed), exponential growth with model error** $dE_r$ **(Eq. (5), blue, dashed), power law** $dE_p$ **(Eq. (3), red, dashed) and approximation of the full curve by quadratic hypothesis** $dE_{qv}$ **(Eq. (2), green), quadratic hypothesis with model error** $dE_q$ **(Eq. (6), blue) and extended power law** $dE_{ep}$ **(Eq. (4), red) for the L05-2 system as the "reality" and the L05-1 system as the "model." The inset shows the early phase. (b) Model error growth** $E$ **as a function of time** $t$ **(** $E_M$ **, Eq. (14), black), initial and model error growth** $E$ **as a function of time** $t$ **(** $E_{M+ie(0.01)}$ **, Eq. (15), black, dashed for** $E(0)=0.01$ **and** $E_{M+ie(0.2)}$ **, Eq. (15), black, dot-dashed for** $E(0)=0.2$ **), approximation of the early part of the growth by integration of** $dE_{ex}$ **(** $E_{ex}$ **, green, dashed) with** $\lambda_{ex}=0.39$ **1/day, integration of** $dE_r$ **(** $E_r$ **, blue, dashed) with** $\lambda_r=0.17$ **1/day and** $\beta_r=0.33$ **unit/day, integrations of** $dE_p$ **(** $E_p$ **, red, dashed) with** $a=0.52$ **unit$^{0.64}$/day and** $b=0.64$ **and approximation of the full curve by integration of** $dE_{qv}$ **(** $E_{qv}$ **, green) with** $\lambda_{qv}=0.38$ **1/day and** $E_{lim}=7.5$ **unit, integration of** $dE_q$ **(** $E_q$ **, blue) with** $\lambda_q=0.27$ **1/day,** $\beta_q=0.34$ **unit/day and** $E_{lim}=7.6$ **unit and integration of** $dE_{ep}$ **(** $E_{ep}$ **, red) with** $a=0.61$ **unit$^{0.39}$/day,** $b=0.39$ **and** $E_{lim}=7.6$ **unit for the L05-2 system as the "reality" and the L05-1 system as the "model." The inset shows the early phase of the time evolution.**



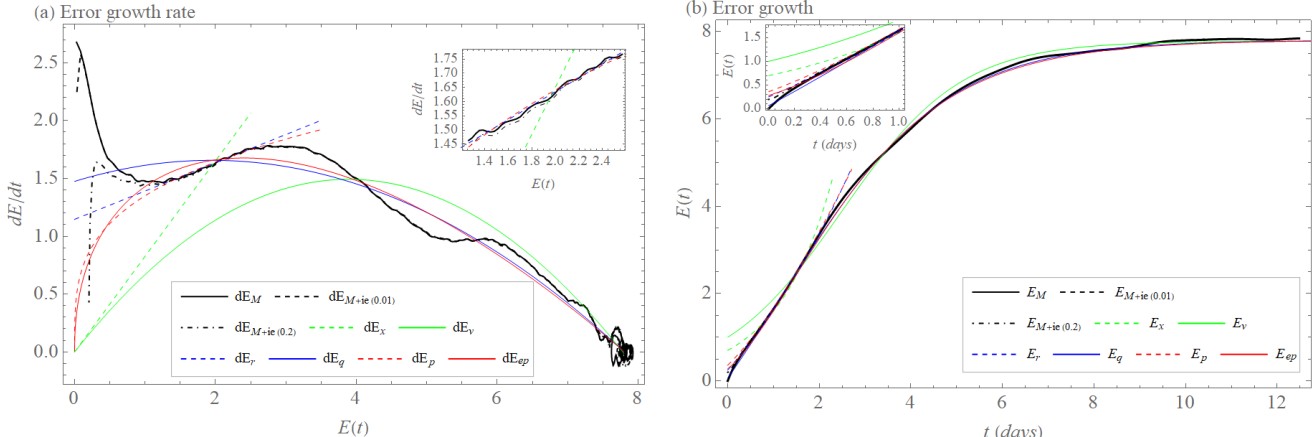

**Figure 9. (a) Model error growth tendency (rate)** $dE/dt$ **as a function of the error magnitude** $E$ **(** $dE_M$ **, black), initial and model error growth tendency** $dE/dt$ **as a function of the error magnitude** $E$ **(** $dE_{M+ie(0.01)}$ **, black, dashed for** $E(0)=0.01$ **and** $dE_{M+ie(0.2)}$ **, black, dot-dashed for** $E(0)=0.2$ **), approximation of the early part of the model growth by exponential growth** $dE_{ex}$ **(Eq. (1), green, dashed), exponential growth with model error** $dE_r$ **(Eq. (5), blue, dashed), power law** $dE_p$ **(Eq. (3), red, dashed) and approximation of the full curve by quadratic hypothesis** $dE_{qv}$ **(Eq. (2), green), quadratic hypothesis with model error** $dE_q$ **(Eq. (6), blue) and extended power law** $dE_{ep}$ **(Eq. (4), red) for the L05-3 system as the "reality" and the L05-1 system as the "model." The inset shows the early phase. (b) Model error growth** $E$ **as a function of time** $t$ **(** $E_M$ **, Eq. (14), black), initial and model error growth** $E$ **as a function of time** $t$ **(** $E_{M+ie(0.01)}$ **, Eq. (15), black, dashed for** $E(0)=0.01$ **and** $E_{M+ie(0.2)}$ **, Eq. (15), black, dot-dashed for** $E(0)=0.2$ **), approximation of the early part of the growth by integration of** $dE_{ex}$ **(** $E_{ex}$ **, green, dashed) with** $\lambda_{ex}=0.83$ **1/day, integration of** $dE_r$ **(** $E_r$ **, blue, dashed) with** $\lambda_r=0.25$ **1/day and** $\beta_r=1.15$ **unit/day, integrations of** $dE_p$ **(** $E_p$ **, red, dashed) with** $a=1.35$ **unit$^{0.72}$/day and** $b=0.72$ **and approximation of the full curve by integration of** $dE_{qv}$ **(** $E_{qv}$ **, green) with** $\lambda_{qv}=0.77$ **1/day and** $E_{\lim}=7.8$ **unit, integration of** $dE_q$ **(** $E_q$ **, blue) with** $\lambda_q=0.38$ **1/day,** $\beta_q=1.47$ **unit/day and** $E_{\lim}=7.8$ **unit and integration of** $dE_{ep}$ **(** $E_{ep}$ **, red) with** $a=1.64$ **unit$^{0.55}$/day,** $b=0.55$ **and** $E_{\lim}=7.8$ **unit for the L05-3 system as the "reality" and the L05-1 system as the "model." The inset shows the early phase of the time evolution.**



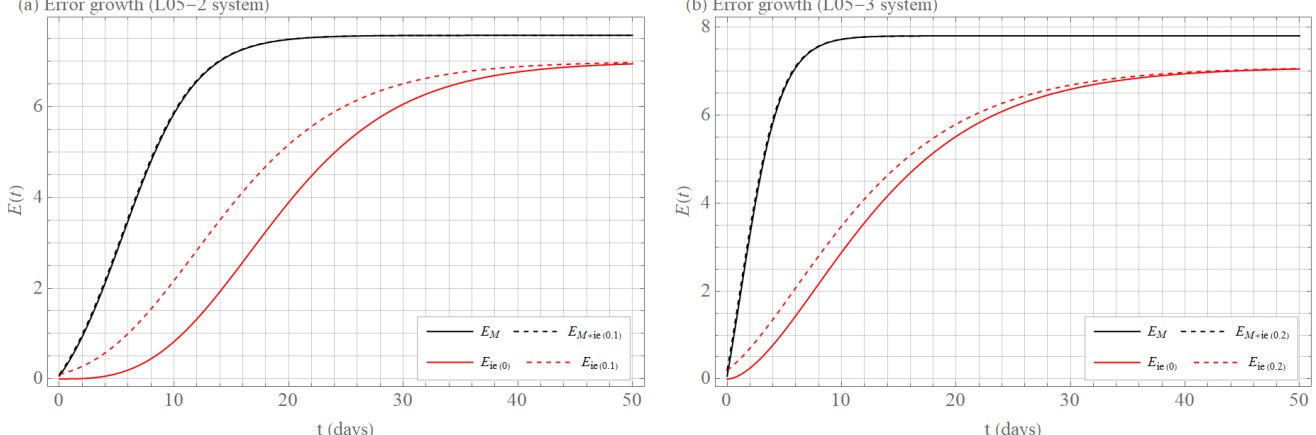

**Figure 10.** Error growth $E$ as a function of time $t$. The full black curve shows model error growth $E_M$ (Eq. (14)), the dashed black curve shows initial and model error growth $E_{M+ie}$ (Eq. (15)), the full red curve displays initial error growth $E_{ie}$ for $E(0) \to 0$ (Eq. (11)), and the dashed red curve displays initial error growth $E_{ie}$ for (a) $E(0) \approx 0.1$ and (b) $E(0) \approx 0.2$ (Eq. (11)). Shown are calculations of the best-fit approximations for given types of error growth (see Section 3 for more details). The initial error growth $E_{ie}$ is calculated for (a) the L05-2 system and (b) the L05-3 system. The model error growth $E_M$ and initial + model error growth $E_{M+ie}$ is calculated as (a) the difference between the L05-1 system and the L05-2 system and (b) between the L05-1 system and the L05-3 system.





870

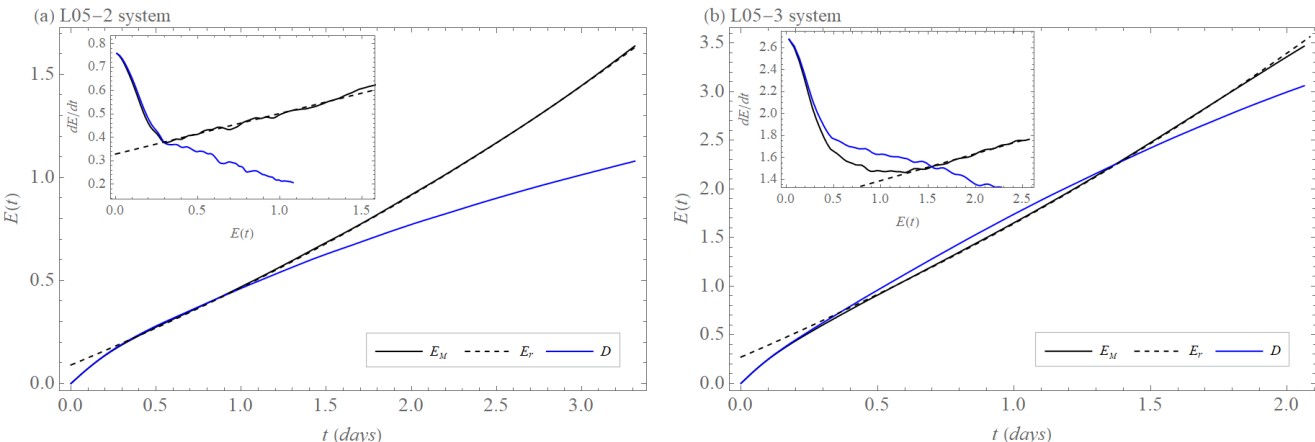

**Figure 11. Time evolution of drift** $D$ **(blue), model error** $E_M$ **(black), and approximation by exponential growth with model error** $E_r$ **(black, dashed). The inset shows the tendency (rate) of drift** $dD/dt$ **(blue), model error** $dE_r/dt$ **(black), and** $dE_r$ **(black, dashed) as a function of** $D(t)$ **,** $E_M(t)$ **, and** $E_r(t)$ **. (a) the difference between the L05-1 system and the L05-2 system and (b) between the L05-1 system and the L05-3 system.**



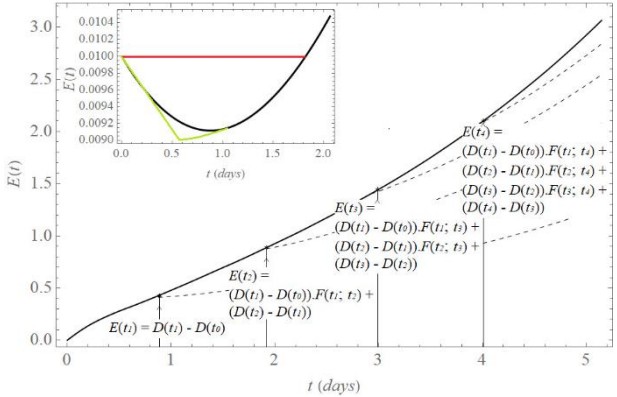

875

**Figure 12. Hypothesis $E_D(t)$ explaining the model error growth $E_M(t)$ (black curve). The drift increment $D(t_k) - D(t_{k-1})$ at each time step $\Delta t = t_k - t_{k-1}$, $k = 1, \ldots, K$ is taken as the error of the initial conditions with an exponential growth $e^{\lambda t}$ driven by the largest Lyapunov exponent $\lambda$ of the "model" (L05-1 system). Since $\vec{d}(t)$ does not point into the locally most unstable direction, time evolution of $D(t_k) - D(t_{k-1})$ decrease in early time (black curve in the inset). A constant (red curve in the inset) or linear decrease (green curve in the inset) approximates this initial decrease. $D(t_k) - D(t_{k-1})$ evolves with time $t_i$ (dashed curves) in the constant approximation as: $F_{con}(t_k; t_i) = 1$ for $t_k \le t_i \le t_{M+k}$ and $F_{lin}(t_k; t_i) = e^{\lambda(t_i - t_k)}$ for $t_{M+k+1} \le t_i \le t_K$, and in the linear approximation as: $F_{lin}(t_k; t_i) = 1 - \sigma(t_i - t_k)$ for $t_k \le t_i \le t_{M+k}$ and $F_{lin}(t_k; t_i) = (1 - \sigma(t_{M+k} - t_k)) e^{\lambda(t_i - t_k)}$ for $t_{M+k+1} \le t_i \le t_K$. $M$ and $\sigma$ are found experimentally. The resulting hypothesis $E_D(t)$ describing the model error growth $E_M(t)$ (black curve) is the sum of the individual increments: $E_M(t_i) \approx E_{D,ap}(t_i) = \sum_{k=1}^{i} (D(t_k) - D(t_{k-1})) \cdot F_{ap}(t_k; t_i)$ where $ap$ is the symbol for the constant ($con$) or linear ($lin$) approximation.**




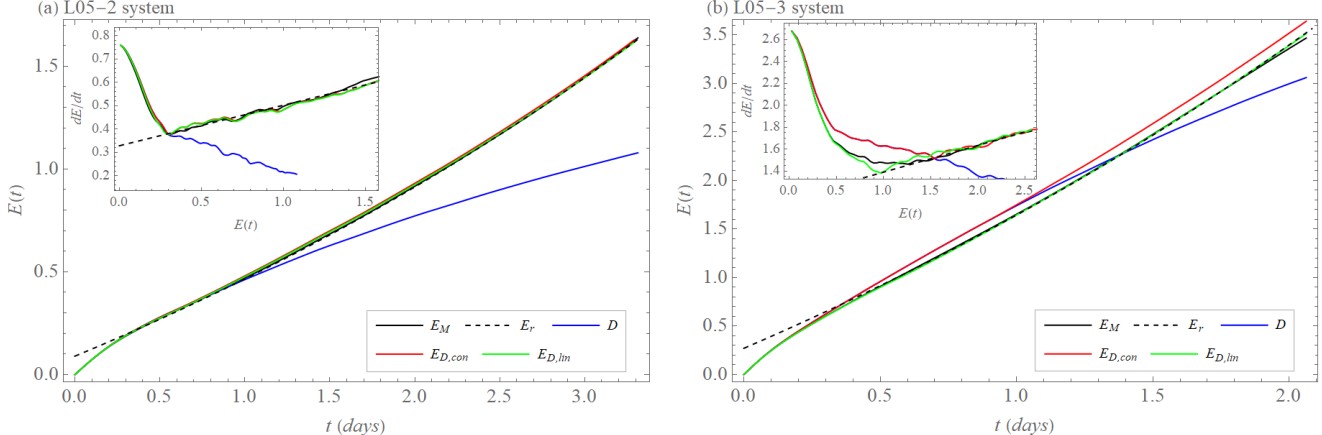

**Figure 13. Approximation of model error growth** $E_M(t)$ **(black curve) by exponential growth with model error** $E_r$ **(black, dashed curve) and by hypotheses** $E_{D,con}$ **(Eqs. (19) and (21), red curve) and** $E_{D,lin}$ **(Eqs. (20) and (21), green curve) based on drift** $D$ **(blue curve). (a) Calculation of model error** $E_M$ **and drift** $D$ **from the difference between the L05-1 and L05-2 systems. The approximation** $E_{D,con}$ **is with** $M = 28$, $\lambda = 0.27$ **1/day, and approximation** $E_{D,lin}$ **is with** $M = 24$, $\sigma = 0.001$, **and** $\lambda = 0.27$ **1/day. (b) Calculation of model error** $E_M$ **and drift** $D$ **from the difference between the L05-1 and L05-3 systems. The approximation** $E_{D,con}$ **is with** $M = 43$, $\lambda = 0.38$ **1/day, and the approximation** $E_{D,lin}$ **is with** $M = 27$, $\sigma = 0.005$, **and** $\lambda = 0.38$ **1/day. The insets show the time differences (rates) of the quantities as a function of the quantities presented in the main figures.**





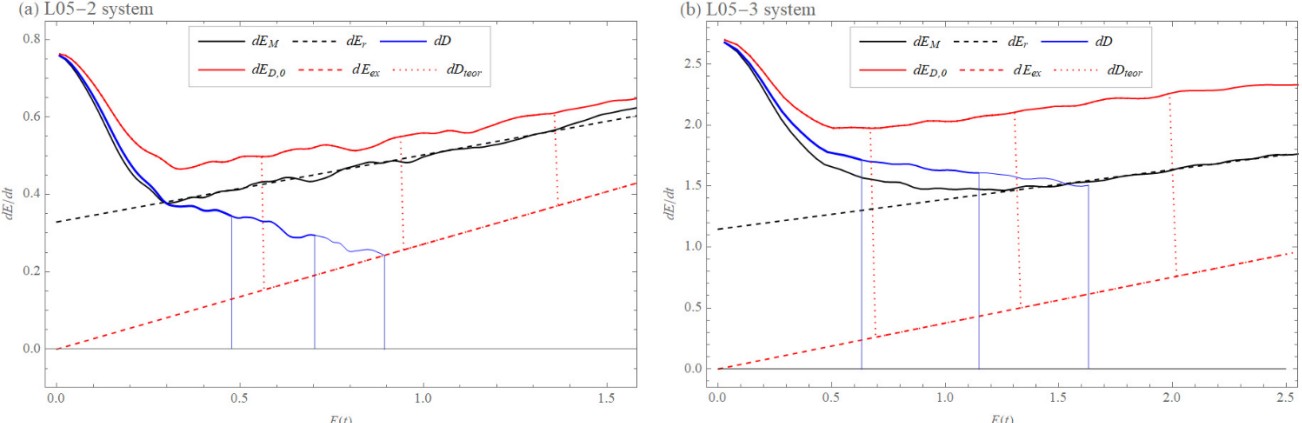

**Figure 14. The validity of Eq. (23) ($dD/dt\big(D(t_k)\big) = dD/dt\big(E_{D,0}(t_k)\big) = \dfrac{dE_{D,0}}{dt}\big(E_{D,0}(t_k)\big) - \lambda \cdot E_{D,0}(t_k)$) is shown by vertical lines, where the length of the blue ones is the same as the length of the red dotted $dD_{teor}$ ones at times $t_1 = 1$ day, $t_2 = 1.75$ day, and $t_3 = 2.5$ day (from left to right) for (a) the difference between the L05-1 and L05-2 systems and at times $t_1 = 0.3$ day, $t_2 = 0.6$ day, and $t_3 = 0.9$ day (from left to right) for (b) difference between L05-1 and L05-3 systems, where $dD = dD/dt(D)$ (Eq. (18), blue curve) is time difference of drift, $dE_M = dE_M/dt(E_M)$ (Eq. (14), black curve) is model error growth, $dE_{D,0} = dE_{D,0}/dt(E_{D,0})$ is a hypothesis Eq. (23) (red curve), $dE_r$ (Eq. (5), black dashed curve) is exponential growth with model error ($dE_{r,L05-2} = 0.17 \cdot E + 0.33$, $dE_{r,L05-3} = 0.25 \cdot E + 1.15$), and $dE_{ex}$ (Eq. (1), red dashed line) is exponential growth with the value of $\lambda$ determined from the quadratic hypothesis with model error $dE_q$ (Eq. (6)) ($\lambda_{L05-2} \rightarrow dE_q = (0.27 \cdot E + 0.34)(1 - E/7.6)$, and $\lambda_{L05-3} \rightarrow dE_q = (0.38 \cdot E + 1.47)(1 - E/7.8)$).**



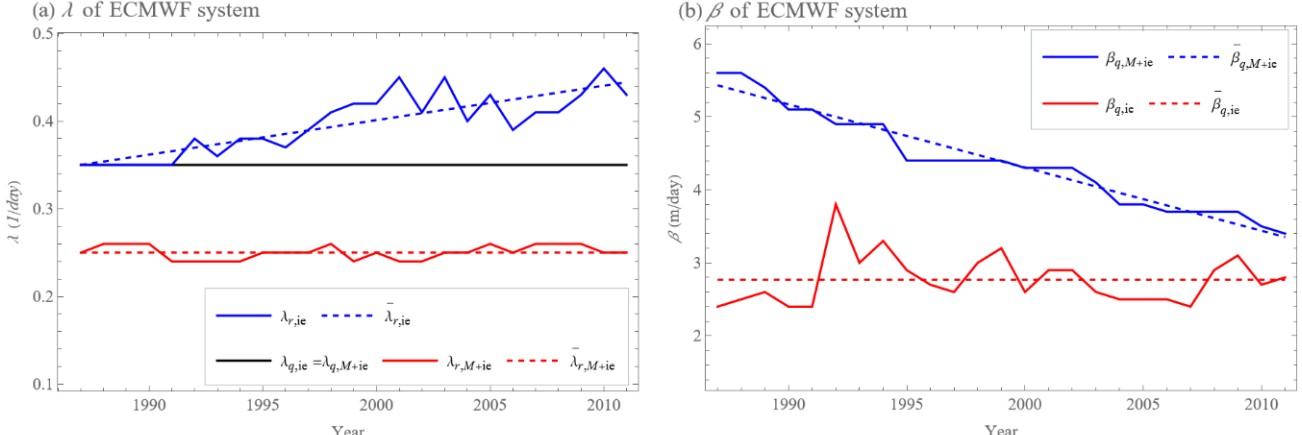

**Figure 15. Values of parameters $\lambda$ (a) and $\beta$ (b) of exponential growth with model error $dE_r$ and quadratic hypothesis with model error $dE_q$ approximated from annual averages (1987 – 2011) of the error growth tendencies (rates) $dE_{EFS,ie}/dt$ and $dE_{EFS,M+ie}/dt$ of the ECMWF forecasting system's 500 hPa geopotential height values over the Northern Hemisphere (for more details, see section 5.1). (a) The black curve is $\overline{\lambda}_q = 0.35$ 1/day determined as the average of the approximated $\lambda_q$ of $dE_q$ hypothesis over 25 annual averages of $E_{EFS,ie}(t)$ and $E_{EFS,M+ie}(t)$. The full red curve shows the values $\lambda_r$ of the $dE_r$ approximation of the 25 annual averages of $dE_{EFS,M+ie}/dt$. The red dashed curve shows that the best approximation of $\lambda_{r,M+ie}$ is a constant function with $\overline{\lambda}_{r,M+ie} = 0.25$ 1/day. The full blue curve shows the values $\lambda_r$ of the $dE_r$ approximation of the 25 annual averages of $dE_{EFS}/dt$. The blue dashed curve shows that the best-fitting approximation of $\lambda_r$ is a linear function $\overline{\lambda}_{r,ie}$ that increases with years. (b) Values of $\beta_q$ of $dE_q$ hypothesis approximating 25 annual averages of $dE_{EFS,ie}/dt$ (full red curve) and $dE_{EFS,M+ie}/dt$ (full blue curve). The red dashed curve shows that the best approximation of $\beta_{q,ie}$ is a constant function with $\overline{\beta}_{q,ie} = 2.8$ m/day. The blue dashed curve shows that the best-fitting approximation of $\beta_{q,M+ie}$ is a linear function $\overline{\beta}_{r,ie}$ decreasing with years.**





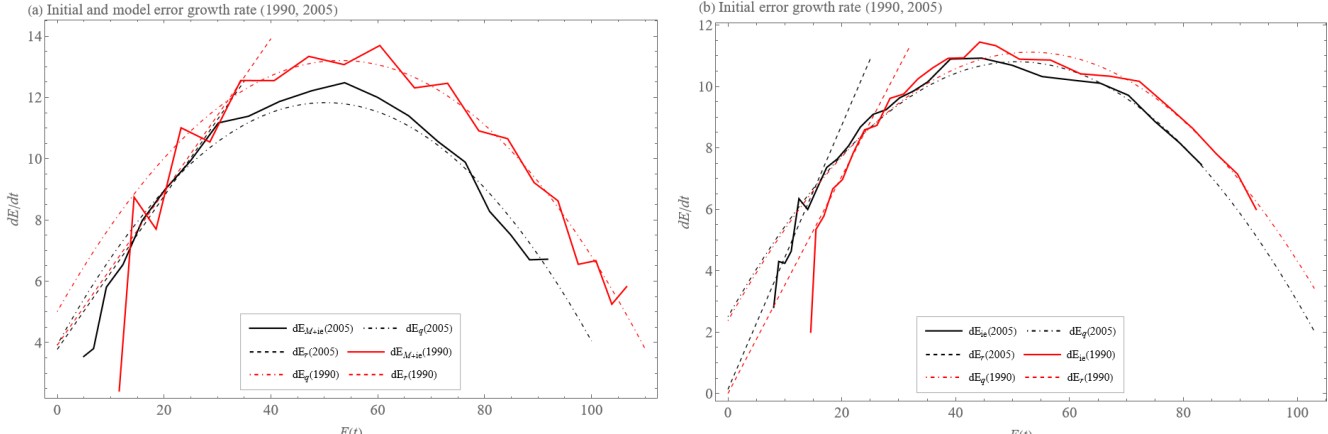

**Figure 16.** Annual average from 2005 data of the error growth tendencies (rates) $dE_{EFS,M+ie}/dt$ **(a)** and $dE_{EFS,ie}/dt$ **(b)** of the ECMWF forecasting system's 500 hPa geopotential height values over the Northern Hemisphere (full curves), approximation by the quadratic hypothesis with model error $dE_q$ **(Eq. (6))** with a constant value of the parameter $\lambda_q = 0.35$ **1/day (dot-dashed curves)** and approximation by the exponential growth with model error $dE_r$ **(Eq. (5), dashed curves). We compare the data from the year 1990 (red) to those of 2005 (black).**