# Peer review of "Analysis of model error in forecast errors of Extended Atmospheric Lorenz' 05 Systems and the ECMWF system"

_EGUsphere, 2023_

## Referee Report (RR1)

A review report for the manuscript, titled "***Analysis of model error in forecast errors of extended atmospheric Lorenz' 05 systems and the ECMWF system*** by Bednář and Kantz"

**Review Summary:**

The study applies the Lorenz 2005 models to investigate the forecast error growth in atmospheric predictions, attributing it to initial and model errors. It focuses on the impact of small-scale phenomena on predictability, questioning whether omitting them would enhance forecast accuracy. Using the Lorenz 2005 one-, two-, and three-scale systems, the research reveals that excluding small spatiotemporal scales diminishes predictability more than modeling them. Contrary to expectations, omitting phenomena does not improve predictability; instead, it results in increased model error. The study proposes a hypothesis explaining this behavior, linking model error to the differences between systems at each time step. Fit parameters are used to compare the hypothesis with approximations of average forecast error growth, interpreting them in the context of model error. The findings were applied to the ECMWF system to reveal the reduction of model error from 1987 to 2011 based on the hypothesis, despite a concurrent growth in system instability related to initial condition errors.

This study is interesting and has the potential to improve our understanding of predicting capabilities within idealized and real-world models. However, there are some major issues that require further clarification. The reviewer suggests major revisions, and specific comments are provided below.

**Major Comments:**

Most of Lorenz's models were developed to effectively illustrate the chaotic and unstable nature of weather and climate and/or estimate the growth rates of the systems (which include the natural system in Lorenz 1969a and/or numerical models). While some of Lorenz's models (e.g., the Lorenz 1963 model) have been extensively studied by researchers in various fields for over 50 years, the Lorenz 1996/2005 model is still relatively young and the 1996 and 2005 versions are not exactly the same. Below, major features of the Lorenz 1996 and 2005 models are provided.
- The 1996 and 2005 models were developed to illustrate the growth of errors for chaotic responses that contain one or two scales.
- However, these models were not derived from physics-based systems (i.e., partial differential equations).
- These models contain constant coefficients for nonlinear terms, dissipative terms, and forcing terms. Thus, these models lack some realistic features (e.g., differences between ocean and land; Lorenz and Emanuel 1998).

Regarding the analysis of errors, the following assumptions were applied in most studies:
- Saturation for error growth,

- Quadratic hypothesis for error growth, and
- Monostability for single type of chaotic solutions (in contrast to multistability for coexisting attractors).

Thus, by considering the above features, assumptions, and the following quote from Lorenz himself:
"

> I have not developed anything resembling a general theory of model design. What successes I have enjoyed have resulted from trial and error, but not, however, from random trial and error. Each satisfactory attempt has been guided by the detailed analysis of previous failures. I make no claim to have discovered the ideal equations (Lorenz 2005),

"

findings obtained using these idealized models should be explained with caution. For example, what can we learn when the 2005 model produces a comparable growth rate (i.e., doubling time), as compared to the real world model?

The 1996 versions of the models, including one-scale and two-scale models, were first proposed in a report (Lorenz, 1996). Later, Prof. Lorenz made an attempt to propose improved models in 2005. While the 1996 and 2005 versions of the models include the same one-scale model, they contain different two-scale models. The 2005 version of the two-scale model has a shorter history. As a result, analysis of stability within the 2005 two-scale models and sensitivities of findings on parameters should be explored further to support their conclusion.

Please provide discussions and/or responses to clarify or address the following:

(A) Different two-scale models in Lorenz (1996) and Lorenz (2005)

Figure R1 displays the two-scale model proposed by Lorenz (1996, 2006), including Eqs. (3.2)-(3.3) of Lorenz (2006). It is worth mentioning that Lorenz (1966) and Lorenz (2006) are the same article. Eq (3.2) for the large-scale flow does not include the explicit forcing term "F", which appears in his one-scale model. This is a typo. For the small-scale flow in Eq. (3.3), where F is not explicitly included, the coupling term acts as the forcing to derive the small scale process. Within the two-scale model, the grid system was illustrated in Figure R2 derived from Wilks (2005). Such a grid system is similar to the grid system of the multiscale modeling framework (MMF, e.g., Tao et al. 2008; Shen et al. 2011), consisting of a general circulation model (GCM, e.g., Lin et al. 2003; Lin 2004; Shen et al. 2006) for large-scale flows, and multiple copies of a cloud model (e.g., Tao 2003) for small-scale flows. Specifically, a copy of the cloud model at fine resolutions is embedded within each grid of the GCM.

Within the 2005 models, Lorenz first included additional nonlinear terms in the 1996 one-scale model (e.g., Eq. 8 in Figure R3) for slow variables (represented as $X_n$). Based on the 1996 one-scale model with coefficients of ("$b^2$", "$b$", "$0$") for nonlinear terms, dissipative terms, and forcing term, respectively, a subsystem for fast variables (represented as $Y_n$) was deployed and coupled with the subsystem for the slow variables. The coupled system with a

coefficient of "c" for coupling terms is referred to as the two-scale system (Eqs. 12a and 12b in Figure R4). The coupling terms were established based on a one-to-one relationship between Xn and Yn. Thus, the Lorenz 2005 two-scale model is different from the 1996 two-scale model. Will it be feasible for providing a diagram for illustrating the grid system within the 2005 two-scale model?

(B) Dependence of findings on temporal spacing (i.e., $\Delta t$) and "spatial" spacing (e.g., the number of sectors, N)

As an analogy, the CFL condition, requiring $c\Delta t/\Delta x < 1$, here c is the space speed, suggests the importance of selecting temporal and spatial spacings for solution's stability. In this study, it is important to explore the impact of $\Delta t$ and N.

Similarly, the concept of computational chaos (Lorenz 1989) also suggests the importance of wisely choosing $\Delta t$. Computational chaos appears "when the exact solution varies periodically with time, there is sometimes a range of time increment where the computed solution is chaotic" (Lorenz 2006). Computational chaos can be illustrated by a comparison of the Logistic differential equation and the Logistic map (i.e., difference equation). While the former has analytical, regular solutions, the latter produces chaotic solutions when a control parameter is sufficiently large. A dependence of the control parameter on a temporal spacing (i.e., $\Delta t$) can be shown by deriving the Logistic map from the Logistic differential equation (Shen et al. 2023).

In this study, $\Delta t$ is $1/240 \sim 4.2 \times 10^{-3}$ unit, N = 360 (indicating a "spatial" spacing), and L = 12 (i.e., indicating complexities of scale interaction). It would be ideal for additional tests with a smaller $\Delta t = 10^{-5}$ (or $\Delta t = 10^{-4}$). Additionally, the choice of N and L should be explored since N = 960 and L = 32 were used in Lorenz (2005).

As discussed below, the values of the coefficients for the coupling terms could impact the growth rate of the system as well.

(C) Impact of model's configuration and complexity on critical points (equilibrium points)

Based on the linearization theorem, critical points of the Lorenz systems could roughly indicate the local behavior of the solutions. As a result, initial error growth should display a dependence on the equilibrium state. Please consider identifying the appearance of the critical points and perform stability analysis using the Jacobian matrix of the linearized system at each of the critical points.

Below, a simple illustration for the linear stability analysis is provided using the 1996 one-scale model with N = 5. Based on the Figure R5 and Table R1, it is suggested that a larger F may produce a larger eigenvalue (a larger real part of the eigenvalue), suggesting a larger growth rate.

Based on the following preliminary analysis of the one- and two-scale models with the same value of the forcing parameter F, the effective forcing parameter for the two-scale model is smaller, yielding a smaller leading eigenvalue (i.e., a smaller real part of the eigenvalue). This is consistent with the finding that Figures 5 and 6 display larger growth rates ($\lambda$) within the one-scale system (e.g., L05-1) than the two-scale system (e.g., L05-2). [Such a finding is supported by the so-called aggregated negative feedback reported by Shen 2014, 2019.]

Consider Eqs. (A2) and (A3). From the nonlinear terms of Eq. (A2) and (A3), we expect that $X_{1,1} = X_{1,2} = X_{1,3} = \cdots X_{1c}$ and $X_{2,1} = X_{2,2} = \cdots X_{2c}$ may be a critical point. Here, $X_{1c}$ and $X_{2c}$ represent the value of steady state solutions for the slow and fast variables, respectively. From Eq. (A3), we have $X_{2c} = cX_{1c}/b$. Plugging the above into Eq. (A2), the right hand side of Eq. (A2) contains two dissipative terms, $-X_{1,n}$ and $-c^2X_{1c}/b$, yielding $X_{1c} = bF/(b + c^2) < F$. Namely, the effective forcing for slow variables is weaker, indicating a smaller growth rate within the two-scale model, as compared to the one-scale model.

On the other hand, the above along with Figure R5 and Table R1 only provide a preliminary, qualitative, analysis. The authors may want to further verify or comment the above since the Jacobian matrix for the two-scale system that includes fast variables is larger, as compared to the Jacobian within the corresponding one-scale system.

For example, with the two-scale or three-scale system, the value of parameter "b1" (b1 > 1) determine the (temporal) scale as well as the magnitude of the fast variables. Please provide justifications for the choice of b1 = 10 for the two-scale system but b1 = 1 for the three-scale system.

Additionally, within the three-scale system, are nonlinear terms (e.g., c1 and c2 in Eq. A9) applied for coupling the "sub-systems" for the small- and medium-scale variables with the large-scale system? Please comment on the impact of c1 and c2 on system's stability.

(D) Separations of initial and model errors

Based on the linearization theorem, a locally linearized system may represent the local feature of the corresponding nonlinear system (for a hyperbolic critical point). The stability of the linearized system depends on locations of the critical points that depend on model's complexity (i.e., nonlinear terms in the system). Thus, the model complexity (i.e., nonlinear terms) could impact the critical points and thus the growth of the initial errors. As a result, it is not easy to separate the initial errors and model errors. (For example, given the same initial error for a large-scale variable, the time varying difference between two nearby trajectories are different in two different models.)

(E) Validity of error saturation for periodic attractors and coexisting attractors

Earlier studies suggest that the Lorenz 1996 two-scale model could produce nonlinear periodic solutions. In your ensemble runs, have you observed periodic solutions? Can you comment on the validity of error saturation for periodic solutions?

Additionally, recent studies reported the appearance of multistability (for coexisting attractors) within the 1996 model (e.g., Van Kekem and Sterk 2018a,b, 2019; Pelzer et al. 2020). Have you observed multistability in your ensemble runs?

**Specific Comments:**

(1) Please check consistency in the capitalization of the initial letters of words within a title.

(2) Lines 45-50, the application of the Lyapunov exponent (LE) is not accurate. A global LE represents a long-term average of "local" growth rates (determined by the separations of two nearby trajectories). Initial separations should remain small. Local growth rates may vary with time. As a result, Eq. (1) with a constant growth rate is valid only for a finite time interval. During different time internals, different growth rates may appear. Note that in addition to one positive LE, solution's boundedness is another important feature that defines a chaotic system.

(3) Lines 45-55, please consider referring to the growth rates in Eqs. (1) and (2) as the exponential growth rate (with a J-shaped curve) and logistic growth rate (with a S-shaped curve), respectively.

(4) Line 80, the term "error growth laws" should be rephrased since they are not necessarily physical laws.

(5) Lines 122, statements are not accurate. Unless additional forcing terms are introduced, improving model's spatial or temporal resolution does not necessarily enhance instability. (Please think of a convergent Taylor series.)

The impact of additional dissipative terms and/or additional heating term has been previously examined using the Lorenz 1963 model (e.g., Shen 2014, 2015, 2019).

(6) Lines 128-130: it is wired that the two-scale system contains large- and small-scale systems while the three-scale system adds a medium scale, in addition to large- and small-scale flows. Any justifications?

(7) Lines 160-165, have you observed coexisting attractors (e.g., more than one attractors) in your ensemble runs? (e.g., see multistability in Van Kekem and Sterk 2018a,b, 2019; Pelzer et al. 2020)

(8) Line 170, does the statement "errors might even shrink in short times" indicates the existence of a stable manifold?

(9) Lines 194, while N=360 was used in this study, N=960 was appied in Lorenz (2005).

(10) line 186, how many time steps for the transfer of error to the small-scale variables?

(11) Section 3.1, please confirm whether the leading LE in the L05-1 system is larger (smaller) than that in the L05-2 (L05-03) system.

(12) Line 382-394: The key point that higher resolution model produces better predictability is acceptable. However, it is not clear whether Figure 10 is sufficient to support this point. Please see details in the last specific comment below.

(13) Line 656: The statement "Based on the fact that scale-dependent error growth implies an intrinsic predictability limit" is not accurate. A finite growth rate may indicate a limit for practical predictability. By comparison, a finite intrinsic predictability is established by the feature of chaos (e.g., sensitive dependence on initial condition, SDIC; e.g., Shen, Pielke Sr., and Zeng, 2023).

(14) Lines 612 - 623, discussions are duplicated; they are the same as those in Lines 600-611.

(15) Line 715, the parameter "K" should be replaced by "L".

(16) Line 716, Lorenz (2005) did not explicitly suggest the ratio of N/L = 30 nor provide justification for the choice of N = 960 and L = 32.

(17) page 40, line 870-875, Figure 10. Figure's title and captions are confusing. Since L05-02 and L05-03 systems were used to provide the "ground true" (or reference) for computing errors, these errors do not represent the errors of the L05-02 and L05-03 systems, respectively, the growth of initial errors within the L05-02 or L05-03 system does contribute to the growth of differences of the solutions between the L05-1 and L05-02 (or L05-03) systems.

For a comparison in Figures 5-7, let's simply choose $\lambda_{ex}$ = 0.33, 0.29, and 0.46 for the L05-1, L05-2, and L05-3 systems, respectively. The comparison of the above selected growth rates produces a consistent finding that larger differences (in error growths) are reported in Figure 10b than in Figure 10a. However, on the other hand, considering differences between the L05-02 and L05-03 systems, the differences may produce the largest growth rates as compared to those in Figure 10a and Figure 10b.

Table R1: An eigenvalue analysis of the Lorenz 1995 one-scale model. The corresponding Jacobian matrix is shown in Figure R5. Here, Xc = F.

| Xc | eigenvalues |
|---|---|
| 0.5 | -0.4410 + 0.7694i |
| | -0.4410 - 0.7694i |
| | -1.0000 + 0.0000i |
| | -1.5590 + 0.1816i |
| | -1.5590 - 0.1816i |
| 1 | 0.1180 + 1.5388i |
| | 0.1180 - 1.5388i |
| | -1.0000 + 0.0000i |
| | -2.1180 + 0.3633i |
| | -2.1180 - 0.3633i |
| 10 | 10.1803 +15.3884i |
| | 10.1803 -15.3884i |
| | -1.0000 + 0.0000i |
| | -12.1803 + 3.6327i |
| | -12.1803 - 3.6327i |
| 20 | 21.3607 +30.7768i |
| | 21.3607 -30.7768i |
| | -1.0000 + 0.0000i |
| | -23.3607 + 7.2654i |
| | -23.3607 - 7.2654i |
| 30 | 32.5410 +46.1653i |
| | 32.5410 - 46.1653i |
| | -1.0000 + 0.0000i |
| | -34.5410 +10.8981i |
| | -34.5410 -10.8981i |

How good are such naive estimates? We can demonstrate some simple systems where they describe the situation rather well, at least on the average. One system is one that I have been exploring in another context as a one-dimensional atmospheric model, even though its equations are not much like those of the atmosphere. It contains the $K$ variables $X_1, \ldots, X_K$, and is governed by the $K$ equations

$$dX_k/dt = -X_{k-2}X_{k-1} + X_{k-1}X_{k+1} - X_k + F, \tag{3.1}$$

where the constant $F$ is independent of $k$. The definition of $X_k$ is to be extended to all values of $k$ by letting $X_{k-K}$ and $X_{k+K}$ equal $X_k$, and the variables may be thought of as values of some atmospheric quantity in $K$ sectors of a latitude circle. The physics of the atmosphere is present only to the extent that there are external forcing and internal dissipation, simulated by the constant and linear terms, while the quadratic terms, simulating advection, together conserve the total energy $(X_1^2 + \cdots + X_K^2)/2$.

distinct time scales. The model has been constructed by coupling two systems, each of which, aside from the coupling, obeys a suitably scaled variant of Eq. (3.1). There are $K$ variables $X_k$ plus $JK$ variables $Y_{j,k}$, defined for $k = 1, \ldots, K$ and $j = 1, \ldots, J$, and the governing equations are

$$dX_k/dt = -X_{k-1}(X_{k-2} - X_{k+1}) - X_k - (hc/b)\sum_{j=1}^{J} Y_{j,k}, \tag{3.2}$$

$$dY_{j,k}/dt = -cbY_{j+1,k}(Y_{j+2,k} - Y_{j-1,k}) - cY_{j,k} + (hc/b)X_k. \tag{3.3}$$

The definitions of the variables are extended to all values of $k$ and $j$ by letting $X_{k-K}$ and $X_{k+K}$ equal $X_k$, as in the simpler model, and letting $Y_{j,k-K}$ and $Y_{j,k+K}$ equal $Y_{j,k}$, while $Y_{j-J,k} = Y_{j,k-1}$ and $Y_{j+J,k} = Y_{j,k+1}$. Thus, as before, the variables $X_k$ can represent the values of some quantity in $K$ sectors of a latitude circle, while the variables $Y_{j,k}$, arranged in the order $Y_{1,1}, Y_{2,1}, \ldots, Y_{J,1}, Y_{1,2}, Y_{2,2}, \ldots, Y_{J,2}, Y_{3,1}, \ldots$, can represent the values of some other quantity in $JK$ sectors. A large value of $J$ implies that many of the latter sectors are contained in one of the former, and we may think of the variables $Y_{j,k}$ as representing a convective-scale quantity, while, in view of the form of the coupling terms, the variables $X_k$ should represent something that favours convective activity, possibly the degree of static instability.

Figure R1: Lorenz 1996 one-scale (top) and two-scale (bottom) systems (Lorenz 2006). Since the 1996 model was applied to represent an atmospheric variable in K sectors of a latitude circle. Thus, the value of K indicates the number of grid points within the large-scale system, while the value of J represents the number of grid points within the small-scale system. Compared to the 2005 version, (1) the last term in Eq. (3.2) represents a feedback term that is a summation of small scale modes and (2) both Eqs. (3.2) and (3.3) contain two nonlinear terms, involving three neighboring grid points (at k-1, k+1, k+2).

The Lorenz '96 system (Lorenz 1996) is given by:

$$\frac{dX_k}{dt} = -X_{k-1}(X_{k-2} - X_{k+1}) - X_k + F - \frac{hc}{b} \sum_{j=J(k-1)+1}^{kJ} Y_j; \quad k = 1, \dots, K$$
(1a)

$$\frac{dY_j}{dt} = -cbY_{j+1}(Y_{j+2} - Y_{j-1}) - cY_j + \frac{hc}{b} X_{\text{int}[(j-1)/J]+1}; \quad j = 1, \dots, JK.$$
(1b)

It is is used here to define 'truth,' i.e. the quantities to be predicted. This system has been used in several previous studies as a metaphor for the atmosphere (Lorenz 1996; Palmer 2001; Smith 2001; Orrell 2002, 2003; Vannitsem and Toth 2002; Roulston and Smith 2003), although with slightly different notation. Equation (1a) describes the linked dynamics of a set of $K$ slow, large-amplitude variables $X_k$, each of which is associated with $J$ fast, small-amplitude variables $Y_j$ whose dynamics are described by Eq. (1b). Here $K = 8$ and $J = 32$, so that there are $JK = 256$ $Y$ variables in total, as illustrated in Fig. 1. The scaling constants $h$, $c$, and $b$ are taken to be 1, 10, and 10, respectively, as is conventional; and $F$ is a forcing taken in the following to be either 18 or 20. The subscripts are cyclic so, for example, $X_0 = X_K$, $X_{-1} = X_{K-1}$, etc. and likewise for the $Y$ variables.

[Figure]

Figure 1.  Schematic illustration of the Lorenz '96 system (Eq. (1)) with $K = 8$ resolved variables $X_k$ (large circles), each associated with $J = 32$ unresolved variables $Y_j$ (unlabeled small circles) grouped according to the $X$ variable to which they belong), so that there are $JK = 256$ $Y$ variables in total. The forecast model (Eq. (2)) represents explicitly only the $X$ variables, with contributions to each tendency that are due to the unresolved scales being parametrized in terms of the local resolved variable only.

Figure R2: Mathematical equations (top) and grid systems (bottom) within the Lorenz 1996 two-scale model (e.g., Wilks 2005). Eight large-scale variables (denoted as X) are selected at 8 data points within the large-scale system. Each large-scale variable acts as a force to drive a small-scale system consisting of thirty-two variables (denoted as Y). Based on the linear stability analysis, local growth rates should display a dependence on the number of data points in both the large-scale and small-scale systems and the coupling terms between the two-scale systems.

> 2. One chooses a number $K$, much smaller than $N$ and let $J = K/2$ if $K$ is even and $J = (K - 1)/2$ if $K$ is odd. Then, for any two sets of variables $X$ and $Y$, one defines
>
> $$[X, Y]_{K,n} = \sum_{j=-J}^{J}{}' \sum_{i=-J}^{J}{}' (-X_{n-2K-i}Y_{n-K-j}$$
>
> $$+ X_{n-K+j-i}Y_{n+K+j})/K^2 \qquad (7)$$
>
> if $K$ is even, with $\Sigma'$ replaced by $\Sigma$ if $K$ is odd. The equation for Model II, where the only set of variables is $X$, will be
>
> $$dX_n/dt = [X, X]_{K,n} - X_n + F. \qquad (8)$$
>
> Note that setting $K = 1$ makes $J = 0$; hence $[X, X]_{1,n}$ represents the single pair of products appearing in Eq. (1). Model II then reduces to Model I.

Figure R3: Equation (8) the above excerpt represents a revised one-scale model, proposed as the uncoupled version of the Model II in Lorenz (2005). The notation of [X, X] defined in Equation (7) indicates nonlinear terms. Compared to the original one-scale model in Figure R1 that contains a pair of nonlinear terms, a value of K > 1 in Eq. (8) suggests more than one pair of nonlinear terms. From a perspective of scale interactions (e.g., Lorenz 1969b), additional nonlinear terms (at different grid points) may improve the representation of scale interaction.

> entirely by the coupling. One obtains the system
>
> $$dX_n/dt = [X, X]_{K,n} - X_n - cY_n + F, \qquad (12a)$$
>
> $$dY_n/dt = b^2[Y, Y]_{1,n} - bY_n + cX_n, \qquad (12b)$$
>
> where, like $b$, the coupling coefficient $c$ is an additional parameter of the model.

Figure R4: Equation (12) in the above excerpt represents a revised two-scale model, proposed as Model II in Lorenz (2005). Here, Eq. (12a) is a revised large-scale system with more than one pair of nonlinear terms (when K > 1). Eq. (12b) indicated a revised small-scale system with one pair of nonlinear terms. In the coupled system, there exists a one-to-one relationship between the large-scale variable Xn and the small-scale variable Yn within the coupling terms.

$$J_{L96} = \begin{pmatrix} -1 & X_c & 0 & -X_c & 0 \\ 0 & -1 & X_c & 0 & -X_c \\ -X_c & 0 & -1 & X_c & 0 \\ 0 & -X_c & 0 & -1 & X_c \\ X_c & 0 & -X_c & 0 & -1 \end{pmatrix}.$$

Figure R5: A Jacobian matrix for the linearized version of the Lorenz 1996 model with N = 5 from Eq. 3.1 in Figure R1. Here, Xc indicates a critical point solution and is equal to F.

**References:**

- Lin, S.-J., B.-W. Shen, W. P. Putman, J.-D. Chern, 2003: Application of the high-resolution finite-volume NASA/NCAR Climate Model for Medium-Range Weather Prediction Experiments. EGS - AGU - EUG Joint Assembly, Nice, France, 6 - 11 April 2003
- Lin, S.-J., 2004: A vertically Lagrangian finite-volume dynamical core for global models, Mon. Weather Rev.,132, 2293–2307.
- Lorenz, E.N., 1969b:The predictability of a flow which possesses many scales of motion. *Tellus*, *21*, 289–307.
- Lorenz, E.N., 1969a: Atmospheric predictability as revealed by naturally occurring analogues. *J. Atmos. Sci.*, *26*, 636–646.
- Lorenz, E. N., 1989: Computational chaos: a prelude to computational instability. Physica, 35D, 299-317.
- Lorenz, E. N., 1996: Predictability: A problem partly solved. Proc. Seminar on Predictability, Vol. 1, ECMWF, Reading, Berkshire, UK, 1–18.
- Lorenz, E. N.,, and K. A. Emanuel, 1998: Optimal sites for supplementary weather observations: Simulation with a small model. J. Atmos. Sci., 55, 399–414.
- Lorenz, E., 2005a: Designing chaotic models. J. Atmos. Sci., 62, 1574-1587.
- Lorenz, E., 2006: Regimes in simple systems. J. Atmos. Sci., 63, 2056–2073.
- Pelzer, Anouk F. G., and Alef E. Sterk. 2020. "Finite Cascades of Pitchfork Bifurcations and Multistability in Generalized Lorenz-96 Models" Mathematical and Computational Applications 25, no. 4: 78. https://doi.org/10.3390/mca25040078
- Shen, B.-W., 2019: Aggregated Negative Feedback in a Generalized Lorenz Model. International Journal of Bifurcation and Chaos, Vol. 29, No. 3 (2019) 1950037 (20 pages). https://doi.org/10.1142/S0218127419500378
- Shen, B.-W., 2015: Nonlinear Feedback in a Six-dimensional Lorenz Model. Impact of an additional heating term. Nonlin. Processes Geophys., 22, 749-764, doi:10.5194/npg-22-749-2015, 2015. (link) (pdf)

- Shen, B.-W., 2014: Nonlinear Feedback in a Five-dimensional Lorenz Model. J. of Atmos. Sci., 71, 1701–1723. doi:http://dx.doi.org/10.1175/JAS-D-13-0223.1
- Shen, B.-W., R. A. Pielke Sr., and X. Zeng 2023: The 50th Anniversary of the Metaphorical Butterfly Effect since Lorenz (1972): Special Issue on Multistability, Multiscale Predictability, and Sensitivity in Numerical Models. [Editorial] Atmosphere 2023, 14(8), 1279; https://doi.org/10.3390/atmos14081279
- Shen, B.-W., W.-K. Tao, and B. Green, 2011: Coupling Advanced Modeling and Visualization to Improve High-Impact Tropical Weather Prediction(CAMVis), IEEE Computing in Science and Engineering (CiSE), vol. 13, no. 5, pp. 56-67, Sep./Oct. 2011, doi:10.1109/MCSE.2010.141
- Shen, B.-W., R. Atlas, O. Oreale, S.-J Lin, J.-D. Chern, J. Chang, C. Henze,and J.-L. Li, 2006b: Hurricane Forecasts with a Global Mesoscale-Resolving Model: Preliminary Results with Hurricane Katrina(2005). Geophys. Res. Lett., L13813, doi:10.1029/2006GL026143.
- Van Kekem, D.; Sterk, A. Travelling waves and their bifurcations in the Lorenz-96 model. Phys. D Nonlinear Phenom. 2018a, 367, 38–60.
- Van Kekem, D.; Sterk, A. Wave propagation in the Lorenz-96 model. Nonlinear Process. Geophys. 2018b, 25, 301–314.
- Van Kekem, D.; Sterk, A. Symmetries in the Lorenz-96 model. Int. J. Bifurc. Chaos 2019, 29, 1950008.
- Tao, W.-K. 2003. Goddard Cumulus Ensemble (GCE) Model: Application for Understanding Precipitation Processes Meteorological Monographs 29 (51): 107-107 [10.1175/0065-9401(2003)029<0107:CGCEGM>2.0.CO;2]
- Tao, W.-K. Tao, D. Anderson, J. Chern, J. Entin, A. Hou, P. Houser, R. Kakar, S. Lang, W. Lau, C. Peters-Lidard, X. Li, T. Matsui, M. Rienecker, M. R. Schoeberl, B.-W. Shen, J. J. Shi, and X. Zeng, 2009: A Goddard Multi-Scale Modeling System with Unified Physics. Special Issue dedicated to The 1st International Conference on From Desert to Monsoons. Ann. Geophys., 27, 3055-3064
- Wilks, D., 2005: Effects of stochastic parametrizations in the Lorenz '96 system. Q. J. Roy. Meteor. 335 Soc., 131 (606), 389–407, doi:10.1256/qj.04.03.

---

## Author Response (AR5)

**Referee 1**

We are grateful to the referee for devoting their time to our manuscript. The valuable comments and suggestions will help us to improve the paper.

We will here respond to comments made:

*The parameters of these systems are set so that all scales behave chaotically. Though it is not totally clear how robust the results are if the parameters are perturbed.*

As long as parameters are such that all scales are chaotic, we do not expect any qualitative changes with respect to the studied scenario. We tried to ensure the robustness of the results by considering two cases of "reality" (L05-2 and L05-3 systems). Furthermore, we tested as "reality" the L05-1 system with 360 variables and as "model" the L05-1 system with 180 and 90 variables. The results are consistent with the presented results. We are aware of the need to test the results on "real" systems.

*The explanation of the initial decline and subsequent growth of the rate of model error growth by the notion of ``drift'' is a nice attempt, though it is not totally clear if this is special for the L-05 systems.*

"Drift" was used by Orrell (2002) to explain the initial decline and subsequent growth of the rate of model error growth for the ECMWF system (500 hPa, Northern hemisphere for 10 d in October 1999 and total energy globally over a 15 d period in December 2000) . Therefore, the results do not appear to apply only to the L05 system. We have further confirmed the behavior resulting from "drift" on the ECMWF system data in Section 5.

*In the abstract, where is the claim "Generally, a system with model error (omitting phenomena) will not improve predictability." supported in the maintext? How general is it? This seems to be a very strong statement. If not, I suggest weaken this statement.*

We have replaced "Generally" with "In other words" (Lines 11 – 12).

*Although it maybe natural, it would be good to give a sentence of explanation about why choosing L05-2 and L05-3 as "reality"*

A full explanation of why L05-2 and L05-3 systems are selected as the reality is given in Section 2.2. In addition, we have added a sentence to the introduction on lines 128 – 129 ("The omitted scale is the small scale for the L05-2 system and the small and medium scale for the L05-3 system. "). Information on why we do not use the L05-2 system as model and L05-3 system as reality is given on lines 614-615.

*Is "this" in line 9 ``initial error growth? Perhaps good to be more specific.*

The word "product" was added to line 9.

*How about adding references to the relevant figures after "as we will see in numerical simulations" in line 270?*

Reference was added.

*Would you explain why geometric mean is used rather than the usual arithmetic mean in model error growth and drift terms in (11), (14), (15), (18) ?*

We added an explanation on line 200: "The geometric mean is chosen because of its suitability for comparison with growth governed by the largest Lyapunov exponent. For further information, see Bednar et al. (2014) or Ding and Li (2011). "

*The drift d(tau) at the beginning of line 271 is not defined yet, It does not seems to be the drift VECTOR in line 269. Please clarify.*

*d(tau)* was changed to the absolute value of drift *d(tau)*.

*Please be consistent in terminology. For example, is ``the drift D(tau)" on Page 279 the same as d(tau) in line 271? Is it the same as the "the averaged drift D" in line 283?*

We improved consistency in terminology. We related *D(t)* to eq. (18) and *d(tau)* to eq. (17).

*Perhaps include a table summarizing the heavy notation involved.*

We itemized the numbers of the equations. We hope this will help readability.

*The reference list will look better it it was itemized.*

We itemized the reference list.

*I am not totally convinced (or understand) that the notion of the "drift" introduced really explain the model error growth as claimed. It seems that, taking time-average without an absolute value is similar to looking at the original system, when the system is ergodic. Perhaps the authors can explain more on what ``explain" means other than showing another summary statistics of the system.*

The difference is that it is the summation of vectors created from the difference in time evolution of different systems (but with the same initial conditions) after one time step. The model errors at successive time steps as vectors are not strongly correlated, and that therefore accumulating their absolute values is very different from accumulating them as vectors, where the absolute values sum will grow much faster than the vector valued sum, and that this slower error growth now gives a better explanation of the deviation of the trajectories.

References:

Bednář, H., Raidl, A., and Mikšovský, J.: Initial Error Growth and Predictability of Chaotic Low-dimensional Atmospheric Model, IJAC, 11, 256–264, https://doi.org/10.1007/s11633-014-0788-3 2014.

Ding, R., Li, J.: Comparisons of two ensemble mean methods in measuring the average error growth and the predictability, Acta Meteorol Sin, 25, 395–404, https://doi.org/10.1007/s13351-011-0401-4, 2011.

Orrell, D.: Role of the metric in forecast error growth: how chaotic is the weather?, Tellus, 54, 350-362, https://doi.org/10.1034/j.1600-0870.2002.01389.x, 2002.

**Referee 2**

We are grateful to the referee for devoting his time to our manuscript. We will here respond to comments made to support the validity of the article for publishing:

*This paper tries to explain why omitting atmospheric phenomena, which contribute little to the final value, will not improve the predictability of the resulting value. However, this paper does not provide a complete theory to show this. Although this article says that a theory explaining and describing this behavior is developed, I did not find any strict mathematical theory in this article.*

The developed theory is not strictly mathematical but is based on a strictly mathematical theory describing the model error growth (Drift - Section 2.4), presented by Orrell (Orrell et al., 2001; Orrell, 2002) and on a strictly mathematical theory of classical low dimensional chaos, where one observes an exponential error growth of a tiny initial error whose exponent is given by the largest Lyapunov exponent of the system. Our extension that sees Drift produced at each time step as the error of the initial conditions is based on an experiment with Lorenz L05 systems (Appendix A) and explains and describes the model error growth in this experimental setting (Section 4). The derived results are then verified in the ECMWF systems (Section 5). Because it is not a theory in a strictly mathematical sense. We replace the term "theory" with the term "hypothesis"**.** We believe that our hypothesis Eq. (21) is as worthy of publication as other already commonly accepted experiment-based hypotheses such as Eqs. (2)-(6).

*Line 20: ..."the instability of the system with respect to initial condition errors has grown"..., the instability is not clear?*

By instability we mean the error growth rate of the initial conditions of ECMWF systems, which is expressed by the Lambda parameter from Eq. (5). The values can be seen in Figure 15a - blue curve. More details can be found in Section 5. For better understanding, we have added "(error growth rate)" to the text.

*Line 95:" .... the constant b in Eq. (5) which, irrespective of initial condition errors, will lead to a deviation of the model solution from reality..."*. *It seems that there is no a constant b in Eq. (5).*

*b* has been replaced in the text by *betha*. We thank the referee for spotting this misprint.

*Line 120: "....Including small spatiotemporal scales, i.e., improving the model's spatial and temporal resolution, therefore enhances the instability with respect to initial condition error".... the exact meaning of the instability is not clear.*

By instability we mean the error growth rate of the initial conditions. Brisch and Kantz (2019) and Zhang et al. (2019) associated initial error growth with scale-dependent error growth, where tiny errors grow much faster than larger ones. Lorenz (1969) gave a sketch of such error growth: a typical quantity to be predicted is a superposition of the dynamics on different scales. After a fast growth of the small-scale errors with saturation at these very same small scales, the large-scale errors continue to grow at a slower rate until even these saturate. We have added "(error growth rate)" to the text.

*Line 140: "....We measure the error magnitude e(t) after fixed time intervals ..". there is not any expression for e(t).*

On line 140, *e(t)* is defined as the error magnitude after fixed time intervals. The expressions for the settings are shown on lines 192, 226 and 245.

*Line 160: ".... For this scheme to be meaningful, we have to ensure that the reference trajectory is on the system's attractor and that the repetition of this scheme samples the whole attractor with correct weights (the invariant measure)....". the existence of attractors in this system is not clear.*

Lorenz L05 systems are widely accepted chaotic systems with a positive largest Lyapunov exponent (which is computed and presented). For L96 system (Lorenz, 1996) the existence of attractor has been shown, and because our system can be expected to be in the same model class, we expect the existence of a chaotic attractor.

*Line 195: There is no definition of . Line 230: There is no definition of . Line 245: There is no definition of .*

It is probably meant that Eqs. (11), (14) and (15) are not definitions from a strictly mathematical sence, so we replace the expression "is defined" by "is calculated".

References:

Brisch, J., and Kantz, H.: Power law error growth in multi-hierarchical chaotic system-a dynamical mechanism for finite prediction horizon, New J. Phys., 21, 1–7, https://doi.org/10.1088/1367-2630/ab3b4c, 2019.

Lorenz, E. N.: The predictability of a flow which possesses many scales of motion, Tellus, 21, 289–307, https://doi.org/10.1111/j.2153-3490.1969.tb00444.x, 1969.

Lorenz, E. N.: Predictability: a problem partly solved, in: Predictability of Weather and Climate, edited by: Palmer, T., and Hagedorn, R., Cambridge University Press, Cambridge, UK, 1–18, https://doi.org/10.1017/CBO9780511617652.004, 1996.

Orrell, D., Smith, L., Barkmeijer, J., and Palmer, T. N.: Model error in weather forecasting, Nonlin. Processes Geophys., 8, 357–371, https://doi.org/10.5194/npg-8-357-2001, 2001.

Orrell, D.: Role of the metric in forecast error growth: how chaotic is the weather?, Tellus, 54, 350-362, https://doi.org/10.1034/j.1600-0870.2002.01389.x, 2002.

Zhang, F., Sun, Q., Magnusson, L., Buizza, R., Lin, S. H.,Chen J. H., and Emanuel K.: What is the Predictability Limit of Multilatitude Weather, J. Atmos. Sci., 76, 1077–1091, https://doi.org/10.1175/JAS-D-18-0269.1, 2019.

Lorenz, E. N.: The predictability of a flow which possesses many scales of motion, Tellus, 21, 289–307, https://doi.org/10.1111/j.2153-3490.1969.tb00444.x, 1969.

**Referee 3 (Report 1)**

We are grateful to the referee for devoting their time to our manuscript. The valuable comments and suggestions will help us to improve the paper.

We will here respond to comments made:

The designed system is based on systems created by Lorenz (2005). The first and simplest of this type is the low-dimensional atmospheric system (L96) presented by Lorenz (1996). It is a nonlinear model, with $N$ variables connected by governing equations

$$dX_n / dt = -X_{n-2}X_{n-1} + X_{n+1}X_{n-1} - X_n + F, \tag{1}$$

$n = 1, \ldots, N$. $X_{n-2}$, $X_{n-1}$, $X_n$, $X_{n+1}$ are unspecified (i.e., unrelated to actual physical variables) scalar meteorological quantities (units), $F$ is a constant representing external forcing, and $t$ is time. The index is cyclic so that $X_{n-N} = X_{n+N} = X_n$ and variables can be viewed as existing around a latitude circle. Nonlinear terms of Eq. (1) simulate advection. Linear terms represent mechanical and thermal dissipation. The model quantitatively, to a certain extent, describes weather systems, but, unlike the well-known Lorenz model of atmospheric convection (Lorenz, 1963), it cannot be derived from any atmospheric dynamic equations. **The motivation was to formulate the simplest possible set of dissipative chaotically behaving differential equations that share some properties with the "real" atmosphere. One of the model's properties is to have 5 to 7 main highs and lows that correspond to planetary waves (Rossby waves) and several smaller waves corresponding to synoptic-scale waves. For Eq. (1), this is only valid for** $N = 30$. Lorenz (2005), therefore, introduced spatial continuity modification (L05). Eq. (1) is then rewritten to the form:

$$\frac{dX_n}{dt} = [X, X]_{L,n} - X_n + F, \tag{2}$$

where

$$[X, X]_{L,n} = \sum_{j=-J}^{J}{}' \sum_{i=-J}^{J}{}' \left( -X_{n-2L-i}X_{n-L-j} + X_{n-L+j-i}X_{n+L+j} \right) / L^2$$

If $L$ is even, $\sum'$ denotes a modified summation, in which the first and last terms are to be divided by 2. If $L$ is odd, $\sum'$ denotes an ordinary summation. Generally, $L$ is much smaller than $N$ and $J = L/2$ if $K$ is even and $J = (L-1)/2$ if $L$ is odd. **To keep a desirable number of main highs and lows, Lorenz (2005) suggested a ratio** $N/L = 30$ **and** $F = 15$. **The choice of parameters** $F$, **and the setting of time unit = 5 days, is also made to obtain a similar value of the largest Lyapunov exponent as the ECMWF forecasting system** (Lorenz, 2005).

A two-level (scales) system (L96-2) was introduced by Lorenz (1996) by coupling two such systems, each of which, aside from the coupling, obeys a suitably scaled variant of Eq. (1). There are $N$ variables $X_n$ plus $J \cdot N$ variables $Y_{j,n}$ defined for $n = 1, \ldots, N$ and $j = 1, \ldots, J$. Governing equations are:

$$dX_n / dt = -X_{n-2}X_{n-1} + X_{n+1}X_{n-1} - X_n + F - (c/b)\sum_{j=1}^{J} Y_{j,n}, \tag{3}$$

$$dY_{j,n} / dt = -cbY_{j-2,n}Y_{j-1,n} + cbY_{j+1,n}Y_{j-1,n} - cY_{j,n} + (c/b)X_n, \tag{4}$$

where $c$ sets the rapidness of small scale compared to large scale, $b$ sets the small scale amplitude size compared to large scale. $Y_{j,n-N} = Y_{j,n+N} = Y_{j,n}$ while $Y_{j+J,n} = Y_{j,n+1}$ and $Y_{j-J,n} = Y_{j,n-1}$. $X_n$ represent the values of some quantity in $N$ sectors of latitude circle, while the variables $Y_{j,n}$ ( $Y_{1,1}, Y_{2,1}, \ldots, Y_{J,1}, Y_{1,2}, Y_{2,2}, \ldots, Y_{J,2}, Y_{3,1}, \ldots$ ) can represent some other quantity in $JN$ sectors.

A two-level (scales) system introduced by Lorenz (2005) is:

$$dX_n / dt = [X,X]_{L,n} - X_n - cY_n + F, \tag{5}$$

$$dY_n / dt = b^2 [Y,Y]_{1,n} - bY_n + cX_n. \tag{6}$$

Eq. (6) is analogue to Eq. (1) (if we substitute $F$ for $X_n$), and Eq. (5) is analogue to Eq. (2) (aside from the coupling where $c$ is the coupling coefficient, and that $Y_n$ fluctuates $b$ times as rapidly, and their amplitude is reduced by the factor $b$ ).

*(A) Different two-scale models in Lorenz (1996) and Lorenz (2005). Will it be feasible for providing a diagram for illustrating the grid system within the 2005 two-scale model?*

Figure RR1 shows the similarity of the 1996 (Eqs. (3) and (4)) and 2005 (Eqs. (5) and (6)) two-scale systems in the attempt to maintain 5 to 7 main highs and lows and several smaller waves for large scales $X_n$. While for the 1996 two-scale system, this is ensured by a number of $N$ large scale variables $X_n$ close to 30 (and a number of $JN$ variables for the small scales), for the 2005 system, it is ensured by linking the $X_n$ variables as described in Eq. (2) (with the same number of small scale variables, however, determined from Eq. (1), Figure RR2). The 2005 two-scale system thus produces a smoother and more realistic evolution of the large-scale variable while maintaining properties similar to the 1996 system.

The systems used in this manuscript, which are described in Appendix A (of the manuscript), address one more condition that brings them closer to real systems. This condition is the fact that the large scale and small scale features in Eqs. (3) – (6) are represented by separate sets of variables instead of appearing as superimposed features of a single set. To satisfy this condition, the coupling of one small-scale variable and one large-scale variable is more realistic than the coupling that is present in the 1996 system (Eqs. (3) and (4)).

[Figure]

Figure RR1: Comparison of longitudinal profiles at one time of two-scale Lorenz systems (a) from 1996 (Eqs. (3) and (4)) and (b) from 2005 (Eqs. (5) and (6)).

[Figure]

Figure RR2: Comparison of schematic illustrations of two-scale Lorenz systems (a) from 1996 (Eqs. (3) and (4)) (taken from Figure R2 of the referee report) and (b) from 2005, where the inner wave curve represents the large-scale variables described by Eq. (5), which produce 5-7 main waves, and where the outer curve represents the small-scale variables described by Eq. (6), which are not limited by the number of waves. In contrast to (a), one large scale variable is coupled to one small scale variable.

*(B) Dependence of findings on temporal spacing (i.e., $\Delta t$) and "spatial" spacing (e.g., the number of sectors, N).  It would be ideal for additional tests with a smaller $\Delta t = 10^{-5}$ (or $\Delta t = 10^{-4}$). Additionally, the choice of N and L should be explored since N = 960 and L = 32 were used in Lorenz (2005).*

The choice of the variable $N = 360$ was made because the value of the largest Lyapunov exponent $\lambda^{L05}$ of the system described by Eq. (2) ($F = 15$, time unit = 5 days) does not change for $N = 360$ and $N = 960$ (Table RR1) and therefore we chose the lower of the two values for computational efficiency.

| $N$ | $\lambda^{L05}$ |
|-----|-----------------|
| 30 | 0.70 |
| 60 | 0.29 |
| 90 | 0.35 |
| 120 | 0.32 |
| 150 | 0.33 |
| 360 | 0.33 |
| 960 | 0.33 |

Table RR1: Values of the largest Lyapunov exponent $\lambda^{L05}$ for selected numbers of variables $N$ in the 2005 Lorenz system (Eq. (2), $F = 15$, time unit = 5 days).

Figure RR3 compares the time evolution of the average value of the variables for the 2005 Lorenz system (Eq. (2)) with time step $\Delta t$=1/240 and $\Delta t$=1/2400. It can be seen that the values are similar. Given this, we use the larger time step dt=1/240 for faster computations.

[Figure]

Figure RR3: Comparison of the time evolution of the mean value of the variables ($N$ = 360) for the 2005 Lorenz system (Eq. (2)) based on the same initial conditions with time step $\Delta t$=1/240 (red dashed curve) and $\Delta t$=1/2400 (black dotted curve).

*(C) Impact of model's configuration and complexity on critical points (equilibrium points).*

*Based on the linearization theorem, critical points of the Lorenz systems could roughly indicate the local behavior of the solutions. As a result, initial error growth should display a dependence on the equilibrium state. Please consider identifying the appearance of the critical points and perform stability analysis using the Jacobian matrix of the linearized system at each of the critical points.*

While for analytical studies the instability of fixed points (critical points) is certainly of high interest, we are interested in the typical error growth and therefore focus on the Lyapunov exponent on the chaotic attractor. Since the phase space is so high dimensional, we are not even sure that unstable fixed points are embedded in the chaotic attractor or whether they are outside, as they are in the Lorenz 1963 low dimensional model. We therefore calculate the maximal LE numerically in the following way: a reference trajectory (considered the "truth" or verification) and a trajectory which is the numerical solution of the systems with a given error, are repeatedly generated. For this scheme to be meaningful, we have to ensure that the reference trajectory is on the system's attractor and that the repetition of this scheme samples the whole attractor with correct weights (the invariant measure). We solve this issue in the following way: We first integrate the system over ten years (175200 steps), starting from arbitrary initial conditions, and assume that after discarding this transient, the trajectory is on the attractor. We continue to integrate this single trajectory and consider segments of it as reference trajectories for error growth, i.e., the many reference trajectories are simply segments of one very long trajectory, which ensures not only that all these segments are

located on the attractor but that in addition, they sample the attractor according to the invariant measure.

*Larger F may produce a larger eigenvalue (a larger real part of the eigenvalue), suggesting a larger growth rate. Based on the following preliminary analysis of the one- and two-scale models with the same value of the forcing parameter F, the effective forcing parameter for the two-scale model is smaller, yielding a smaller leading eigenvalue (i.e., a smaller real part of the eigenvalue). This is consistent with the finding that Figures 5 and 6 display larger growth rates ( λ) within the one-scale system (e.g., L05-1) than the two-scale system (e.g., L05-2).*

Figure RR4 compares the error growth rates of the L05-1 (Eq. (A1) in manuscript), L05-2 (Eq. (A8) in manuscript), and L05-3 (Eq. (A9) in manuscript) systems. In contrast to the reviewer's findings, the figure shows the smallest growth rate for the L05-1 system and the largest for the L05-3 system. We confirm that the effective forcing for slow variables is weaker, indicating a smaller growth rate within the two-scale model, as compared to the one-scale model. However, it should be noted that in Figure RR4 the values of the single-scale system (L05-1) are not compared with the large-scale values of the multi-scale systems (L05-2 and L05-3), but are compared with the total values of the L05-2 and L05-3 systems, where the large-scale and small-scale features are appearing as superimposed features of a single set.

[Figure]

Figure RR4: Initial error growth tendency (rate) $dE/dt$ as a function of the error magnitude $E$ for L05-1 system (Black, Eq. (A1) in manuscript), for L05-2 system (Red, Eq. (A8) in manuscript), and for L05-3 system (Blue, Eq. (A9) in manuscript).

A justification for the use of the L05-2 (Eq. (A8) in manuscript) and L05-3 (Eq. (A9) in manuscript) systems as the "reality" and the L05-1 system as the "model." is presented in the manuscript (Lines 210-220):

"This approach is justified by the fact that the L05-2 and L05-3 systems can be viewed as a variant of the L05-1 system:

$$dX_{tot,n} / dt = \left[ X_1, X_1 \right]_{L,n} - X_{1,n} + \tilde{F}_n(t), \tag{12}$$

where $\tilde{F}_n(t) = b^2 [X_2, X_2]_{1,n} + c [X_2, X_1]_{1,n} - bX_{2,n} + F$ for the L05-2 system and

$\tilde{F}_n(t) = b_1^2 [X_2, X_2]_{1,n} + b_2^2 [X_3, X_3]_{1,n} + c_1 [X_2, X_1]_{1,n} + c_2 [X_3, X_2]_{1,n} - b_1 X_{2,n} - b_2 X_{3,n} + F$ for the L05-3 system
are treated as a forcing, which varies in a complicated manner with time. We parameterize these small-scale
phenomena contained in $\tilde{F}_n(t)$ by the average value of these phenomena, which is close to zero, and therefore
we can write:

$$\langle \tilde{F}_n(t) \rangle \approx F = 15, \tag{13}$$

where $\langle \ldots \rangle$ represents the mean calculated over a long orbit on the L05-2 and L05-3 systems attractors."

*Please provide justifications for the choice of b1 = 10 for the two-scale system but b1 = 1 for
the three-scale system. Additionally, within the three-scale system, are nonlinear terms (e.g., c1
and c2 in Eq. A9) applied for coupling the "sub-systems" for the small- and medium-scale
variables with the large-scale system? Please comment on the impact of c1 and c2 on system's
stability.*

The parameters of any multi-level Lorenz's system (L96-2, L05-2, L05-3) should be set so
that all levels behave chaotically (the largest Lyapunov exponent of each level is positive) and
that all levels have a significant difference in amplitudes and fluctuation rates. For the L-96
system (Eq. (1)), the chaotic behavior is determined by the value of $F$, and the number of
variables $N$. Lorenz (2005) states that as long as $N \geq 12$ chaos is found when $F > 5$ (for
$N = 4$ it is when $F > 12$ and for $N > 6$ when $F > 8$). In cases such as the L96-2 system
(Eqs. (3) and (4)), where the forcing $F$ acts only on the largest scale, the chaotic behavior of
smaller scales is created by coupling. The size of the coupling is cascaded from the largest
scale to the smaller ones. Because the values of the largest scale variables are determined by
the forcing $F$, the $F$ value indirectly affects the smaller scales' chaotic behavior and must be
chosen large enough to ensure chaotic behavior through coupling for all scales (levels). For
the L05-2 system (Eq. (A8)), variables are superposed features of a single set calculated by
Eqs (A4) and (A5). In addition to those mentioned above, this procedure affects the chaotic
behavior, amplitude, and fluctuation rate of the levels, and the choice of $I$ between 10 and 20
may be optimal (Lorenz, 2005). In order to maintain the required properties of the two scales
L05-2 system, Lorenz (2005) chose $N = 960$, $L = 32$, $I = 12$, $F = 15$, $b = 10$, and $c = 2.5$ (**note
that for L05-2 and L05-3 systems it is not possible to directly determine the amplitude
and fluctuation rate of smaller scales using spatiotemporal scaling factors $b$, because
these values are mainly determined by the procedure for expressing variables and the
length of the intervals** $[-I, I]$).

For the L05-3 system (Eqs. (A9) – (A12)), it is necessary to specify eight parameters. We tested
that the values of coupling coefficients $c_1$ and $c_2$ do not affect the L05-3 system compared to
the values of other parameters, and therefore for simplification $c_1 = 1$ and $c_2 = 1$. The parameter
$F = 15$ is set the same as for other L05 systems. For the medium scale amplitude to be
approximately ten times smaller than the large scale amplitude and the small scale amplitude to
be approximately ten times smaller than the medium scale amplitude and for the scales to have
different oscillation rates, the spatiotemporal scale factors are chosen $b_1 = 1$ and $b_2 = 10$ and

interval lengths $I_1 = 20$, and $I_2 = 10$. $N = 360$ turned out to be most suitable for the chaotic behavior of all three levels (found experimentally).

*(D) Separations of initial and model errors*

We fully agree with the comment. We simulate the initial error growth in the same systems (perfect model assumption), and the model error growth with zero initial error (perfect initial conditions assumption). Combination of both is studied in section 3.3 of the manuscript.

*(E) Validity of error saturation for periodic attractors and coexisting attractors. Have you observed periodic solutions? Can you comment on the validity of error saturation for periodic solutions? Have you observed multistability in your ensemble runs?*

In our research, we focused only on the average value of error growth (over variables and number of runs). We set all the scales through the parameters of the Lorenz systems to behave chaotically (details can be found in Bednar and Kantz (2022)) and the evolution of the average error growth did not show signs of periodic solution or multistability.

**Specific Comments:**

*(1) Please check consistency in the capitalization of the initial letters of words within a title.*

We checked and fixed it. Thank you for pointing this out. (Lines 1-2).

*(2) Lines 45-50, the application of the Lyapunov exponent (LE) is not accurate. A global LE represents a long-term average of "local" growth rates (determined by the separations of two nearby trajectories). Initial separations should remain small. Local growth rates may vary with time. As a result, Eq. (1) with a constant growth rate is valid only for a finite time interval. During different time internals, different growth rates may appear. Note that in addition to one positive LE, solution's boundedness is another important feature that defines a chaotic system.*

We have added information about boundedness and validity for a finite time interval. (Lines 47-48)

*(3) Lines 45-55, please consider referring to the growth rates in Eqs. (1) and (2) as the exponential growth rate (with a J-shaped curve) and logistic growth rate (with a S-shaped curve), respectively.*

We changed the description of Eqs. (1) and (2). (Lines 816-819, 828-830, 841-843, 855-856, 870-871).

*(4) Line 80, the term "error growth laws" should be rephrased since they are not necessarily physical laws.*

We replaced the term law with the term hypothesis. (Lines 81, 307)

*(5) Lines 122, statements are not accurate. Unless additional forcing terms are introduced, improving model's spatial or temporal resolution does not necessarily enhance instability. (Please think of a convergent Taylor series.)*

We added to the introduction: "Buizza (2010), Magnusson and Kallen (2013) or Jacobson (2001) show that improving the model's spatial and temporal resolution will improve the

ability to predict, especially for short forecast range (Buizza, 2010). However, the cited studies work with models that do not model small spatiotemporal phenomena (they are parameterized) and whose initial condition error magnitude is larger than the magnitude of these phenomena. We have verified the fact that the high resolution model (that models small scales) is less stable than the low resolution model (that doesn't model small scales) against initial condition errors (Bednar and Kantz, 2022; Budanur and Kantz, 2022), and that therefore the issue of omitting small scales has another facet. Our new approach models and omits small spatiotemporal scales using…" (Lines 129-135)

*(6) Lines 128-130: it is wired that the two-scale system contains large- and small-scale systems while the three-scale system adds a medium scale, in addition to large- and small-scale flows. Any justifications?*

It would be more natural to take the L05-2 and L05-1 systems as the model and the L05-3 system as the reality. ). A variant where the L05-2 system was used as the model and the L05-3 system as the "reality" was also tested. The resulting model error growth is approximately identical to the previous variant (L05-1 system as the model and L05-3 system as the "reality"). That's why we chose the settings we present. Further, it would be more natural for the L05-2 system to have a small scale comparable to the medium scale of the L05-3 system. However, our intention was to be close to the L05-2 system presented by Lorenz (2005), whose small scale is equivalent to the small scale of our L05-3 system.

*(7) Lines 160-165, have you observed coexisting attractors (e.g., more than one attractors) in your ensemble runs? (e.g., see multistability in Van Kekem and Sterk 2018a,b, 2019; Pelzer et al., 2020).*

In our research, we focused only on the average value of error growth (over variables and number of runs) and we did not observe signs of multistability.

*(8) Line 170, does the statement "errors might even shrink in short times" indicates the existence of a stable manifold?*

Yes, the Lorenz L05-systems possess rather high dimensional stable manifolds, along which trajectories are attracted towards the attractor. Calculation of the Lyapunov-dimension done by us for L05-2 show this very clearly, the attractor dimension is much smaller than the phase space dimension, where the attractor is the unstable manifold. But the statement on line 170 does not indicate the existence of a stable manifold but the fact that initial perturbations might not point into the locally most unstable direction.

*(9) Lines 194, while N=360 was used in this study, N=960 was appied in Lorenz (2005).*

Thank you for pointing this out. The problem is already discussed in comment (B).

*(10) line 186, how many time steps for the transfer of error to the small-scale variables?*

The error would immediately (one time step) propagate into the small-scale variables.

*(11) Section 3.1, please confirm whether the leading LE in the L05-1 system is larger (smaller) than that in the L05-2 (L05-03) system.*

Figure RR4 compares the error growth rates of the L05-1 (Eq. (A1) in manuscript), L05-2 (Eq. (A8) in manuscript) and L05-3 (Eq. (A9) in manuscript) systems. The figure shows the smallest

growth rate for the L05-1 system and the largest for the L05-3 system (therefore also for LE). It should be noted that for the L05-2 and L05-3 systems, the error growth rate is scale dependent.

*(12) Line 382-394: The key point that higher resolution model produces better predictability is acceptable. However, it is not clear whether Figure 10 is sufficient to support this point. Please see details in the last specific comment below.*

Please see the discussion at comment (17)

*(13) Line 656: The statement "Based on the fact that scale-dependent error growth implies an intrinsic predictability limit" is not accurate. A finite growth rate may indicate a limit for practical predictability. By comparison, a finite intrinsic predictability is established by the feature of chaos (e.g., sensitive dependence on initial condition, SDIC; e.g., Shen, Pielke Sr., and Zeng, 2023)*

Our statement really refers to the finite intrinsic predictability that is established by the features of chaos. The statement is based on Brisch and Kantz (2019), Bednar and Kantz (2022), and Budanur and Kantz (2022).

*(14) Lines 612 - 623, discussions are duplicated; they are the same as those in Lines 600-611.*

We deleted the duplicated part. Thank you for pointing this out.

*(15) Line 715, the parameter "K" should be replaced by "L".*

We replaced $K$ by $L$. Thank you for pointing this out.

*(16) Line 716, Lorenz (2005) did not explicitly suggest the ratio of N/L = 30 nor provide justification for the choice of N = 960 and L = 32.*

We assume the requirement for a model to have 5 to 7 main highs and lows that correspond to planetary waves (Rossby waves) and several smaller waves corresponding to synoptic-scale waves, and we follow the text of Lorenz (2005) on the pages 1579 (Fig. RR5) and 1580 (Fig RR6).

Figure 4a shows typical profiles produced by Eq. (8) when $N = 240$ and $F = 10$ for selected values of $K$. When $K = 2$, there are nearly as many waves as if Eq. (1) had been retained. Increasing $K$ to 4 decreases the number, but there are still too many compared with Fig. 1a. When $K = 8$, one evidently succeeds in producing an acceptable number of major waves, although weaker smaller-amplitude waves are superposed. In drawing the curve I have, as usual, connected the successive values of $X_n$ with straight-line segments, but these are hard to detect. Any other reasonable interpolation procedure would have produced an indistinguishable curve. Increasing $K$ to 16, 32, or 64 lengthens the waves still more, and, evidently, one can produce any wavenumber desired by choosing $K$ judiciously.

Since the ratio $N/K$ is 30 in the third profile, whose dominant wavenumber agrees most closely with Fig. 1a, where $N = 30$, there is a suggestion that the appearance of a profile may depend largely upon $N/K$. Figure 4b is constructed with $F = 10$ and $N/K = 30$ in each profile, and with $N$ successively doubling from 30 in the leading profile to 960 in the final one. The conjecture seems to be well supported; the profiles in Fig. 4b show little resemblance to any profile in Fig. 4a except the third one.

With $N = 960$ and again with $K = 32 = N/30$ and $F = 10$, Fig. 5a has been constructed in the manner of Fig. 1a; it shows profiles produced by Eq. (8) at 6-h intervals for five days. Again, at least for the five days, the major crests and troughs retain their identities, while minor ones come and go. One can conclude that Model II is ready for some applications for which Model I would have been inadequate.

For Model I the doubling time for small errors, as seen in Fig. 2a, depends strongly upon $F$, but is nearly independent of $N$ if $N$ is not too small. For Model II, with $K > 1$, it also proves to depend strongly upon $F$ while being nearly independent of $N$ and $K$ if $N/K$ is not too small, but, for a given value of $F$, it is much smaller when $K > 1$ than when $K = 1$. Thus, for the values used in Fig. 5a, the doubling time is about four days—considerably longer than expected in the atmosphere. It can be restored to a more nearly atmospheric value by increasing $F$.

Figure 5b is constructed like Fig. 5a, again with $N =$

Figure RR5: Page 1579 in Lorenz (2005).

[Figure]

FIG. 4. (a) Profiles of X produced by Eq. (8) with N = 240, F = 10, and values of K indicated by numbers at left. Scale at bottom is gridpoint number. (b) Profiles of X produced by Eq. (8) with values of N indicated by numbers at left and with K = N/30 and F = 10. Scale at bottom is gridpoint number for bottom curve.

960 and K = 32, but with F = 15. There is still a suggestion of six or seven longer waves, but the shorter waves are more in evidence. Note that, with N so large, even these shorter waves are 30 or more grid intervals long—the point-to-point variations are very smooth. The doubling time has been reduced to about two days.

Apparently, in trying to make the curves produced by Model II look like reasonable spatial interpolations of the kind of curve produced by Model I, one must choose between too long a doubling time (smaller F) or unanticipated shorter waves (larger F). The value F = 15 is a compromise.

FIG. 5. (a) Profiles of X produced by Eq. (8) with N = 240, K = 8, and F = 10, at 12-h intervals for 5 days. Scale at bottom is gridpoint number. Numbers at left indicate chronological order of profiles. (b) Same as (a) but with F = 15.

Figure RR6: Page 1580 in Lorenz (2005).

*(17) page 40, line 870-875, Figure 10. Figure's title and captions are confusing. Since L05-02 and L05-03 systems were used to provide the "ground true" (or reference) for computing errors, these errors do not represent the errors of the L05-02 and L05-03 systems, respectively, the growth of initial errors within the L05-02 or L05-03 system does contribute to the growth of differences of the solutions between the L05-1 and L05-02 (or L05-03) systems.*

 *For a comparison in Figures 5-7, let's simply choose λ+, = 0.33, 0.29, and 0.46 for the L05-1, L05-2, and L05-3 systems, respectively. The comparison of the above selected growth rates produces a consistent finding that larger differences (in error growths) are reported in Figure 10b than in Figure 10a. However, on the other hand, considering differences between the L05-02 and L05-03 systems, the differences may produce the largest growth rates as compared to those in Figure 10a and Figure 10b.*

The question under investigation in this paper is whether omitting small scale atmospheric phenomena, which contribute little to the final value, will improve the predictability of the resulting value. In other words, how does the average forecast error growth change in a model where small-scale phenomena are omitted but where model errors are therefore introduced, compared to a model where all phenomena are present but the average forecast error growth is scale-dependent. So if we use L05-02 and L05-03 systems to provide the "ground true" (or

reference) then, when searching for an answer to the research question, it is reasonable to use the results presented in Figure 10.

Figures 5-7 show that, the L05-1 system is a classical chaotic system with the largest Lyapunov exponent of about $\lambda \approx 0.33$ 1/ day. The data of the L05-2 and L05-3 are best approximated by the power law . For a power law: $\lambda_p(E) := \dfrac{d\ln(E)}{dt} = \dfrac{\dot{E}}{E} = aE^{-\sigma}$, with an exponent $\sigma$ and a coefficient $a > 0$, the error growth rate $\lambda(E) \approx \dfrac{1}{\Delta t}\ln(E(t+\Delta t)/E(t))$ is expected to be a function of the error magnitude $E$, and is not constant as for classical chaotic systems. For exponential growth (classical chaos) $E_{exp}(t) = E_0 e^{\lambda_{exp}t}$ and for an initial error $E_0$ going to zero, the time $t_{lim}$ at which the error reaches a limiting value $E_{lim}$, goes to infinity: $t_{lim} = \dfrac{\ln E_{lim} - \ln E_0}{\lambda_{exp}} \to \infty$ for $E_0 \to 0$.

However, a strict predictability limit $t_{lim}$ exists for scale-dependent error growth even when the initial error $E_0$ vanishes. For a description by a power law $dE_p$, the predictability limit $t_{lim}$ is: $t = \left(E^b(t) - E_0^b\right)/\left(a\cdot b\right) \to t_{lim} = E_{lim}^b /(a\cdot b) < \infty$ for $E_0 \to 0$.

It is true that if we show the growth of the model and initial error in Figure 10, this is the initial error of the L05-1 system, but this is consistent with the question under investigation. At the same time, Figure 10 compares the strictly model error growth (no initial error) with the strictly initial error growth (L05-2, L05-3 systems), where the initial error is limiting towards zero and is then a strict predictability limit.

References:

Bednář, H., and Kantz, H.: Prediction error growth in a more realistic atmospheric toy model with three spatiotemporal scales, Geosci. Model Dev., 15, 4147–4161, https://doi.org/10.5194/gmd-15-4147-2022, 2022.

Brisch, J., and Kantz, H.: Power law error growth in multi-hierarchical chaotic system-a dynamical mechanism for finite prediction horizon, New J. Phys., 21, 1–7, https://doi.org/10.1088/1367-2630/ab3b4c, 2019.

Budanur, N. R., Kantz, H.: Scale-dependent error growth in Navier-Stokes simulations, Phys. Rev. E, 106, 1–7, https://doi.org/10.1103/PhysRevE.106.045102, 2022.

Buizza, R.: Horizontal resolution impact on short- and long-range forecast error, Quarterly Journal of the Royal Meteorological Society, 136, 1020–1035, https://doi.org/10.1002/qj.613, 2010.

Jacobson, M. Z.: GATOR-GCMM: 2. A study of day- and nighttime ozone layers aloft, ozone in national parks, and weather during the SARMAP field campaign, J. Geophys. Res., 106, 5403-5420, https://doi.org/10.1029/2000JD900559, 2001.

Lorenz, E. N.: Predictability: a problem partly solved, in: Predictability of Weather and Climate, edited by: Palmer, T., and Hagedorn, R., Cambridge University Press, Cambridge, UK, 1–18, https://doi.org/10.1017/CBO9780511617652.004, 1996.

Lorenz, E. N.: Designing chaotic models, J. Atmos. Sci., 62, 1574–1587, https://doi.org/10.1175/JAS3430.1, 2005.

Magnusson, L., and Kallen, E.: Factors Influencing Skill Improvements in the ECMWF Forecasting System, Mon. Wea. Rev., 141, 3142–3153, https://doi.org/10.1175/MWR-D-12-00318.1, 2013.

**Referee 4 (Report 2)**

We are grateful to the referee for devoting their time to our manuscript. The valuable comments and suggestions will help us to improve the paper.

We will here respond to comments made:

*The abstract states, "This system shows that omitting small spatiotemporal scales will reduce predictability more than modeling it. In other words, a system with model error (omitting phenomena) will not improve predictability." However, this conclusion is not new. The abstract of Jacobson (2001), for example, states, "Statistics from outer nested domains indicated that the coarser the grid spacing, the greater the underprediction of ozone." Table 2 of the same paper quantifies the impact of grid spacing on model accuracy against data for 25 parameters, including meteorological (wind speed/direction, temperature, pressure, RH), and air quality parameters, in each of four nested domains. The paper concludes (Section 6), "For many parameters…accuracy improved from the coarsest to finest regional domains." Please include a discussion of Jacobson (2001) in your Introduction and indicate whether any other reference you are aware of have also shown the conclusion you are making (that omitting spatiotemporal scales reduces model predictability against data) through a comparison of model results at different scales with data.*

We added to the abstract: "that significantly affect the ability to predict" (Line 11)

We added to the introduction: "Buizza (2010), Magnusson and Kallen (2013) or Jacobson (2001) show that improving the model's spatial and temporal resolution will improve the ability to predict, especially for short forecast range (Buizza, 2010). However, the cited studies work with models that do not model small spatiotemporal phenomena (they are parameterized) and whose initial condition error magnitude is larger than the magnitude of these phenomena. We have verified the fact that the high resolution model (that models small scales) is less stable than the low resolution model (that doesn't model small scales) against initial condition errors (Bednar and Kantz, 2022; Budanur and Kantz, 2022), and that therefore the issue of omitting small scales has another facet. Our new approach models and omits small spatiotemporal scales using…" (Lines 129-135)

*Abstract. Also, what is missing in the abstract is a summary of results relative to model resolution. How much does improving the resolution, say by a factor of 2 in each the north-south and east-west direction, reduce the error over a specified period of time?*

A comparison of how much an improvement in resolution reduce the error over a specified period of time is made in Section 3.4 (lines 367-422). This full comparison is too extensive for the requirements of the abstract, so we have restricted information in the abstract to: "This system shows that omitting small spatiotemporal scales that significantly affect the ability to predict will reduce predictability more than modeling it. In other words, a system with model error (omitting phenomena) will not improve predictability." (lines 11-13).

We are also more interested in the general qualitative perspective (whether omitting small scale phenomena that contribute little to the forecasted product but significantly affect the ability to predict this product will improve the predictability of the resulting value) than in specific quantitative values, because these depend on the parameters of the particular system and its setting.

*The authors use the ECMWF model. Please clarify what parameters this model conserves. Does it conserve mass, momentum, kinetic energy, vorticity, enstrophy, and/or potential enstrophy? Do you hypothesize that the non-conservation of some of these properties may affect the results. Can you hypothesize whether results using the ECMWF would give different results from those of a different model, such as the UCLA GCM, which conserves different properties (mass, kinetic energy, vorticity, and potential enstrophy in that case)?*

We used 500 hPa geopotential height values of ECMWF systems calculated as 25 annual averages over the Northern Hemisphere (20–90 ) obtained daily from 1 January 1987 to 31 December 2011. Data was obtained from Magnusson (2013).

Magnusson and Kallen (2013) summarized the development of the ECMWF system during that period: "Since the operational start in 1979, the ECMWF forecast model and the data assimilation system have been continuously developed. Among the important upgrades is the introduction of four-dimensional variational data assimilation (4D-Var) at the end of 1997 and subsequent changes in the use of data in the assimilation were undertaken (Simmons and Hollingsworth 2002). One important change here was the upgrade of the usage of raw microwave radiances from the Television Infrared Observation Satellite (TIROS) Operational Vertical Sounder (TOVS) and Advanced TIROS TOVS (ATOVS) satellite-borne instruments in the year 2000. A major change in the model physics took place in 2007 when changes to the convection scheme and the vertical diffusion were introduced (Bechtold et al. 2008). A comprehensive description of the changes between 2005 and 2008 is given in Jung et al. (2010)." Unfortunately, we did not find in the cited articles what parameters the systems conserve (we suppose that it is based on the primitive equations and hence conserves mass and momentum, but certainly there is some damping (modeling viscosity), so that energy might not be conserved).

Regarding the question of whether non-conservation of some of these properties may affect the results. Drift described in Section 2.4 is a general description of how to characterize a model error and is therefore universal. The extension described in Section 4.1 describes the time evolution of the drift generated at each time step using exponential growth. The universality of this hypothesis has to be confirmed.

*Along those lines, in general, do you think the conclusions drawn with this model apply to other models?*

We examined whether omitting atmospheric phenomena, which contribute little to the final value, will improve the predictability of the resulting value. For this, we used the L05 systems defined by Lorenz (2005) and Bednar and Kantz (2022) and the ECMWF systems with data from Magnusson (2013). We have shown that omitting atmospheric phenomena, which contribute little to the final value, **will not** improve the predictability of the resulting value. The average prediction error grows faster in a model where small-scale phenomena are omitted, but the model error is therefore created, compared to a model where all phenomena are present, but the average forecast error growth is scale-dependent. We think that our conclusions are general and can by applied to other models.

References:

Bechtold, P., M. Köhler, T. Jung, F. Doblas-Reyes, M. Leutbecher, M. J. Rodwell, F. Vitart, and G. Balsamo: Advances in simulating atmospheric variability with the ECMWF model:

From synoptic to decadal time-scales, Quarterly Journal of the Royal Meteorological Society, 134, 1337–1351, https://doi.org/10.1002/qj.289, 2008.

Bednář, H., and Kantz, H.: Prediction error growth in a more realistic atmospheric toy model with three spatiotemporal scales, Geosci. Model Dev., 15, 4147–4161, https://doi.org/10.5194/gmd-15-4147-2022, 2022.

Budanur, N. R., Kantz, H.: Scale-dependent error growth in Navier-Stokes simulations, Phys. Rev. E, 106, 1–7, https://doi.org/10.1103/PhysRevE.106.045102, 2022.

Buizza, R.: Horizontal resolution impact on short- and long-range forecast error, Quarterly Journal of the Royal Meteorological Society, 136, 1020–1035, https://doi.org/10.1002/qj.613, 2010.

Jacobson, M. Z.: GATOR-GCMM: 2. A study of day- and nighttime ozone layers aloft, ozone in national parks, and weather during the SARMAP field campaign, J. Geophys. Res., 106, 5403-5420, https://doi.org/10.1029/2000JD900559, 2001.

Jung, T., and Coauthors: The ECMWF model climate: Recent progress through improved physical parametrizations, Quarterly Journal of the Royal Meteorological Society, 136, 1145–1160, https://doi.org/10.1002/qj.634, 2010.

Magnusson, L.: Factors Influencing Skill Improvements in the ECMWF Forecasting System, available from personal repository: linus.magnusson@ecmwf.int [data set], 2013.

Magnusson, L., and Kallen, E.: Factors Influencing Skill Improvements in the ECMWF Forecasting System, Mon. Wea. Rev., 141, 3142–3153, https://doi.org/10.1175/MWR-D-12-00318.1, 2013.

Simmons, A., and A. Hollingsworth: Some aspects of the improvement in skill of numerical weather prediction, Quarterly Journal of the Royal Meteorological Society, 128, 647–677, https://doi.org/10.1256/003590002321042135, 2002.

**Referee 3 (Report 1 04 Mar 2024)**

**We are grateful to the referee for devoting time to our manuscript.
We will here respond to comments made:**

**(Comment 1)** The reviewer acknowledges the authors' efforts in addressing the review comments. However, the current format of their responses lacks point-by-point clarification, posing challenges in evaluating their responses. Therefore, it is recommended to reformat the responses for clarity……………………………………………………….…23

**(Comment 2)** Furthermore, considering the authors' assertion that the proposed 05 system simulates "5 to 7 main highs and lows that correspond to planetary waves (Rossby wave)," it would be advantageous to discuss whether the proposed system, without the Coriolis force, could replicate key features of the Rossby wave, including phase speeds. Historically, experiments such as dishpan experiments aimed to "simulate" weather features, yielding diverse outcomes like chaotic solutions and vacillation (e.g., limit cycle)…………………………………………………………………….……38

**(Comment 3)** This study extends from the authors' previous research. The reviewer acknowledges the related efforts. However, after examining their earlier studies, the reviewer proposes the following: 1. Document and report the calculation of Lyapunov exponents (LEs) within the proposed 05 system. For instance, employing the 1963 model with common parameters, the largest LE (LE1) is 0.906, as exemplified in the link provided (https://sprott.physics.wisc.edu/chaos/lorenzle.htm). This task holds significant importance. 2. Develop the error growth model, e.g., $dE/dt = \sigma E (1 - E/E_s)$, and furnish a mathematical expression for sigma and LE1 of the proposed system. It should be noted that the long-time average of $(1/E \, dE/dt)$ is not precisely equal to sigma………………………..40

**(Comment 1)** - Referee 3 (Report 1 04 Dec 2023) - reformated

We are grateful to the referee for devoting their time to our manuscript. The valuable comments and suggestions will help us to improve the paper.
We will here respond to comments made:

**(Major Comment A)** Different two-scale models in Lorenz (1996) and Lorenz (2005)
Figure R1 displays the two-scale model proposed by Lorenz (1996, 2006), including Eqs. (3.2)-(3.3) of Lorenz (2006). It is worth mentioning that Lorenz (1966) and Lorenz (2006) are the same article. Eq (3.2) for the large-scale flow does not include the explicit forcing term "F", which appears in his one-scale model. This is a typo. For the small-scale flow in Eq. (3.3), where F is not explicitly included, the coupling term acts as the forcing to derive the small scale process. Within the two-scale model, the grid system was illustrated in Figure R2 derived from Wilks (2005). Such a grid system is similar to the grid system of the multiscale modeling framework (MMF, e.g., Tao et al. 2008; Shen et al. 2011), consisting of a general circulation model (GCM, e.g., Lin et al. 2003; Lin 2004; Shen et al. 2006) for large-scale flows, and multiple copies of a cloud model (e.g., Tao 2003) for small-scale flows. Specifically, a copy of the cloud model at fine resolutions is embedded within each grid of the GCM.
Within the 2005 models, Lorenz first included additional nonlinear terms in the 1996 one-scale model (e.g., Eq. 8 in Figure R3) for slow variables (represented as Xn). Based on the 1996 one-scale model with coefficients of ("b2 ", "b", "0") for nonlinear terms, dissipative terms, and forcing term, respectively, a subsystem for fast variables (represented as Yn) was deployed and coupled with the subsystem for the slow variables. The coupled system with a 3 coefficient of "c" for coupling terms is referred to as the two-scale system (Eqs. 12a and 12b in Figure R4). The coupling terms were established based on a one-to-one relationship between Xn and Yn. Thus, the Lorenz 2005 two-scale model is different from the 1996 two-scale model. Will it be feasible for providing a diagram for illustrating the grid system within the 2005 two-scale model?

How good are such naive estimates? We can demonstrate some simple systems where they describe the situation rather well, at least on the average. One system is one that I have been exploring in another context as a one-dimensional atmospheric model, even though its equations are not much like those of the atmosphere. It contains the $K$ variables $X_1, \ldots, X_K$, and is governed by the $K$ equations

$$dX_k/dt = -X_{k-2}X_{k-1} + X_{k-1}X_{k+1} - X_k + F, \qquad (3.1)$$

where the constant $F$ is independent of $k$. The definition of $X_k$ is to be extended to all values of $k$ by letting $X_{k-K}$ and $X_{k+K}$ equal $X_k$, and the variables may be thought of as values of some atmospheric quantity in $K$ sectors of a latitude circle. The physics of the atmosphere is present only to the extent that there are external forcing and internal dissipation, simulated by the constant and linear terms, while the quadratic terms, simulating advection, together conserve the total energy $(X_1^2 + \cdots + X_K^2)/2$.

distinct time scales. The model has been constructed by coupling two systems, each of which, aside from the coupling, obeys a suitably scaled variant of Eq. (3.1). There are $K$ variables $X_k$ plus $JK$ variables $Y_{j,k}$, defined for $k = 1, \ldots, K$ and $j = 1, \ldots, J$, and the governing equations are

$$dX_k/dt = -X_{k-1}(X_{k-2} - X_{k+1}) - X_k - (hc/b)\sum_{j=1}^{J} Y_{j,k}, \qquad (3.2)$$

$$dY_{j,k}/dt = -cbY_{j+1,k}(Y_{j+2,k} - Y_{j-1,k}) - cY_{j,k} + (hc/b)X_k. \qquad (3.3)$$

The definitions of the variables are extended to all values of $k$ and $j$ by letting $X_{k-K}$ and $X_{k+K}$ equal $X_k$, as in the simpler model, and letting $Y_{j,k-K}$ and $Y_{j,k+K}$ equal $Y_{j,k}$, while $Y_{j-J,k} = Y_{j,k-1}$ and $Y_{j+J,k} = Y_{j,k+1}$. Thus, as before, the variables $X_k$ can represent the values of some quantity in $K$ sectors of a latitude circle, while the variables $Y_{j,k}$, arranged in the order $Y_{1,1}, Y_{2,1}, \ldots, Y_{J,1}, Y_{1,2}, Y_{2,2}, \ldots, Y_{J,2}, Y_{3,1}, \ldots$, can represent the values of some other quantity in $JK$ sectors. A large value of $J$ implies that many of the latter sectors are contained in one of the former, and we may think of the variables $Y_{j,k}$ as representing a convective-scale quantity, while, in view of the form of the coupling terms, the variables $X_k$ should represent something that favours convective activity, possibly the degree of static instability.

Figure R1: Lorenz 1996 one-scale (top) and two-scale (bottom) systems (Lorenz 2006). Since the 1996 model was applied to represent an atmospheric variable in K sectors of a latitude circle. Thus, the value of K indicates the number of grid points within the large-scale system, while the value of J represents the number of grid points within the small-scale system. Compared to the 2005 version, (1) the last term in Eq. (3.2) represents a feedback term that is a summation of small scale modes and (2) both Eqs. (3.2) and (3.3) contain two nonlinear terms, involving three neighboring grid points (at k-1, k+1, k+2).

The Lorenz '96 system (Lorenz 1996) is given by:

$$\frac{dX_k}{dt} = -X_{k-1}(X_{k-2} - X_{k+1}) - X_k + F - \frac{hc}{b}\sum_{j=J(k-1)+1}^{kJ} Y_j; \quad k = 1, \ldots, K$$

(1a)

$$\frac{dY_j}{dt} = -cbY_{j+1}(Y_{j+2} - Y_{j-1}) - cY_j + \frac{hc}{b}X_{\text{int}[(j-1)/J]+1}; \quad j = 1, \ldots, JK.$$ (1b)

It is is used here to define 'truth,' i.e. the quantities to be predicted. This system has been used in several previous studies as a metaphor for the atmosphere (Lorenz 1996; Palmer 2001; Smith 2001; Orrell 2002, 2003; Vannitsem and Toth 2002; Roulston and Smith 2003), although with slightly different notation. Equation (1a) describes the linked dynamics of a set of K slow, large-amplitude variables $X_k$, each of which is associated with J fast, small-amplitude variables $Y_j$ whose dynamics are described by Eq. (1b). Here $K = 8$ and $J = 32$, so that there are $JK = 256$ Y variables in total, as illustrated in Fig. 1. The scaling constants $h$, $c$, and $b$ are taken to be 1, 10, and 10, respectively, as is conventional; and $F$ is a forcing taken in the following to be either 18 or 20. The subscripts are cyclic so, for example, $X_0 = X_K$, $X_{-1} = X_{K-1}$, etc. and likewise for the Y variables.

[Figure]

Figure 1. Schematic illustration of the Lorenz '96 system (Eq. (1)) with $K = 8$ resolved variables $X_k$ (large circles), each associated with $J = 32$ unresolved variables $Y_j$ (unlabeled small circles) grouped according to the X variable to which they belong), so that there are $JK = 256$ Y variables in total. The forecast model (Eq. (2)) represents explicitly only the X variables, with contributions to each tendency that are due to the unresolved scales being parametrized in terms of the local resolved variable only.

Figure R2: Mathematical equations (top) and grid systems (bottom) within the Lorenz 1996 two-scale model (e.g., Wilks 2005). Eight large-scale variables (denoted as X) are selected at 8 data points within the large-scale system. Each large-scale variable acts as a force to drive a small-scale system consisting of thirty-two variables (denoted as Y). Based on the linear stability analysis, local growth rates should display a dependence on the number of data points in both the large-scale and small-scale systems and the coupling terms between the two-scale systems.

2. One chooses a number $K$, much smaller than $N$ and let $J = K/2$ if $K$ is even and $J = (K - 1)/2$ if $K$ is odd. Then, for any two sets of variables $X$ and $Y$, one defines

$$[X, Y]_{K,n} = \sum_{j=-J}^{J}{}' \sum_{i=-J}^{J}{}' (-X_{n-2K-i}Y_{n-K-j}$$
$$+ X_{n-K+j-i}Y_{n+K+j})/K^2$$

(7)

if $K$ is even, with $\Sigma'$ replaced by $\Sigma$ if $K$ is odd. The equation for Model II, where the only set of variables is $X$, will be

$$dX_n/dt = [X, X]_{K,n} - X_n + F.$$ (8)

Note that setting $K = 1$ makes $J = 0$; hence $[X, X]_{1,n}$ represents the single pair of products appearing in Eq. (1). Model II then reduces to Model I.

Figure R3: Equation (8) the above excerpt represents a revised one-scale model, proposed as the uncoupled version of the Model II in Lorenz (2005). The notation of [X, X] defined in Equation (7) indicates nonlinear terms. Compared to the original one-scale model in Figure R1 that contains a pair of nonlinear terms, a value of K > 1 in Eq. (8) suggests more than one pair of nonlinear terms. From a perspective of scale interactions (e.g., Lorenz 1969b), additional nonlinear terms (at different grid points) may improve the representation of scale interaction.

entirely by the coupling. One obtains the system

$$dX_n/dt = [X, X]_{K,n} - X_n - cY_n + F,$$ (12a)

$$dY_n/dt = b^2[Y, Y]_{1,n} - bY_n + cX_n,$$ (12b)

where, like $b$, the coupling coefficient $c$ is an additional parameter of the model.

Figure R4: Equation (12) in the above excerpt represents a revised two-scale model, proposed as Model II in Lorenz (2005). Here, Eq. (12a) is a revised large-scale system with more than one pair of nonlinear terms (when K > 1). Eq. (12b) indicated a revised small-scale system with one pair of nonlinear terms. In the coupled system, there exists a one-to-one relationship between the large-scale variable Xn and the small-scale variable Yn within the coupling terms.

**Response:** Figure RR1 shows the similarity of the 1996 (Eqs. (1a) and (1b) in Figure R1) and 2005 (Eqs. (12a) and (12b) in Figure R4) two-scale systems in the attempt to maintain 5 to 7 main highs and lows and several smaller waves for large scales $X_n$. While for the 1996 two-scale system, this is ensured by a number of $N$ large scale variables $X_n$ close to 30 (and a number of $JN$ variables for the small scales), for the 2005 system, it is ensured by linking the $X_n$ variables as described in Eq. (8) in Figure R3 (with the same number of small scale variables, however, determined from Eq. (3.1) in Figure R1, see Figure RR2). The 2005 two-scale system thus produces a smoother and more realistic evolution of the large-scale variable while maintaining properties similar to the 1996 system.

The systems used in this manuscript, which are described in Appendix A (of the manuscript), address one more condition that brings them closer to real systems. This condition is the fact that the large scale and small scale features in Eqs. (1a) – (1b) in Figure R2 and Eqs. (12a) – (12b) in Figure R4 are represented by separate sets of variables instead of appearing as superimposed features of a single set. To satisfy this condition, the coupling of one small-scale variable and one large-scale variable is more realistic than the coupling that is present in the 1996 system (Eqs. (1a) and (1b) in Figure R2).

[Figure]

Figure RR1: Comparison of longitudinal profiles at one time of two-scale Lorenz systems (a) from 1996 (Eqs. (1a) and (1b) in Figure R2) and (b) from 2005 (Eqs. (12a) and (12b) in Figure R4).

[Figure]

Figure RR2: Comparison of schematic illustrations of two-scale Lorenz systems (a) from 1996 (Eqs. (1a) and (1b) in Figure R2) and (b) from 2005, where the inner wave curve represents the large-scale variables described by Eq. (12a) in Figure R4, which produce 5-7 main waves, and where the outer curve represents the small-scale variables described by Eq. (12b) in Figure R4, which are not limited by the number of waves. In contrast to (a), one large scale variable is coupled to one small scale variable.

**(Major Comment B)** Dependence of findings on temporal spacing (i.e., Δt) and "spatial" spacing (e.g., the number of sectors, N)

As an analogy, the CFL condition, requiring cΔt/Δx < 1, here c is the space speed, suggests the importance of selecting temporal and spatial spacings for solution's stability. In this study, it is important to explore the impact of Δt and N.

Similarly, the concept of computational chaos (Lorenz 1989) also suggests the importance of wisely choosing Δt. Computational chaos appears "when the exact solution varies periodically with time, there is sometimes a range of time increment where the computed solution is chaotic" (Lorenz 2006). Computational chaos can be illustrated by a comparison of the Logistic differential equation and the Logistic map (i.e., difference equation). While the former has analytical, regular solutions, the latter produces chaotic solutions when a control parameter is sufficiently large. A dependence of the control parameter on a temporal spacing (i.e., Δt) can be shown by deriving the Logistic map from the Logistic differential equation (Shen et al. 2023). In this study, Δt is $1/240 \sim 4.2 \times 10^{-3}$ unit, N = 360 (indicating a "spatial" spacing), and L = 12 (i.e., indicating complexities of scale interaction). It would be ideal for additional tests with a smaller $\Delta t = 10^{-5}$ (or $\Delta t = 10^{-4}$). Additionally, the choice of N and L should be explored since N = 960 and L = 32 were used in Lorenz (2005).
As discussed below, the values of the coefficients for the coupling terms could impact the growth rate of the system as well.

**Response:** The choice of the variable $N = 360$ was made because the value of the largest Lyapunov exponent $\lambda^{L05}$ of the system described by Eq. (8) in Figure R3 ($F = 15$, time unit = 5 days) does not change for $N = 360$ and $N = 960$ (Table RR1) and therefore we chose the lower of the two values for computational efficiency.

| $N$ | $\lambda^{L05}$ |
| --- | --- |
| 30 | 0.70 |
| 60 | 0.29 |
| 90 | 0.35 |
| 120 | 0.32 |
| 150 | 0.33 |
| 360 | 0.33 |
| 960 | 0.33 |

Table RR1: Values of the largest Lyapunov exponent $\lambda^{L05}$ for selected numbers of variables $N$ in the 2005 Lorenz system (Eq. (8) in Figure R3, $F = 15$, time unit = 5 days).

Figure RR3 compares the time evolution of the average value of the variables for the 2005 Lorenz system (Eq. (8) in Figure R3) with time step $\Delta t=1/240$ and $\Delta t=1/2400$. It can be seen that the values are similar. Given this, we use the larger time step dt=1/240 for faster computations.

[Figure]

Figure RR3: Comparison of the time evolution of the mean value of the variables ($N = 360$) for the 2005 Lorenz system (Eq. (8) in Figure R3) based on the same initial conditions with time step $\Delta t$=1/240 (red dashed curve) and $\Delta t$=1/2400 (black dotted curve).

**(Major Comment C)** Impact of model's configuration and complexity on critical points (equilibrium points)

Based on the linearization theorem, critical points of the Lorenz systems could roughly indicate the local behavior of the solutions. As a result, initial error growth should display a dependence on the equilibrium state. Please consider identifying the appearance of the critical points and perform stability analysis using the Jacobian matrix of the linearized system at each of the critical points.

Below, a simple illustration for the linear stability analysis is provided using the 1996 one-scale model with N = 5. Based on the Figure R5 and Table R1, it is suggested that a larger F may produce a larger eigenvalue (a larger real part of the eigenvalue), suggesting a larger growth rate.

Based on the following preliminary analysis of the one- and two-scale models with the same value of the forcing parameter F, the effective forcing parameter for the two-scale model is smaller, yielding a smaller leading eigenvalue (i.e., a smaller real part of the eigenvalue). This is consistent with the finding that Figures 5 and 6 display larger growth rates ($\lambda$) within the one-scale system (e.g., L05-1) than the two-scale system (e.g., L05-2). [Such a finding is supported by the so-called aggregated negative feedback reported by Shen 2014, 2019.]

Consider Eqs. (A2) and (A3). From the nonlinear terms of Eq. (A2) and (A3), we expect that $X_{1,1} = X_{1,2} = X_{1,3} = \cdots X_{1,c}$ and $X_{2,1} = X_{2,2} = \cdots X_{2,c}$ may be a critical point. Here, $X_{1,c}$ and $X_{2,c}$ represent the value of steady state solutions for the slow and fast variables, respectively. From Eq. (A3), we have $X_{2,c} = cX_{1,c}/b$. Plugging the above into Eq. (A2), the right hand side of Eq. (A2) contains two dissipative terms, $-X_{1,n}$ and $-c^2 X_{1,c}/b$, yielding $X_{1,c} = bF/(b + c^2) < F$. Namely, the effective forcing for slow variables is weaker, indicating a smaller growth rate within the two-scale model, as compared to the one-scale model.

On the other hand, the above along with Figure R5 and Table R1 only provide a preliminary, qualitative, analysis. The authors may want to further verify or comment the above since the Jacobian matrix for the two-scale system that includes fast variables is larger, as compared to the Jacobian within the corresponding one-scale system.

 For example, with the two-scale or three-scale system, the value of parameter "b1" (b1 > 1) determine the (temporal) scale as well as the magnitude of the fast variables. Please provide justifications for the choice of b1 = 10 for the two-scale system but b1 = 1 for the three-scale system. Additionally, within the three-scale system, are nonlinear terms (e.g., c1 and c2 in Eq. A9) applied for coupling the "sub-systems" for the small- and medium-scale variables with the large-scale system? Please comment on the impact of c1 and c2 on system's stability.

$$
J_{L96} = \begin{pmatrix}
-1 & X_c & 0 & -X_c & 0 \\
0 & -1 & X_c & 0 & -X_c \\
-X_c & 0 & -1 & X_c & 0 \\
0 & -X_c & 0 & -1 & X_c \\
X_c & 0 & -X_c & 0 & -1
\end{pmatrix}.
$$

Figure R5: A Jacobian matrix for the linearized version of the Lorenz 1996 model with N = 5 from Eq. 3.1 in Figure R1. Here, Xc indicates a critical point solution and is equal to F.

Table R1: An eigenvalue analysis of the Lorenz 1995 one-scale model. The corresponding Jacobian matrix is shown in Figure R5. Here, Xc = F.

| Xc | eigenvalues |
|---|---|
| 0.5 | -0.4410 + 0.7694i |
|  | -0.4410 - 0.7694i |
|  | -1.0000 + 0.0000i |
|  | -1.5590 + 0.1816i |
|  | -1.5590 - 0.1816i |
| 1 | 0.1180 + 1.5388i |
|  | 0.1180 - 1.5388i |
|  | -1.0000 + 0.0000i |
|  | -2.1180 + 0.3633i |
|  | -2.1180 - 0.3633i |
| 10 | 10.1803 +15.3884i |
|  | 10.1803 -15.3884i |
|  | -1.0000 + 0.0000i |
|  | -12.1803 + 3.6327i |
|  | -12.1803 - 3.6327i |
| 20 | 21.3607 +30.7768i |
|  | 21.3607 -30.7768i |
|  | -1.0000 + 0.0000i |
|  | -23.3607 + 7.2654i |
|  | -23.3607 - 7.2654i |
| 30 | 32.5410 +46.1653i |
|  | 32.5410 - 46.1653i |
|  | -1.0000 + 0.0000i |
|  | -34.5410 +10.8981i |
|  | -34.5410 -10.8981i |

**Response:** While for analytical studies the instability of fixed points (critical points) is certainly of high interest, we are interested in the typical error growth and therefore focus on the Lyapunov exponent on the chaotic attractor. Since the phase space is so high dimensional, we are not even sure that unstable fixed points are embedded in the chaotic attractor or whether they are outside, as they are in the Lorenz 1963 low dimensional model. We therefore calculate the maximal LE numerically in the following way: a reference trajectory (considered the "truth" or verification) and a trajectory which is the numerical solution of the systems with a given error, are repeatedly generated. For this scheme to be meaningful, we have to ensure that the reference trajectory is on the system's attractor and that the repetition of this scheme samples the whole attractor with correct weights (the invariant measure). We solve this issue in the following way: We first integrate the system over ten years (175200 steps), starting from arbitrary initial conditions, and assume that after discarding this transient, the trajectory is on the attractor. We continue to integrate this single trajectory and consider segments of it as reference trajectories for error growth, i.e., the many reference trajectories are simply segments of one very long trajectory, which ensures not only that all these segments are located on the attractor but that in addition, they sample the attractor according to the invariant measure.

Figure RR4 compares the error growth rates of the L05-1 (Eq. (A1) in manuscript), L05-2 (Eq. (A8) in manuscript), and L05-3 (Eq. (A9) in manuscript) systems. In contrast to the reviewer's findings, the figure shows the smallest growth rate for the L05-1 system and the largest for the L05-3 system. We confirm that the effective forcing for slow variables is weaker, indicating a smaller growth rate within the two-scale model, as compared to the one-scale model. However, it should be noted that in Figure RR4 the values of the single-scale system (L05-1) are not compared with the large-scale values of the multi-scale systems (L05-2 and L05-3), but are compared with the total values of the L05-2 and L05-3 systems, where the large-scale and small-scale features are appearing as superimposed features of a single set.

[Figure]

Figure RR4: Initial error growth tendency (rate) $dE/dt$ as a function of the error magnitude $E$ for L05-1 system (Black, Eq. (A1) in manuscript), for L05-2 system (Red, Eq. (A8) in manuscript), and for L05-3 system (Blue, Eq. (A9) in manuscript).

A justification for the use of the L05-2 (Eq. (A8) in manuscript) and L05-3 (Eq. (A9) in manuscript) systems as the "reality" and the L05-1 system as the "model." is presented in the manuscript (Lines 220-228 in revised manuscript):

"This approach is justified by the fact that the L05-2 and L05-3 systems can be viewed as a variant of the L05-1 system:

$$dX_{tot,n} / dt = [X_1, X_1]_{L,n} - X_{1,n} + \tilde{F}_n(t),\qquad(12)$$

where $\tilde{F}_n(t) = b^2 [X_2, X_2]_{1,n} + c [X_2, X_1]_{1,n} - bX_{2,n} + F$ for the L05-2 system and

$\tilde{F}_n(t) = b_1^2 [X_2, X_2]_{1,n} + b_2^2 [X_3, X_3]_{1,n} + c_1 [X_2, X_1]_{1,n} + c_2 [X_3, X_2]_{1,n} - b_1 X_{2,n} - b_2 X_{3,n} + F$ for the L05-3 system are treated as a forcing, which varies in a complicated manner with time. We parameterize these small-scale phenomena contained in $\tilde{F}_n(t)$ by the average value of these phenomena, which is close to zero, and therefore we can write:

$$\langle \tilde{F}_n(t) \rangle \approx F = 15,\qquad(13)$$

where $\langle ... \rangle$ represents the mean calculated over a long orbit on the L05-2 and L05-3 systems attractors."

The parameters of any multi-level Lorenz's system (L96-2, L05-2, L05-3) should be set so that all levels behave chaotically (the largest Lyapunov exponent of each level is positive) and that all levels have a significant difference in amplitudes and fluctuation rates. For the L-96 system (Eq. (3.1) in FIgure R1), the chaotic behavior is determined by the value of $F$, and the number of variables $N$. Lorenz (2005) states that as long as $N \geq 12$ chaos is found when $F > 5$ (for $N = 4$ it is when $F > 12$ and for $N > 6$ when $F > 8$). In cases such as the L96-2 system (Eqs. (1a) and (1b) in Figure R2), where the forcing $F$ acts only on the largest scale, the chaotic behavior of smaller scales is created by coupling. The size of the coupling is cascaded from the largest scale to the smaller ones. Because the values of the largest scale variables are determined by the forcing $F$, the $F$ value indirectly affects the smaller scales' chaotic behavior and must be chosen large enough to ensure chaotic behavior through coupling for all scales (levels). For the L05-2 system (Eq. (A8)), variables are superposed features of a single set calculated by Eqs (A4) and (A5). In addition to those mentioned above, this procedure affects the chaotic behavior, amplitude, and fluctuation rate of the levels, and the choice of $I$ between 10 and 20 may be optimal (Lorenz, 2005). In order to maintain the required properties of the two scales L05-2 system, Lorenz (2005) chose $N = 960$, $L = 32$, $I = 12$, $F = 15$, $b = 10$, and $c = 2.5$ (**note that for L05-2 and L05-3 systems it is not possible to directly determine the amplitude and fluctuation rate of smaller scales using spatiotemporal scaling factors $b$, because these values are mainly determined by the procedure for expressing variables and the length of the intervals $[-I, I]$**).

For the L05-3 system (Eqs. (A9) – (A12)), it is necessary to specify eight parameters. We tested that the values of coupling coefficients $c_1$ and $c_2$ do not affect the L05-3 system compared to the values of other parameters, and therefore for simplification $c_1 = 1$ and $c_2 = 1$. The parameter $F = 15$ is set the same as for other L05 systems. For the medium scale amplitude to be approximately ten times smaller than the large scale amplitude and the small scale amplitude to be approximately ten times smaller than the medium scale amplitude and for the scales to have different oscillation rates, the spatiotemporal scale factors are chosen $b_1 = 1$ and $b_2 = 10$ and interval lengths $I_1 = 20$, and $I_2 = 10$. $N = 360$ turned out to be most suitable for the chaotic behavior of all three levels (found experimentally).

**(Major Comment D)** Separations of initial and model errors

Based on the linearization theorem, a locally linearized system may represent the local feature of the corresponding nonlinear system (for a hyperbolic critical point). The stability of the linearized system depends on locations of the critical points that depend on model's complexity (i.e., nonlinear terms in the system). Thus, the model complexity (i.e., nonlinear terms) could impact the critical points and thus the growth of the initial errors. As a result, it is not easy to separate the initial errors and model errors. (For example, given the same initial error for a large-scale variable, the time varying difference between two nearby trajectories are different in two different models.)

**Response:** We fully agree with the comment. We simulate the initial error growth in the same systems (perfect model assumption), and the model error growth with zero initial error (perfect initial conditions assumption). Combination of both is studied in section 3.3 of the manuscript.

**(Major Comment E)** Validity of error saturation for periodic attractors and coexisting attractors

Earlier studies suggest that the Lorenz 1996 two-scale model could produce nonlinear periodic solutions. In your ensemble runs, have you observed periodic solutions? Can you comment on the validity of error saturation for periodic solutions?

Additionally, recent studies reported the appearance of multistability (for coexisting attractors) within the 1996 model (e.g., Van Kekem and Sterk 2018a,b, 2019; Pelzer et al. 2020). Have you observed multistability in your ensemble runs?

**Response:** In our research, we focused only on the average value of error growth (over variables and number of runs). We set all the scales through the parameters of the Lorenz systems to behave chaotically (details can be found in Bednar and Kantz (2022)) and the evolution of the average error growth did not show signs of periodic solution or multistability.

**Specific Comments:**

**(Specific Comments 1)** Please check consistency in the capitalization of the initial letters of words within a title.

**Response:** We checked and fixed it. Thank you for pointing this out. (Lines 1-2 in revised manuscript):

"Analysis of model error in forecast errors of xtended tmospheric Lorenz' 05 ystems and the ECMWF system"

**(Specific Comments 2)** Lines 45-50, the application of the Lyapunov exponent (LE) is not accurate. A global LE represents a long-term average of "local" growth rates (determined by the separations of two nearby trajectories). Initial separations should remain small. Local growth rates may vary with time. As a result, Eq. (1) with a constant growth rate is valid only for a finite time interval. During different time internals, different growth rates may appear. Note that in addition to one positive LE, solution's boundedness is another important feature that defines a chaotic system.

**Response:** We have added information about boundedness and validity for a finite time interval. (Lines 47-48 in revised manuscript)

"In low-dimensional bounded chaotic systems with at least one positive Lyapunov exponent, the growth of infinitesimal errors is exponential for a finite time interval, given by a linear time derivative:"

**(Specific Comments 3)** Lines 45-55, please consider referring to the growth rates in Eqs. (1) and (2) as the exponential growth rate (with a J-shaped curve) and logistic growth rate (with a S-shaped curve), respectively.

**Response:** We changed the description of Eqs. (1) and (2). (Lines 808-810, 820-822, 833-835, 846-848, 861-863 in revised manuscript).

"the early part of the growth by exponential growth rate $dE_{ex}$ (Eq. (1), green, dashed), exponential growth rate with model error $dE_r$ (Eq. (5), blue, dashed), power law $dE_p$ (Eq. (3), red, dashed) and approximation of the full curve by growth rate of quadratic hypothesis $dE_{qu}$ (Eq. (2), green), growth rate of quadratic hypothesis with model error $dE_q$ (Eq. (6), blue) and extended power law"

"the early part of the growth by exponential growth rate $dE_{ex}$ (Eq. (1), green, dashed), exponential growth rate with model error $dE_r$ (Eq. (5), blue, dashed), power law $dE_p$ (Eq. (3), red, dashed) and approximation of the full curve by growth rate of quadratic hypothesis $dE_{qu}$ (Eq. (2), green), growth rate of hypothesis with model error $dE_q$ (Eq. (6), blue) and extended power law $dE_{ep}$ "

"the early part of the growth by exponential growth rate $dE_{ex}$ (Eq. (1), green, dashed), exponential growth rate with model error $dE_r$ (Eq. (5), blue, dashed), power law $dE_p$ (Eq. (3), red, dashed) and approximation of the full curve by growth rate of quadratic hypothesis $dE_{qu}$ (Eq. (2), green), growth rate of quadratic hypothesis with model error $dE_q$ (Eq. (6), blue) and extended power law"

"black, dot-dashed for $E(0) = 0.2$ ), approximation of the early part of the model growth by exponential growth rate $dE_{ex}$ (Eq. (1), green, dashed), exponential growth rate with model error $dE_r$ (Eq. (5), blue, dashed), power law $dE_p$ (Eq. (3), red, dashed) and approximation of the full curve by growth rate of quadratic hypothesis $dE_{qu}$ (Eq. (2), green), growth rate of quadratic hypothesis"

"black, dot-dashed for $E(0) = 0.2$ ), approximation of the early part of the model growth by exponential growth rate $dE_{ex}$ (Eq. (1), green, dashed), exponential growth rate with model error $dE_r$ (Eq. (5), blue, dashed), power law $dE_p$ (Eq. (3), red, dashed) and approximation of the full curve by growth rate of quadratic hypothesis $dE_{qu}$ (Eq. (2), green), growth rate of quadratic hypothesis"

**(Specific Comments 4)** Line 80, the term "error growth laws" should be rephrased since they are not necessarily physical laws.

**Response:** We replaced the term law with the term hypothesis. (Lines 81, 309 in revised manuscript)

"While the above-listed error growth lawsapproximations are supposed to approximate the effectively observed average error"

"numerical error growth curves using the hypotheses or laws Eqs. (1) - (6) and try to identify the most appropriate description. "

**(Specific Comments 5)** Lines 122, statements are not accurate. Unless additional forcing terms are introduced, improving model's spatial or temporal resolution does not necessarily enhance instability. (Please think of a convergent Taylor series.)

**Response:** We added to the introduction:

"Buizza (2010), Magnusson and Kallen (2013) or Jacobson (2001) show that improving the model's spatial and temporal resolution will improve the ability to predict, especially for short forecast range (Buizza, 2010). However, the cited studies work with models that do not model small spatiotemporal phenomena (they are parameterized) and whose initial condition error magnitude is larger than the magnitude of these phenomena. We have verified the fact that the high resolution model (that models small scales) is less stable than the low resolution model (that doesn't model small scales) against initial condition errors (Bednar and Kantz, 2022; Budanur and Kantz, 2022), and that therefore the issue of omitting small scales has another facet. Our new approach models and omits small spatiotemporal scales using…"

(Lines 129-135 in revised manuscript)

**(Specific Comments 6)** Lines 128-130: it is wired that the two-scale system contains large- and small-scale systems while the three-scale system adds a medium scale, in addition to large- and small-scale flows. Any justifications?

**Response:** It would be more natural to take the L05-2 and L05-1 systems as the model and the L05-3 system as the reality. ). A variant where the L05-2 system was used as the model and the L05-3 system as the "reality" was also tested. The resulting model error growth is approximately identical to the previous variant (L05-1 system as the model and L05-3 system as the "reality"). That's why we chose the settings we present. Further, it would be more natural for the L05-2 system to have a small scale comparable to the medium scale of the L05-3 system. However, our intention was to be close to the L05-2 system presented by Lorenz (2005), whose small scale is equivalent to the small scale of our L05-3 system.

**(Specific Comments 7)** Lines 160-165, have you observed coexisting attractors (e.g., more than one attractors) in your ensemble runs? (e.g., see multistability in Van Kekem and Sterk 2018a,b, 2019; Pelzer et al., 2020).

**Response:** In our research, we focused only on the average value of error growth (over variables and number of runs) and we did not observe signs of multistability.

**(Specific Comments 8)** Line 170, does the statement "errors might even shrink in short times" indicates the existence of a stable manifold?

**Response:** Yes, the Lorenz L05-systems possess rather high dimensional stable manifolds, along which trajectories are attracted towards the attractor. Calculation of the Lyapunov-dimension done by us for L05-2 show this very clearly, the attractor dimension is much smaller than the phase space dimension, where the attractor is the unstable manifold. But the statement on line 170 does not indicate the existence of a stable manifold but the fact that initial perturbations might not point into the locally most unstable direction.

**(Specific Comments 9)** Lines 194, while N=360 was used in this study, N=960 was appied in Lorenz (2005).

**Response:** Thank you for pointing this out. The problem is already discussed in comment **(Major Comment B)**.

**(Specific Comments 10)** line 186, how many time steps for the transfer of error to the small-scale variables?

**Response:** The error would immediately (one time step) propagate into the small-scale variables.

**(Specific Comments 11)** Section 3.1, please confirm whether the leading LE in the L05-1 system is larger (smaller) than that in the L05-2 (L05-03) system.

**Response:** Figure RR4 compares the error growth rates of the L05-1 (Eq. (A1) in manuscript), L05-2 (Eq. (A8) in manuscript) and L05-3 (Eq. (A9) in manuscript) systems. The figure shows the smallest growth rate for the L05-1 system and the largest for the L05-3 system (therefore also for LE). It should be noted that for the L05-2 and L05-3 systems, the error growth rate is scale dependent.

**(Specific Comments 12)** Line 382-394: The key point that higher resolution model produces better predictability is acceptable. However, it is not clear whether Figure 10 is sufficient to support this point. Please see details in the last specific comment below.

**Response:** Please see the discussion at: **(Specific Comments 17)**

**(Specific Comments 13)** Line 656: The statement "Based on the fact that scale-dependent error growth implies an intrinsic predictability limit" is not accurate. A finite growth rate may indicate a limit for practical predictability. By comparison, a finite intrinsic predictability is established by the feature of chaos (e.g., sensitive dependence on initial condition, SDIC; e.g., Shen, Pielke Sr., and Zeng, 2023)

**Response:** Our statement really refers to the finite intrinsic predictability that is established by the features of chaos. The statement is based on Brisch and Kantz (2019), Bednar and Kantz (2022), and Budanur and Kantz (2022).

**(Specific Comments 14)** Lines 612 - 623, discussions are duplicated; they are the same as those in Lines 600-611.

**Response:** We deleted the duplicated part. Thank you for pointing this out.

**(Specific Comments 15)** Line 715, the parameter "K" should be replaced by "L".

**Response:** We replaced $K$ by $L$. Thank you for pointing this out.

**(Specific Comments 16)** Line 716, Lorenz (2005) did not explicitly suggest the ratio of N/L = 30 nor provide justification for the choice of N = 960 and L = 32.

**Response:** We assume the requirement for a model to have 5 to 7 main highs and lows that correspond to planetary waves (Rossby waves) and several smaller waves corresponding to synoptic-scale waves, and we follow the text of Lorenz (2005) on the pages 1579 (Fig. RR5) and 1580 (Fig RR6).

Figure 4a shows typical profiles produced by Eq. (8) when $N = 240$ and $F = 10$ for selected values of $K$. When $K = 2$, there are nearly as many waves as if Eq. (1) had been retained. Increasing $K$ to 4 decreases the number, but there are still too many compared with Fig. 1a. When $K = 8$, one evidently succeeds in producing an acceptable number of major waves, although weaker smaller-amplitude waves are superposed. In drawing the curve I have, as usual, connected the successive values of $X_n$ with straight-line segments, but these are hard to detect. Any other reasonable interpolation procedure would have produced an indistinguishable curve. Increasing $K$ to 16, 32, or 64 lengthens the waves still more, and, evidently, one can produce any wave-number desired by choosing $K$ judiciously.

Since the ratio $N/K$ is 30 in the third profile, whose dominant wavenumber agrees most closely with Fig. 1a, where $N = 30$, there is a suggestion that the appearance of a profile may depend largely upon $N/K$. Figure 4b is constructed with $F = 10$ and $N/K = 30$ in each profile, and with $N$ successively doubling from 30 in the leading profile to 960 in the final one. The conjecture seems to be well supported; the profiles in Fig. 4b show little resemblance to any profile in Fig. 4a except the third one.

With $N = 960$ and again with $K = 32 = N/30$ and $F = 10$, Fig. 5a has been constructed in the manner of Fig. 1a; it shows profiles produced by Eq. (8) at 6-h intervals for five days. Again, at least for the five days, the major crests and troughs retain their identities, while minor ones come and go. One can conclude that Model II is ready for some applications for which Model I would have been inadequate.

For Model I the doubling time for small errors, as seen in Fig. 2a, depends strongly upon $F$, but is nearly independent of $N$ if $N$ is not too small. For Model II, with $K > 1$, it also proves to depend strongly upon $F$ while being nearly independent of $N$ and $K$ if $N/K$ is not too small, but, for a given value of $F$, it is much smaller when $K > 1$ than when $K = 1$. Thus, for the values used in Fig. 5a, the doubling time is about four days—considerably longer than expected in the atmosphere. It can be restored to a more nearly atmospheric value by increasing $F$.

Figure 5b is constructed like Fig. 5a, again with $N =$

Figure RR5: Page 1579 in Lorenz (2005).

[Figure]

Fig. 4. (a) Profiles of $X$ produced by Eq. (8) with $N = 240$, $F = 10$, and values of $K$ indicated by numbers at left. Scale at bottom is gridpoint number. (b) Profiles of $X$ produced by Eq. (8) with values of $N$ indicated by numbers at left and with $K = N/30$ and $F = 10$. Scale at bottom is gridpoint number for bottom curve.

960 and $K = 32$, but with $F = 15$. There is still a suggestion of six or seven longer waves, but the shorter waves are more in evidence. Note that, with $N$ so large, even these shorter waves are 30 or more grid intervals long—the point-to-point variations are very smooth. The doubling time has been reduced to about two days.

Apparently, in trying to make the curves produced by Model II look like reasonable spatial interpolations of the kind of curve produced by Model I, one must choose between too long a doubling time (smaller $F$) or unanticipated shorter waves (larger $F$). The value $F = 15$ is a compromise.

Fig. 5. (a) Profiles of $X$ produced by Eq. (8) with $N = 240$, $K = 8$, and $F = 10$, at 12-h intervals for 5 days. Scale at bottom is gridpoint number. Numbers at left indicate chronological order of profiles. (b) Same as (a) but with $F = 15$.

Figure RR6: Page 1580 in Lorenz (2005).

**(Specific Comments 17)** page 40, line 870-875, Figure 10. Figure's title and captions are confusing. Since L05-02 and L05-03 systems were used to provide the "ground true" (or reference) for computing errors, these errors do not represent the errors of the L05-02 and L05-03 systems, respectively, the growth of initial errors within the L05-02 or L05-03 system does contribute to the growth of differences of the solutions between the L05-1 and L05-02 (or L05-03) systems.

For a comparison in Figures 5-7, let's simply choose λ+, = 0.33, 0.29, and 0.46 for the L05-1, L05-2, and L05-3 systems, respectively. The comparison of the above selected growth rates produces a consistent finding that larger differences (in error growths) are reported in Figure 10b than in Figure 10a. However, on the other hand, considering differences between the L05-02 and L05-03 systems, the differences may produce the largest growth rates as compared to those in Figure 10a and Figure 10b.

**Response:** The question under investigation in this paper is whether omitting small scale atmospheric phenomena, which contribute little to the final value, will improve the predictability of the resulting value. In other words, how does the average forecast error growth change in a model where small-scale phenomena are omitted but where model errors are therefore introduced, compared to a model where all phenomena are present but the average forecast error growth is scale-dependent. So if we use L05-02 and L05-03 systems to provide

the "ground true" (or reference) then, when searching for an answer to the research question, it is reasonable to use the results presented in Figure 10.

Figures 5-7 show that, the L05-1 system is a classical chaotic system with the largest Lyapunov exponent of about $\lambda \approx 0.33$ 1/ day. The data of the L05-2 and L05-3 are best approximated by the power law . For a power law: $\lambda_p(E) := \dfrac{d\ln(E)}{dt} = \dfrac{\dot{E}}{E} = aE^{-\sigma}$, with an exponent $\sigma$ and a coefficient $a > 0$, the error growth rate $\lambda(E) \approx \dfrac{1}{\Delta t}\ln(E(t+\Delta t)/E(t))$ is expected to be a function of the error magnitude $E$, and is not constant as for classical chaotic systems. For exponential growth (classical chaos) $E_{\exp}(t) = E_0 e^{\lambda_{\exp}t}$ and for an initial error $E_0$ going to zero, the time $t_{lim}$ at which the error reaches a limiting value $E_{lim}$, goes to infinity: $t_{lim} = \dfrac{\ln E_{lim} - \ln E_0}{\lambda_{\exp}} \to \infty \ for \ E_0 \to 0.$

However, a strict predictability limit $t_{lim}$ exists for scale-dependent error growth even when the initial error $E_0$ vanishes. For a description by a power law $dE_p$, the predictability limit $t_{lim}$ is: $t = \left(E^b(t) - E_0^b\right)/(a \cdot b) \ \to \ t_{lim} = E_{lim}^b/(a \cdot b) < \infty \ for \ E_0 \to 0.$

It is true that if we show the growth of the model and initial error in Figure 10, this is the initial error of the L05-1 system, but this is consistent with the question under investigation. At the same time, Figure 10 compares the strictly model error growth (no initial error) with the strictly initial error growth (L05-2, L05-3 systems), where the initial error is limiting towards zero and is then a strict predictability limit.

References:

Bednář, H., and Kantz, H.: Prediction error growth in a more realistic atmospheric toy model with three spatiotemporal scales, Geosci. Model Dev., 15, 4147–4161, https://doi.org/10.5194/gmd-15-4147-2022, 2022.

Brisch, J., and Kantz, H.: Power law error growth in multi-hierarchical chaotic system-a dynamical mechanism for finite prediction horizon, New J. Phys., 21, 1–7, https://doi.org/10.1088/1367-2630/ab3b4c, 2019.

Budanur, N. R., Kantz, H.: Scale-dependent error growth in Navier-Stokes simulations, Phys. Rev. E, 106, 1–7, https://doi.org/10.1103/PhysRevE.106.045102, 2022.

Buizza, R.: Horizontal resolution impact on short- and long-range forecast error, Quarterly Journal of the Royal Meteorological Society, 136, 1020–1035, https://doi.org/10.1002/qj.613, 2010.

Jacobson, M. Z.: GATOR-GCMM: 2. A study of day- and nighttime ozone layers aloft, ozone in national parks, and weather during the SARMAP field campaign, J. Geophys. Res., 106, 5403-5420, https://doi.org/10.1029/2000JD900559, 2001.

Lorenz, E. N.: Predictability: a problem partly solved, in: Predictability of Weather and Climate, edited by: Palmer, T., and Hagedorn, R., Cambridge University Press, Cambridge, UK, 1–18, https://doi.org/10.1017/CBO9780511617652.004, 1996.

Lorenz, E. N.: Designing chaotic models, J. Atmos. Sci., 62, 1574–1587, https://doi.org/10.1175/JAS3430.1, 2005.

Magnusson, L., and Kallen, E.: Factors Influencing Skill Improvements in the ECMWF Forecasting System, Mon. Wea. Rev., 141, 3142–3153, https://doi.org/10.1175/MWR-D-12-00318.1, 2013.

**(Comment 2)** Furthermore, considering the authors' assertion that the proposed 05 system simulates "5 to 7 main highs and lows that correspond to planetary waves (Rossby wave))," it would be advantageous to discuss whether the proposed system, without the Coriolis force, could replicate key features of the Rossby wave, including phase speeds. Historically, experiments such as dishpan experiments aimed to "simulate" weather features, yielding diverse outcomes like chaotic solutions and vacillation (e.g., limit cycle).

**Response:** Lorenz and Emanuel (1998) showed that the initial wave of the L96 system has a westward phase velocity and an eastward group velocity, which is in agreement with the evolution of Rossby waves. We show a description of the evolution of the incipient waves of the L96 system in Figure RR7. Lorenz and Emanuel (1998) also showed a numerical calculation of the evolution of the L96 system, which is presented in Figure RR8. In the same manner, we present in Figure RR9 the numerical calculation of the evolution of the L05 system for $N = 30$ (left column) and $N = 360$ (right column). From Figures RR8 and RR9, we can see the agreement and confirmation of the theoretical calculation.

$$dX_j/dt = (X_{j+1} - X_{j-2})X_{j-1} - X_j + F, \qquad (1)$$

As with typical nonlinear systems, any solutions found analytically are likely to be rather specialized, but certain properties may be deduced without solving the equations at all. First, if a bar (¯) over a quantity denotes an average over all values of $j$ and over a long enough time to make average time derivatives negligibly small, it follows from multiplying (1) by $X_j$ and averaging that

$$\overline{X^2} = F\overline{X}, \qquad (2)$$

whence

$$\overline{X^2} - \overline{X}^2 = \overline{X}(F - \overline{X}). \qquad (3)$$

Since the variance $\sigma^2 = \overline{X^2} - \overline{X}^2$ of $X$ is nonnegative, it follows from (3) that the mean $\overline{X}$ of $X$ lies in the interval $[0, F]$, whence the standard deviation $\sigma$ lies in the interval $[0, F/2]$. In the obvious steady solution where $X_j = F$ for each $j$, $\overline{X} = F$ and $\sigma = 0$.

Small perturbations $x_j$ about the steady solution obey the equation

$$dx_j/dt = F(x_{j+1} - x_{j-2}) - x_j. \qquad (4)$$

If we let

$$x_j = \sum_k p_k \exp(ikj), \qquad (5)$$

$$dp_k/dt = [(e^{ik} - e^{-2ik})F - 1]p_k. \qquad (6)$$

The steady state is therefore unstable if $(\cos k - \cos 2k)F > 1$ for some $k$. The factor $\cos k - \cos 2k$ assumes its maximum positive value 9/8 when $\cos k = 1/4$; thus the steady solution becomes unstable with respect to waves of length $2\pi/\cos^{-1}(1/4) = 4.77$ zones when $F$ exceeds 8/9. Since the actual number of zones in a wave must be a divisor of $J$, we find, for $J = 40$, that waves of five zone lengths, or wavenumber 8, will begin to grow when $F$ exceeds $(\cos 2\pi/5 - \cos 4\pi/5)^{-1} = (4/5)^{1/2} = 0.894$, a value only slightly exceeding 8/9.

The incipient waves will move with velocity $c = -(\sin k + \sin 2k)(F/k)$, which, when $k = 2\pi/5$ and $F = 0.894$, equals $-1.09$; thus they will drift westward. The group velocity $c_g = -(\cos k + 2\cos 2k)F$, on the other hand, equals $+1.17$, implying eastward propagation of regions of enhanced activity; this feature has been an important consideration in our decision to adopt the model.

Figure RR7: Description of the evolution of the incipient waves of the L96 system (Pages 400-401 in (Lorenz and Emanuel, 1998)).

[Figure]

FIG. 1. Longitudinal profiles of $X_j$ at 6-h intervals, as determined by Eq. (1) with $N = 40$ and $F = 8.0$, when initially $X_{20} = F + 0.008$ and $X_j = F$ when $j \neq 20$. On horizontal portion of each curve, $X_j = F$. Interval between successive short marks at left and right is 0.01 units.

Figure RR8: Numerical documentation of the evolution of the incipient waves of the L96 system (Page 401 in (Lorenz and Emanuel, 1998)).

[Figure]

Figure RR9: Numerical documentation of the evolution of the incipient waves of the L05 system for $N = 30$ (left column) and $N = 360$ (right column). Longitudinal profiles of $Xj$ at 6-h intervals, with $F = 15$, when initially $X_{15} = F + 0.008$ or $X_{180} = F + 0.008$ and $X_j = F$ when $j \neq 15$ or $j \neq 180$.

**(Comment 3)** This study extends from the authors' previous research. The reviewer acknowledges the related efforts. However, after examining their earlier studies, the reviewer proposes the following:

**(Comment 3.1)** Document and report the calculation of Lyapunov exponents (LEs) within the proposed 05 system. For instance, employing the 1963 model with common parameters, the largest LE (LE1) is 0.906, as exemplified in the link provided (https://sprott.physics.wisc.edu/chaos/lorenzle.htm). This task holds significant importance.

**Response:** Table RR1 shows the values of the largest Lyapunov exponent *LE1* of the L05 system described by Eq. (8) in Figure R3 for selected numbers of variables $N$ ($F = 15$, time unit = 5 days) calculated by Sprott's (2006) method.

| N | LE1 |
|---|-----|
| 30 | 0.70 |
| 60 | 0.29 |
| 90 | 0.35 |
| 120 | 0.32 |
| 150 | 0.33 |
| 360 | 0.33 |
| 960 | 0.33 |

Table RR1: Values of the largest Lyapunov exponent *LE1* for selected numbers of variables $N$ in the 2005 Lorenz system (Eq. (8) in Figure R3, $F = 15$, time unit = 5 days).

We also calculated the largest Lyapunov exponent *LE1* in the L05-3 system (three scales, N = 390, $F = 15$, time unit = 5 days) using the method of Sprott (2006). We determined the maximal Lyapunov exponents in all four cases and find the values $LE1 = 2.5$ (day)$^{-1}$ for overall and small scale and $LE1 = 2$ (day)$^{-1}$ for medium and small scale. The similarity of the values for all levels indicates that they are coupled, so that the maximal Lyapunov exponent when calculated in the double limit $E_0 \to 0$ and $t \to \infty$ shows up in arbitrary subsystems. The evolution of the errors $E$ can always be studied in a way to see the largest exponent of the system (done here), but also in a way to see a value which would be the exponent of the corresponding sub-system if one were able to isolate this, but this cannot be calculated using standard methods for calculating the largest Lyapunov exponent.

**(Comment 3.2)** Develop the error growth model, e.g., dE/dt = sigma E (1 - E/Es), and furnish a mathematical expression for sigma and LE1 of the proposed system. It should be noted that the long-time average of (1/E dE/dt) is not precisely equal to sigma.

For a chaotic L05 system with average initial error growth E(t), the largest Lyapunov exponent is defined as: $LE1 = \lim_{t \to \infty} \lim_{\epsilon \to 0} (1/t) \ln(E(t)/\epsilon)$. This exponential growth is associated with single scale systems, infinitesimal initial error $\epsilon$, and the early part of the error growth. For a not infinitesimally small initial error and the entire evolution of the error, Lorenz (1982) defined the quadratic hypothesis: dE/dt = sigma E (1 - E/Es). The error growth rate, for comparison with LE1, can be determined as: 1/E dE/dt = sigma (1 - E/Es). Thus, the sigma determines the value at the beginning of the decrease in the error growth rate (dE/dt = sigma*E(t) in the limit E/Es << 1 with sigma ≈ LE1, Figure RR10). Sigma is an approximation of LE1, which is biased by the error due to the approximation of the data, not the infinitesimal initial error, and the use of data from the entire development period.

[Figure]

Figure RR10: Exponential growth E(t) = E(0) exp(sigma*t) (left figure, black curve) and sigma determined from 1/E(dE/dt) of E(t) = E(0) exp(sigma*t) as a function of E (right figure, black curve). Growth of E(t) determined from the quadratic hypothesis (left figure, red curve) and linear decline determined from 1/E dE/dt = sigma (1 - E/Es) as a function of E (right figure, red curve).

References:

Lorenz, E. N.: Atmospheric predictability experiments with a large numerical model, Tellus, 34, 505–513, https://doi.org/10.1111/j.2153-3490.1982.tb01839.x 1982.

Lorenz, E. N. and Emanuel, K. A.: Optimal sites for supplementary weather observations: Simulation with a small model, Journal of the Atmospheric Sciences, 55, 399–414, https://doi.org/10.1175/1520-0469(1998)055<0399:OSFSWO>2.0.CO;2 1998.

Sprott, J. C.: Chaos and Time-series Analysis, Oxford University Press, New York, USA, 2006.

**Referee 3 (Report 1 15 May 2024)**

**We are grateful to the referee for devoting time to our manuscript.**
**We will here respond to comments made:**

**(Major Comment A)** Concerning the physical relevance of the model, the qualitative features such as patterns, phase, and group velocities within the system fail to convince, primarily because the equation omits mechanisms like potential vorticity (PV) generation necessary for planetary waves (Rossby waves). To address this issue, it is advisable to include a disclaimer stating: "Although the equation lacks the forcing terms essential for generating planetary waves, it qualitatively mimics features similar to those of the Rossby wave."

**Response:** We have added the information that planetary waves are not generated by potential vorticity. (Lines 717-721 in revised manuscript):

To a certain extent, the model quantitatively describes weather systems, but unlike the well-known Lorenz model of atmospheric convection (Lorenz, 1963), it cannot be derived from any atmospheric dynamic equations. The motivation was to formulate the simplest possible set of dissipative chaotically behaving differential equations that share some properties with the "real" atmosphere. Although mechanisms such as potential vorticity generation are lacking in the equations, the model generates 5 to 7 main highs and lows corresponding to planetary waves (Rossby waves). To keep 5 to 7 main highs and lows , Lorenz (2005) suggested a ratio $N / L = 30$ and $F = 15$.

**(Major Comment B)** Regarding the impact of small-scale processes, this study and the existing literature identify two types of such processes.

Type (1) involves small-scale processes introduced through an increased number of grid points or spectral modes. In this study, models with a larger number $N$ incorporate smaller scale processes. However, as shown in Table RR1, increasing N reduces the magnitude of the Lyapunov exponent for smaller values of $N$. For larger $N$ values (N=150, 360, 960), the Lyapunov exponents remain the same (but why?), indicating that the inclusion of smaller scale processes does not enhance instability.

In contrast, type (2) involves small scale processes through model coupling. The reviewer notes that not only do the newly introduced small scale processes but also the coupling methodology potentially impact the system's stability. [It should be noted that in real-world models, smaller spatial and temporal scale processes associated with parameterizations typically fall into this category.]

Indeed, the reviewer has previously formulated a set of generalized Lorenz models demonstrating several key insights: (1) incorporating spectral modes and additional dissipative terms at higher wavenumbers can lead to systems that only exhibit chaotic behaviors at higher critical values of the Rayleigh parameter (e.g., Shen 2019); (2) integrating smaller-scale heating processes could lead to system destabilization (e.g., Shen 2015); (3) selecting specific spectral models, akin to model coupling, can significantly influence the stability of the system.

This is why the reviewer recommended investigating (i) the impact of the coupling coefficients (ii) the potential role of nonlinear terms in generating additional critical points. Regrettably, these comment have not been addressed adequately. To remedy this, it is advised to acknowledge the following: (a) the two types of small scale processes that may generate different feedback effects, and (b) the role of coupling as another influential factor in

determining the stability of the coupled system and the true impact of small scale processes on stability.

(As a result, the following statement in Abstract is not accurate: When studying the initial error growth, it turns out that small scale phenomena, which contribute little to the forecast product, significantly affect the ability to predict this product.)

**Response:** For the L05-1 system, it is not valid that increasing N allows the involvement of smaller scale processes. The reason is that for the L05-1 system, there is an attempt to keep 6-7 main waves and several smaller waves through the linking of $X_n$ variables. For a smaller number of variables N, the smaller waves are more pronounced and therefore the value of the Lyapunov exponent is larger for smaller N. For larger N, the ratio of major and minor waves is similar and therefore the value of the Lyapunov exponent remains the same for higher N.

Smaller scales are added to L05 systems using the procedure described on lines 731-764 (in revised manuscript):

Lorenz (2005) wanted to keep the system as simple as possible, so instead of, for example, Fourier analysis, a procedure for expressing variables $X_{tot,n}$ as sums of $X_{1,n}$ and $X_{2,n}$ was introduced:

$$X_{1,n} = \sum_{i=-I}^{I}{}'\left(\alpha - \omega|i|\right) X_{tot,n+i}, \tag{A7}$$

$$X_{2,n} = X_{tot,n} - X_{1,n}. \tag{A8}$$

Parameters $\alpha$, $\omega$, and $I$ are chosen so that $X_1$ is a low-pass filtered version of $X_{tot}$, and $X_2$ represents the difference between the full signal $X_{tot}$ and the filtered signal. By this procedure, $X_2$ has a much smaller amplitude than $X_1$, and also its time evolution should be faster since the temporal derivative is related to the spatial derivative via the difference $(X_{1,n+1} - X_{1,n-2})$, which for the low pass filtered signal $X_1$ typically is smaller than for the signal $X_2$.

More precisely, Lorenz's (2005) idea is that the parameters $\alpha$, $\omega$ are chosen so that $X_1$ equals $X_{tot}$ whenever $X_{tot}$ changes quadratically over the longitudes (variables) $n - I$ through $n + I$. It is when $\sum_{i=-I}^{I}{}'\left(\alpha - \omega|i|\right) = 1$ and $\sum_{i=-I}^{I}{}'i^2\left(\alpha - \beta|i|\right) = 0$. By solving these equations, we get:

$$\alpha = \left(3I^2 + 3\right)/\left(2I^3 + 4I\right), \tag{A9}$$

$$\omega = \left(2I^2 + 1\right)/\left(I^4 + 2I^2\right). \tag{A10}$$

The procedures (Eqs. (A4) and (A5)) are functions of the interval length $[-I, I]$.

When creating a system $dX_{tot}/dt$ as the sum of $dX_1/dt$ and $dX_2/dt$ (sum of Eqs. (A2) and (A3)), the coupling term $cX_{1,n}$ in Eq. (A3), which enables short waves to develop, is combined with the dissipation term $-X_{1,n}$ in Eq. (A2). Therefore, the coupling term can be canceled entirely, or it can appear in $X_1$ rather than $X_2$ when $X_{tot}$ is

analyzed, and there might be nothing to enable the short waves in $X_2$ to grow. Lorenz (2005) reformulated the coupling process by adding a small fraction of $X_1$ to $X_2$ so small waves in $X_2$ can amplify. It is done by replacing $b^2[X_2, X_2]_{1,n} + cX_{1,n}$ by $[X_2, X_2 + c'X_1]_{1,n}$ in Eq. (A3) , and L05-2 system would be:

$$dX_{tot,n} / dt = [X_1, X_1]_{L,n} + b^2[X_2, X_2]_{1,n} + c[X_2, X_1]_{1,n} - X_{1,n} - bX_{2,n} + F, \tag{A11}$$

where $c = c' \cdot b^2$.

Based on the L05-2 system (Eqs. (A4) - (A8)), Bednar and Kantz (2022) designed a three levels (scales) system (L05-3):

$$dX_{tot,n} / dt = [X_1, X_1]_{L,n} + b_1^2[X_2, X_2]_{1,n} + b_2^2[X_3, X_3]_{1,n} + c_1[X_2, X_1]_{1,n} + c_2[X_3, X_2]_{1,n} - X_{1,n} - b_1 X_{2,n} - b_2 X_{3,n} + F,$$

$$\tag{A12}$$

where $c_1$, $c_2$, $b_1$, $b_2$ are parameters, and the procedure for expressing the variables are:

$$X_{1,n} = \sum_{i=-I_1}^{I_1}{}' \left( \left( \left( 3I_1^2 + 3 \right) / \left( 2I_1^3 + 4I_1 \right) \right) - \left( \left( 2I_1^2 + 1 \right) / \left( I_1^4 + 2I_1^2 \right) \right)|i| \right) X_{tot,n+i}, \tag{A13}$$

$$X_{2,n} = \sum_{j=-I_2}^{I_2}{}' \left( \left( \left( 3I_2^2 + 3 \right) / \left( 2I_2^3 + 4I_2 \right) \right) - \left( \left( 2I_2^2 + 1 \right) / \left( I_2^4 + 2I_2^2 \right) \right)|j| \right) \left( X_{tot,n+j} - X_{1,n+j} \right), \tag{A14}$$

$$X_{3,n} = X_{tot,n} - X_{2,n} - X_{1,n}, \tag{A15}$$

where $I_1$ and $I_2$ set the length of the intervals $[-I, I]$.

In our case, coupling refers to the linking of different scales that allows the formation of smaller waves. It should be noted that the smaller scales are filtered out of the overall $X_{tot}$ variable by the method described by equations A4-A5 for the L05-2 system and A10-A12 for the L05-3 system, and thus this is more of a reviewer's type of process (1). The processes described in Shen (2019) and Shen (2015) would then be comparable to adding another variable that affects the variable $X_{tot}$, but which is not filtered out of $X_{tot}$. For example, if we define the constant F in L05 systems as a variable and describe it by its own ordinary differential equation dF/dt. Depending on the definition of dF/dt, we could then obtain phenomena similar to those described in Shen (2019) and Shen (2015).

It is shown in Bednar and Kantz (2022) that in the power law dEp/dt = aE$^{(1-b)}$, the coupling rate (as defined in L05 systems) is described by the value of the parameter a and does not affect the value of b. From this we conclude the general validity of the published results for different coupling rates.

We agree that the statement in the abstract may not always be valid in general and have therefore modified it. (Lines 8-9 in revised manuscript):

When studying the initial error growth, it may turns out that small scale phenomena, which contribute little to the forecast product, significantly affect the ability to predict this product.

**(Major Comment C)** This research commenced by presuming the existence of a Lyapunov exponent (LE) and proceeded to formulate various ODEs aimed at determining the predictability horizons, emphasizing the impact of (reducing) initial conditions. However, the reviewer wishes to highlight several crucial points: (1) the LE signifies a time-averaged measure; (2) all the ODEs (with b > 0) examined in this study maintain continuous dependence on initial conditions (CDIC) across infinite time intervals, which precludes them from disclosing finite predictability for chaotic solutions. Additional insights and related discussions, which are available in Shen (2024), are provided below:

(1) For a linear ODE given by E' = sigma E, the solution E grows unboundedly. Predictability horizons can be extended by reducing initial errors or elevating thresholds.

(2) In the Logistic ODE, the solution E is bounded. However, the zero state (backward in time) and saturation value (forward in time) are only asymptotically reachable, implying that predictability horizons may also be extended by reducing initial errors or raising thresholds.

(3) For the major ODE in this study, E' = a E^{1-b} with a > 0 and Eo > 0, its solution is written as follows:

E = (Eo^b + abt)^{1/b}

Please note that the above solution is (1) unbounded for b > 0 and (2) unbounded within a finite time interval (i.e., Tmax = - Eo^b / (ab)) for b < 0 and finite Eo. It appears t have been considered in this study. For instance, Equation (9) becomes invalid for b < 0.

The author appreciates the discussion using the above solution that initial growth rates might not always be exponential. However, such formulations do not yield reliable pred horizons over longer periods. More importantly, what is the relationship between the parameter "a" and the Lyapunov exponent? The same inquiry applies to the Logistic ODE: between the parameter "sigma" and the Lyapunov exponent?

**Response:** The authors and most certainly also the referee understand that the mathematical concept of the Lyapunov exponent is of limited value for the growth of forecast errors starting from finite (i.e., non-infinitesimal) perturbations and having a finite extension of the attractor. Therefore, empirical/numerical error growth is always limited to growing not beyond a finite value, and it might show behaviors different from an exponential growth also in the initial growth phase, in which case the relation to the value of the mathematically defined Lyapunov exponent of the system cannot be exact.

Our definition of the predictability horizon is the time when the mean error growth reaches 95% of a saturation value $E_{lim}$. The intrinsic predictability is then this time when the average error of initial conditions (ie(0)) goes limitingly to zero. The intrinsic predictability horizon for scale-dependent error growth was determined using an **extended** power law (Eq. (4) in the revised manuscript), where the values of the parameters a, b, $E_{lim}$ were determined from an approximation of the data and at $E_{ie}(0) \to 0$.

The "sigma" parameter in the logistic ODE approximates the largest Lyapunov exponent for scale-independent error growth. For the power law (Eq. (3) in revised manuscript) and the extended power law (Eq. (4) in revised manuscript), when describing scale-dependent error growth, the parameter $b$ is related to the largest Lyapunov exponents of each scale. The

parameter $a$ describes the degree of coupling of each scale. A detailed discussion and meaning of the parameters $a$, $b$ can be found in Bednar and Kantz (2022) in Section 3.3.1

**(Major Comment D)** While Figure RR4 suggests the smallest growth rate for the L05-1 system and the largest for the L05-3 system, this appears inconsistent with the data from Figures 5-7, show 0.46 for the L05-1, L05-2, and L05-3 systems, respectively. Could you provide any explanations for this discrepancy?

**Response:** For finite perturbations, where the initial error growth is not perfectly exponential and later some saturation occurs, there is no uniquely defined error growth rate, but only fitted values of the parameters of the different error growth laws.

Based on the reviewer's reported value of 0.46, we assume that the reviewer is considering the value of the error growth rate for exponential growth (lambda$_{ex}$) reported for the L05-3 system. Fig. 7 shows the value lambda$_{ex}$ = 0.46 1/day. (Lines 843-846 in revised manuscript)

the early part of the growth by integration of $dE_{ex}$ ( $E_{ex}$ , green, dashed) with $\lambda_{ex} = 0.46$ 1/day, integration of $dE_r$ ( $E_r$ , blue, dashed) with $\lambda_r = 0.35$ 1/day and $\beta_r = 0.07$ unit/day, integrations of $dE_p$ ( $E_p$ , red, dashed) with $a = 0.37$ unit$^{0.63}$/day and $b = 0.63$ and approximation of the full curve by integration of $dE_{qv}$ ( $E_{qv}$ , green) with $\lambda_{qv} = 0.2$ 1/day and $E_{\lim} = 6.9$ unit, integration of $dE_q$ ( $E_q$ , blue) with $\lambda_q = 0.14$ 1/day, $\beta_q = 0.17$ unit/day and $E_{\lim} = 6.9$ unit

Fig. 5 shows the value lambda$_{ex}$ = 0.33 1/day. (Lines 818-821 in revised manuscript)

the early part of the growth by integration of $dE_{ex}$ ( $E_{ex}$ , green, dashed) with $\lambda_{ex} = 0.33$ 1/day, integration of $dE_r$ ( $E_r$ , blue, dashed) with $\lambda_r = 0.32$ 1/day and $\beta_r = 0.00006$ unit/day, integrations of $dE_p$ ( $E_p$ , red, dashed) with $a = 0.34$ unit$^{0.02}$/day and $b = 0.02$ and approximation of the full curve by integration of $dE_{qv}$ ( $E_{qv}$ , green) with $\lambda_{qv} = 0.32$ 1/day and $E_{\lim} = 8.1$ unit, integration of $dE_q$ ( $E_q$ , blue) with $\lambda_q = 0.32$ 1/day, $\beta_q = 0.003$ unit/day and $E_{\lim} = 8.1$ unit

Fig. 6 shows the value lambda$_{ex}$ = 0.29 1/day. (Lines 829-833 in revised manuscript)

the early part of the growth by integration of $dE_{ex}$ ( $E_{ex}$ , green, dashed) with $\lambda_{ex} = 0.29$ 1/day, integration of $dE_r$ ( $E_r$ , blue, dashed) with $\lambda_r = 0.26$ 1/day and $\beta_r = 0.02$ unit/day, integrations of $dE_p$ ( $E_p$ , red, dashed) with $a = 0.25$ unit$^{0.32}$/day and $b = 0.32$ and approximation of the full curve by integration of $dE_{qv}$ ( $E_{qv}$ , green) with $\lambda_{qv} = 0.2$ 1/day and $E_{\lim} = 6.8$ unit, integration of $dE_q$ ( $E_q$ , blue) with $\lambda_q = 0.18$ 1/day, $\beta_q = 0.05$ unit/day and $E_{\lim} = 6.8$ unit

It can be seen that the lambda$_{ex}$ value is not the same for the L05-1, L05-2 and L05-3 systems. It is true that the lambda$_{ex}$ value is lowest for the L05-2 system and not for the L05-1 system, but this is because lambda$_{ex}$ is not a suitable indicator for scale-dependent error growth.

References:

Bednář, H., and Kantz, H.: Prediction error growth in a more realistic atmospheric toy model with three spatiotemporal scales, Geosci. Model Dev., 15, 4147–4161, https://doi.org/10.5194/gmd-15-4147-2022, 2022.